# Functional connectomics spanning multiple areas of mouse visual cortex

The MICrONS Consortium*

Understanding the brain requires understanding neurons' functional responses to the circuit architecture shaping them. Here we introduce the MICrONS functional connectomics dataset with dense calcium imaging of around 75,000 neurons in primary visual cortex (VISp) and higher visual areas (VISrl, VISal and VISlm) in an awake mouse that is viewing natural and synthetic stimuli. These data are co-registered with an electron microscopy reconstruction containing more than 200,000 cells and 0.5 billion synapses. Proofreading of a subset of neurons yielded reconstructions that include complete dendritic trees as well the local and inter-areal axonal projections that map up to thousands of cell-to-cell connections per neuron. Released as an open-access resource, this dataset includes the tools for data retrieval and analysis[1,2]. Accompanying studies describe its use for comprehensive characterization of cell types[3–6], a synaptic level connectivity diagram of a cortical column[4], and uncovering cell-type-specific inhibitory connectivity that can be linked to gene expression data[4,7]. Functionally, we identify new computational principles of how information is integrated across visual space[8], characterize novel types of neuronal invariances[9] and bring structure and function together to uncover a general principle for connectivity between excitatory neurons within and across areas[10,11].

Francis Crick wrote in 1979[12] that "It is no use asking for the impossible, such as, say, the exact wiring diagram for a cubic millimetre of brain tissue and the way all its neurons are firing". Crick's request was presumably motivated by the idea that the function of every neuron depends on its synaptic connections[13], and such dataset would allow the rigorous test and refinement of hypotheses about network anatomy. For decades, these relationships were studied through challenging single-cell experiments[14–17] or electrophysiology recordings[18,19]. Later, by combining calcium imaging with in vitro electrophysiological[20] and viral tracing methods[21] it was possible to link the functional recordings to the underlying connectivity. Much has been learned from these experiments, but they provide fragmentary information.

To realize Crick's vision, volumetric electron microscopy (EM) can be combined with calcium imaging[22,23], as demonstrated at smaller scales in the visual cortex[24–26], retina[27–29] and other systems[30,31]. Here we present a dataset (Fig. 1) that bridges neuronal function and connectivity at the cubic millimetre scale in mouse visual cortex (in vivo dimensions $1.3 \times 0.87 \times 0.82$ mm³). To measure visual responses, we performed calcium imaging of excitatory neurons across cortical layers in response to visual stimuli. To map connectivity, we imaged the same cubic millimetre with serial section transmission EM (TEM). Using scalable convolutional networks and custom computational systems, we reconstructed neurons and their synaptic connections in 3D, with extensive proofreading to ensure accuracy. Finally, we co-registered the calcium-imaging and TEM data to match neuronal responses to neurons and their connectivity.

As proofreading of the automated reconstruction continues, the dataset is becoming increasingly accurate. It includes pyramidal neurons from all layers (the following examples link to public data in Neuroglancer, our data visualization tool (https://www.microns-explorer.org/ngl-instructions), such as cortical layer 5 thick tufted (https://go.nature.com/L5tt), layer 5 near-projecting (https://go.nature.com/l5np), layer 4 (https://go.nature.com/l4) and layer 2/3 (https://go.nature.com/l2-3) neurons. It includes inhibitory neurons from many classes, such as bipolar cells (https://go.nature.com/bip), basket cells (https://go.nature.com/bkt) a chandelier cell (https://go.nature.com/cdl) and Martinotti cells (https://go.nature.com/mar). It also includes non-neuronal cells, such as astrocytes (https://go.nature.com/asc) and microglia (https://go.nature.com/mg) and the network of blood vessels (https://go.nature.com/bv). Using the interactive tools, one can visualize the input and

¹Princeton Neuroscience Institute, Princeton University, Princeton, NJ, USA. ²Electrical and Computer Engineering Department, Princeton University, Princeton, NJ, USA. ³Department of Neuroscience, Baylor College of Medicine, Houston, TX, USA. ⁴Center for Neuroscience and Artificial Intelligence, Baylor College of Medicine, Houston, TX, USA. ⁵Research and Exploratory Development Department, Johns Hopkins University Applied Physics Laboratory, Laurel, MD, USA. ⁶Allen Institute for Brain Science, Seattle, WA, USA. ⁷Department of Electrical and Computer Engineering, Rice University, Houston, TX, USA. ⁸Computer Science Department, Princeton University, Princeton, NJ, USA. ⁹Institute of Molecular Biology and Biotechnology, Foundation for Research and Technology Hellas, Heraklion, Greece. ¹⁰Brain and Cognitive Sciences Department, Massachusetts Institute of Technology, Cambridge, MA, USA. ¹¹Neuroscience Institute, Carnegie Mellon University, Pittsburgh, PA, USA. ¹²Department of Machine Learning, Carnegie Mellon University, Pittsburgh, PA, USA. ¹³Department of Computer Science, Rice University, Houston, TX, USA. ¹⁴NSF AI Institute of Artificial and Natural Intelligence, New York, NY, USA. ¹⁵Institute for Bioinformatics and Medical Informatics, University Tübingen, Tübingen, Germany. ¹⁶Institute for Computer Science, University Göttingen, Göttingen, Germany. ¹⁷Department of Ophthalmology, Byers Eye Institute, Stanford Bio-X, Wu Tsai Neurosciences Institute, Human-Centered Artificial Intelligence Institute, Department of Electrical Engineering, Stanford University, Stanford, CA, USA. ¹⁸International Max Planck Research School for Intelligent Systems, University Tübingen, Tübingen, Germany. ¹⁹School of Applied and Engineering Physics, Cornell University, Ithaca, NY, USA. ²⁰Paul Scherrer Institute, Villigen, Switzerland. *A list of authors and their affiliations appears at the end of the paper. ✉e-mail: forrestc@alleninstitute.org; nunod@alleninstitute.org; xaq@cmu.edu; clayr@alleninstitute.org; reimer@bcm.edu; sseung@princeton.edu; tolias@stanford.edu

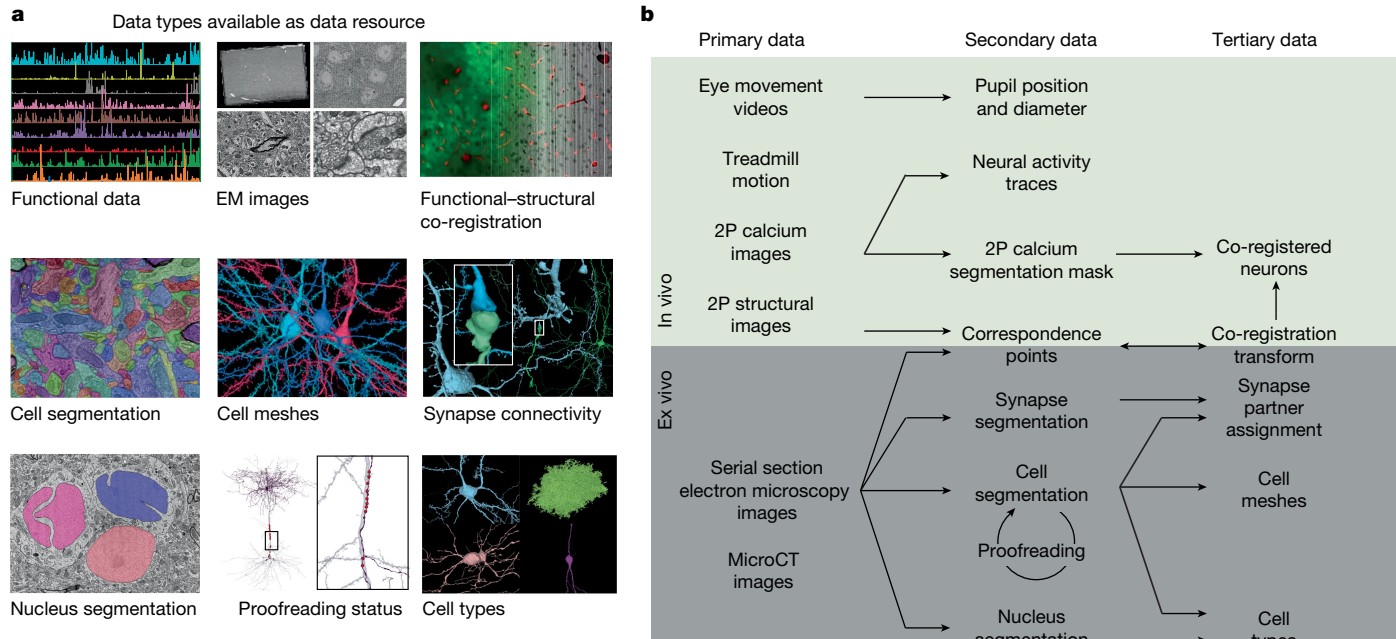

**a** Data types available as data resource

Functional data

EM images

Functional–structural co-registration

Cell segmentation

Cell meshes

Synapse connectivity

Nucleus segmentation

Proofreading status

Cell types

**b** Primary data · Secondary data · Tertiary data

**Fig. 1 | Resource data type and data products. a**, The nine data resources that are publicly available at https://www.microns-explorer.org/. **b**, Relationship between different data types. The primary in vivo data resource consists of 2P calcium images, 2P structural images, natural and parametric video stimuli used as visual input, and behavioural measurements. The secondary (derived) in vivo data resource includes the responses of approximately 75,909 pyramidal cells from cortical layer 2 to 5 segmented from the calcium videos, along with the pupil position and diameter extracted from the video of eye movements and locomotion measured on a single-axis treadmill. The primary anatomical data are composed of ex vivo serial section transmission EM images registered with the in vivo 2P structural stack. The volume includes a portion of VISp and three higher visual areas—VISlm, VISrl and VISal—for all cortical layers except extremes of layer 1. The secondary anatomical data is derived from the serial section EM image stack, and consists of semi-automated segmentation of cells, automated segmentation of nuclei, and automatically detected synapses. The tertiary anatomical data consists of assignments of the synapses to presynaptic and postsynaptic cells, triangle meshes for these segments, classification of nuclei as neuronal versus non-neuronal, and classification of neurons into excitatory and inhibitory cell classes. Secondary data for co-registration of in vivo and ex vivo images consists of manually chosen correspondence points between 2P structural images and EM images. Tertiary co-registration data are a transformation derived from these correspondence points. The transformation is then used to facilitate the matching of cell indices between the 2P calcium cell segmentation masks and the EM segmentation cells. MicroCT, micro-computed tomography.

output synapses of a single cell (https://go.nature.com/io). The database of functional recordings (https://www.microns-explorer.org/cortical-mm3#f-data) is also available for download to explore how cells responded to visual stimuli.

The first set of scientific findings emerging from the data are described in the accompanying studies. Detailed morphological and synaptic data enabled novel approaches to characterize cell types[3–7] and show that connectivity can be used to identify cell types that are difficult to identify by morphology alone[4], a recurring theme in connectomic cell typing. We also began to establish correspondences between connectivity and transcriptomics-defined cell types[7]. The combination of structural connectivity and functional similarity across thousands of pairs of individual neurons enabled a new examination of 'like-to-like' connectivity[25,32] and shows that this principle generalizes across cortical layers and visual areas[10]. This work relied on a novel approach using an artificial neural network that was trained to predict neural activities from visual stimuli[10,11]. Further linked Articles utilize this model to point the way to experimental studies of the mechanisms supporting contextual interactions[8–10] and invariances[9] in visual cortical computations.

The potential of the dataset extends far beyond these initial findings. To maximize its impact, we have made the data publicly available as a resource (https://www.microns-explorer.org/) with tools for interactive exploration and programmatic analysis. Finally, the accompanying studies highlight the tools that we developed to scale up connectomics to a cubic millimetre[1,2,11,33]. These technologies are enabling broader applications, such as reconstruction of the entire wiring diagram of a whole fly brain[34–36], the first adult connectome to be completed since that of *Caenorhabditis elegans*.

## Overview

The data were collected from a single mouse and involved a pipeline spanning three primary sites. First, two-photon (2P) in vivo calcium imaging under various visual stimulation conditions was performed at Baylor College of Medicine. Then the mouse was shipped to the Allen Institute, where the imaged tissue volume was extracted, prepared for EM imaging, sectioned and imaged over a period of six months of continuous imaging. The EM data were then montaged, roughly aligned and delivered to Princeton University, where fine alignment was performed and the volume was densely segmented. Finally, extensive proofreading was performed on a subset of neurons to correct errors of automated segmentation, and cell types and various other structural features were annotated (Fig. 2).

## 2P calcium imaging

The calcium-imaging data include the responses to visual stimuli of an estimated 75,909 excitatory neurons spanning cortical layers 2 to 5 across 4 visual areas in a transgenic mouse that expressed GCaMP6s in excitatory neurons via *Slc17a7*-Cre and Ai162. The dataset contains 14 individual scans, collected between postnatal day 75 (P75) and P81, spanning a volume of approximately 1,200 × 1,100 × 500 μm³ (anteroposterior × mediolateral × radial depth; Fig. 3a). The centre of the

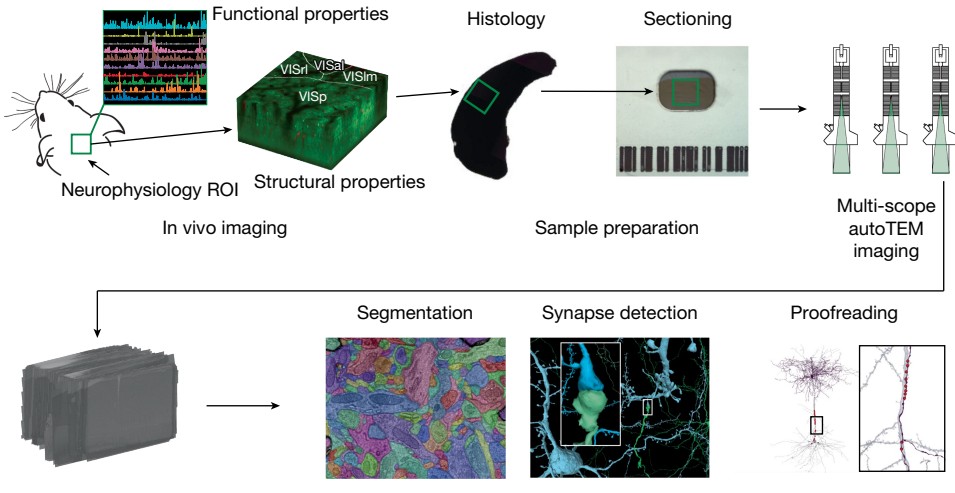

Functional properties

Histology

Sectioning

VISrl  VISal  VISlm
VISp

Neurophysiology ROI    Structural properties

In vivo imaging

Sample preparation

Multi-scope autoTEM imaging

Segmentation    Synapse detection    Proofreading

Volume assembly

Reconstruction

**Fig. 2 | Major experimental steps in the data acquisition workflow.** Outline of the major sequential steps used to generate the MICrONS dataset. First, in vivo measurements of neuronal functional properties are acquired from a region of interest (ROI) in the mouse visual cortex. In addition, a spatial overlapping in vivo structural image stack is collected to facilitate later registration with postmortem data. Following fixation of the brain, the tissue encompassing the functional ROI is processed for histology and sectioned. These sections are then imaged by TEM, and the resulting images are assembled into a 3D volume. Automated methods subsequently reconstruct the cellular processes and synapses within this volume, and the automated reconstructions are proofread as needed to ensure accuracy for further analysis. Image panels are adapted from Yin et al.[63], Springer Nature Limited, and mouse and autoTEM drawings are adapted from Mahalingam et al.[64], CC BY 4.0 (https://creativecommons.org/licenses/by/4.0/).

volume was placed at the junction of primary visual cortex (VISp) and three higher visual areas—lateromedial area (VISlm), rostrolateral area (VISrl) and anterolateral area (VISal)—in order to image retinotopically matched neurons that were potentially connected via inter-areal feed-forward and feedback connections.

Each scan consisted of two adjacent overlapping 620-µm-wide fields at multiple imaging planes, imaged with the wide field of view (FOV) of the 2P random access mesoscope (2P-RAM). The scans ranged up to approximately 500 µm in depth, with a target spacing of 10–15 µm to maximize the coverage of imaged cells in the volume (Fig. 3b,c). For 11 of the 14 scans, 4 imaging planes were distributed widely in depth using the mesoscope remote focus, spanning roughly 300–400 µm with an average spacing of approximately 125 µm between planes for near-simultaneous recording across multiple cortical layers. In the remaining 3 scans, fewer planes were imaged at 10–20 µm spacing to achieve a higher effective pixel density (Extended Data Table 1). These higher-resolution scans were designed to be amenable to future efforts to extract signals from large apical dendrites from deeper layer 5 and layer 6 neurons. However, for this release, imaging data were automatically segmented only from somas using a constrained non-negative matrix factorization approach and fluorescence traces were extracted and deconvolved to yield activity traces. In total, 125,413 masks were generated across 14 scans, of which 115,372 were automatically classified as somatic masks by a trained classifier (Fig. 3d).

The functional data collection relied on newly established technologies, especially the 2P-RAM mesoscope. In addition, we developed an imaging workflow with the goal of full coverage within the target volume. This required several optimizations—for example, to densely target scan planes across multiple days, we needed a common reference frame to assess the coverage of scans within the volume. Therefore, in addition to the functional scans, high-resolution (0.5–1.0 pixels per µm) structural volumes were acquired for registration with the subsequent EM data. At the end of each imaging day, individual imaging fields of the functional scans were independently registered into a structural stack (Fig. 3b,c). This enabled us to target scans in subsequent sessions to optimize coverage across depth. On the last day of imaging, a 2-channel (green, red) 1,412 × 1,322 × 670 µm³ (anteroposterior × mediolateral × radial depth) structural stack was collected at 0.5 pixels per µm after injection of fluorescent dye (Texas Red) to label vasculature, enhancing fiducial labelling for co-registration with the EM volume (Fig. 3a).

After registration of the functional imaging field with the structural stack, 2D centroids from the segmentation were assigned 3D centroids in the shared structural stack coordinate space, on the basis of a greedy assignment of 3D proximity. Based on this analysis, we estimate the functional imaging volume contains 75,909 unique functionally imaged neurons consolidated from 115,372 segmented somatic masks, with many neurons imaged in 2 or more scans.

## Behavioural tracking and visual stimulation

During imaging, the mouse was head-restrained, and the stimulus was presented to the left visual field. Treadmill rotation (single axis) and video of the left eye were captured throughout the scan, yielding locomotion velocity, eye movements and pupil diameter data.

The stimulus for each scan lasted approximately 84 min, and consisted of naturalistic (complex scenes with real-world statistics) and parametric (simpler, artificially generated) video stimuli. The majority of the stimulus (64 min) was made up of 10 s clips drawn from films, the Sports-1M dataset[37] or rendered first-person point of view (POV) movement through a virtual environment (Fig. 3e). Our goal was to approximate natural statistical complexity to cover a sufficiently large feature space. These data can support multiple lines of investigation, including applying deep learning-based systems identification methods to build highly accurate models that predict neural responses to arbitrary visual stimuli[11,38]. These models enable a systematic characterization of tuning functions with minimal assumptions relative to classical methods using parametric stimuli[38].

The stimulus composition included a mixture of unique stimuli for each scan, some that were repeated across every scan, and some that were repeated within each scan. In particular, 6 natural film stimuli clips totalling 1 min (oracle natural videos) were repeated in the same order 10 times per scan, and were used to evaluate the reliability of the neural responses to repeated visual stimuli (Fig. 3f). Variations in this 'oracle score' from scan to scan serve as an important indicator of scan quality, since reliable responses are not observed when

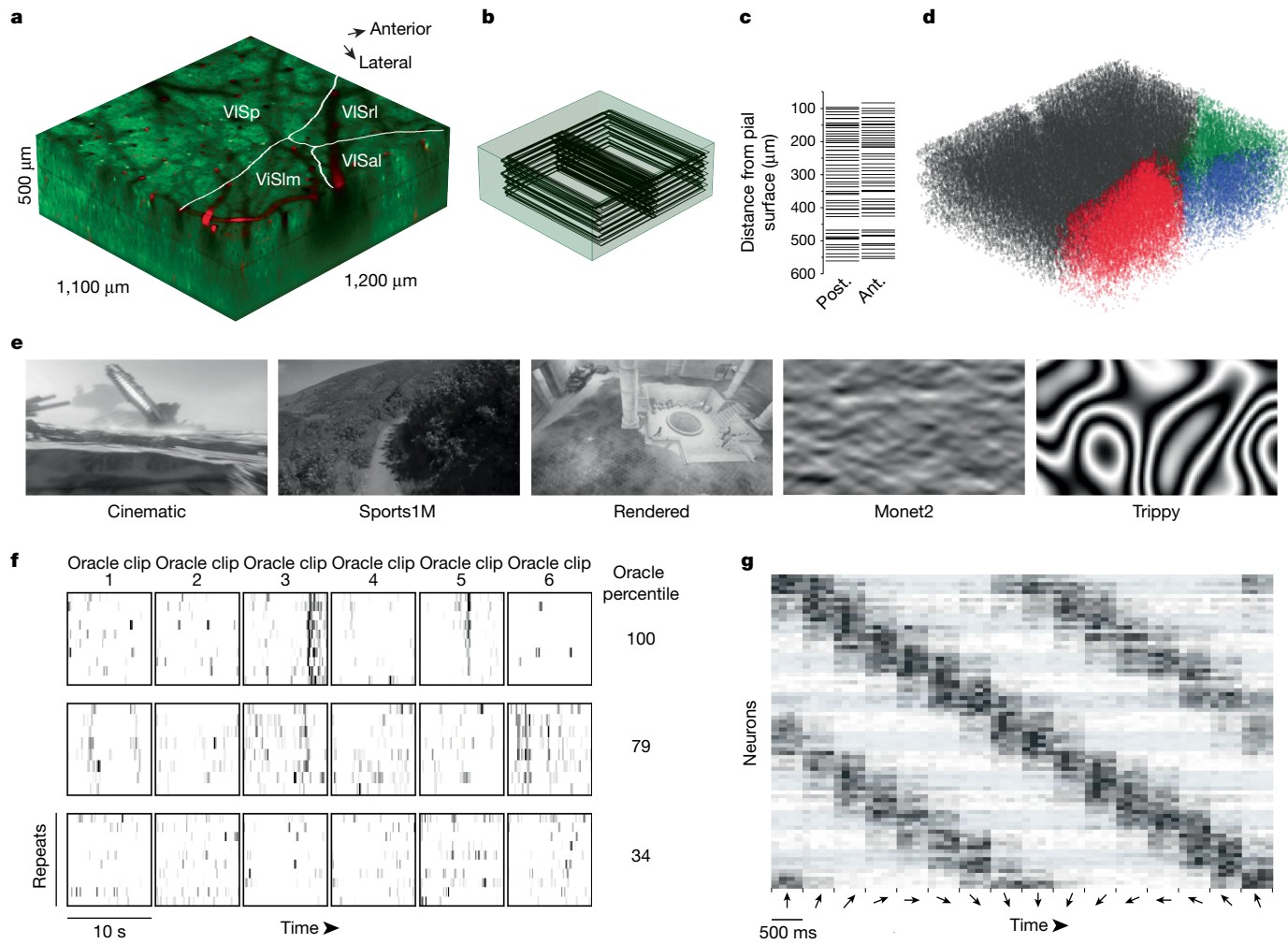

**Fig. 3 | In vivo calcium-imaging data. a**, Representation of the 2P functionally imaged volume with area boundaries (white) and vascular label from structural stack (red). **b**, Wireframe representation of 104 planes registered in the structural 2P stack. **c**, Mean depth of posterior (post.) and anterior (ant.) registered fields relative to the pial surface. **d**, 3D scatter plot of each functional mask in its registered location in the structural 2P stack. Black, VISp; red, VISlm; blue, VISal; green, VISrl. **e**, Example frames from each of the five stimulus types (cinematic, Sports-1M, rendered, Monet2 and Trippy) shown to the mouse.

**f**, Raster of deconvolved calcium activity for three neurons to repeated stimulus trials (oracle trials; ten repeats of six sequential clips, with each repeat normalized independently). Rasters for high (top), medium (middle) and low (bottom) oracle scores with the percentile shown on the right. **g**, Trial-averaged raster (central 500 ms of trial-average raster for each direction, out of 937 ms) of deconvolved calcium activity for 80 neurons in 40 Monet2 trials (16 randomly ordered directions) grouped by preferred direction (5 neurons per direction; alternating blue shading) and sorted according to the stimulus directions.

imaging conditions are poor or the mouse is not engaged with the stimulus.

To relate our findings to previous work, we also included a battery of parametric stimuli (Monet2 and Trippy, 10 min each; Methods, 'Stimulus composition') that were generated to produce spatially decorrelated stimuli that were suitable for characterizing receptive fields while also containing local or global directional and orientation components for extracting basic tuning properties such as orientation selectivity (Fig. 3e,g).

## The EM volume

After the in vivo neurophysiology data collection, we imaged the same volume of cortex ex vivo using TEM, which enabled us to map the connectivity of neurons for which we measured functional properties. These required considerable scaling from previous state-of-the-art datasets, with particular emphasis on automation and on reducing rare but potentially catastrophic events that could incur loss of multiple serial sections.

The tissue sample was trimmed and sectioned into 27,972 serial sections (nominal thickness 40 nm) onto grid tape to facilitate automated

imaging. Although the cutting was automated, it was supervised by humans who worked in shifts around the clock for 12 days. They were ready to stop and restart the ultramicrotome immediately if there was a risk of multiple section loss. As will be described later, the EM dataset is subdivided into two subvolumes owing to sectioning and imaging events (details of sectioning timeline and artefacts are presented in Methods).

A total of 26,652 sections were imaged by 5 customized automated TEMs (autoTEMs), which took approximately 6 months to complete and produced a dataset composed of 2 Pb of raw data at a resolution of approximately 4 nm (Fig. 4d–h).

An 800-µm region (sections 7,931–27,904) (Fig. 4a) was selected for further processing, as it had no consecutive section loss and an overall section loss of around 0.1%. This region contains approximately 95 million individual tiles that were stitched into 2D montages per section and then aligned in 3D. Owing to the re-trimming of the block and the requirement for a knife change (Methods), the EM data are divided into two subvolumes (Fig. 4a). One subvolume contains approximately 35% of the sections (sections 7,931–14,815) and the other contains 65% of the sections (sections 14,816–27,904). The two subvolumes were

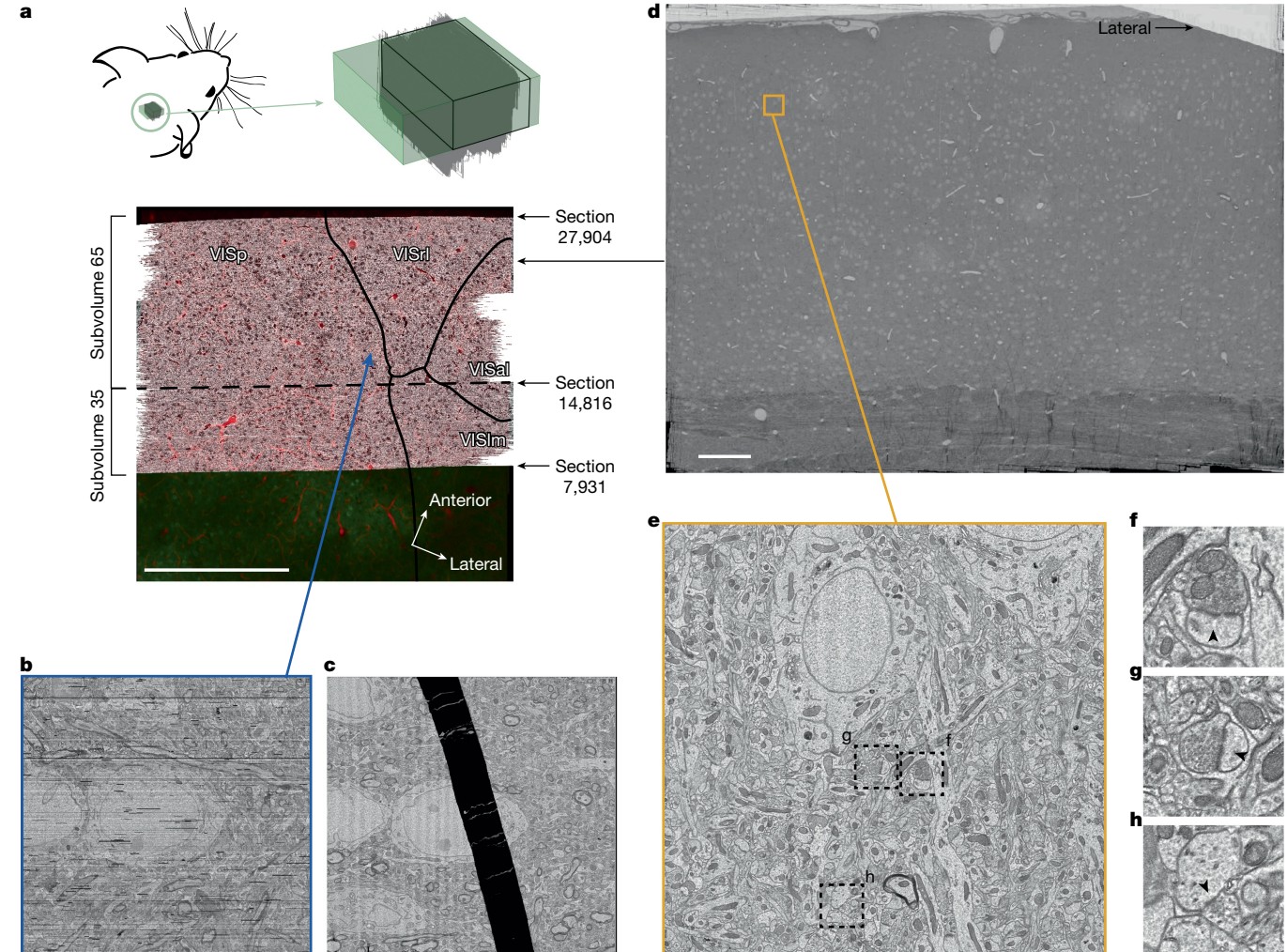

**Fig. 4 | EM dataset. a**, Top view of EM dataset (grey) registered with the in vivo 2P structural dataset (vasculature in red and GCaMP in green). Area borders calculated from calcium imaging are shown as black lines. The two portions of the dataset are separated by a dashed line. Scale bar, 500 μm. Mouse drawing adapted from from Mahalingam et al.[64], CC BY 4.0 (https://creativecommons.org/licenses/by/4.0/). **b,c**, Top view of small region showing the quality of the fine alignment and its robustness to large folds shown in **c** (the dataset is available at https://ngl.microns-explorer.org/#!gs://microns-static-links/mm3/data_fig/4b.json). Scale bars, 5 μm. **d**, Montage of a single section showing the coverage from pia to white matter and across three different cortical regions. Scale bar, 100 μm. **e**, Example of a single tile from the section shown in in **d**, with dashed squares representing the locations in **f**–**h**. Scale bar, 5 μm. **f,g**, Examples of excitatory synapses indicated with arrowheads (dataset available at https://ngl.microns-explorer.org/#!gs://microns-static-links/mm3/data_fig/4f.json (**f**) and https://ngl.microns-explorer.org/#!gs://microns-static-links/mm3/data_fig/4g.json (**g**)). **h**, Example of an inhibitory synapse (arrowhead) (dataset available at https://ngl.microns-explorer.org/#!gs://microns-static-links/mm3/data_fig/4h.json).

processed individually and later aligned to each other in the same global coordinate frame, enabling the tracing of axons and dendrites across their border (Fig. 5). To facilitate the reconstruction process across the division between the two subvolumes, a composite image of the partial sections was created at the interface. However, the two subvolumes were reconstructed separately and each has a distinct representation in the analysis infrastructure and database.

Accurate reconstruction requires extremely accurate stitching and alignment of images with hundreds of thousands of pixels on a side. To achieve this at petabyte scale, we split the process into distinct coarse and fine pipelines. For the coarse pipeline, sections were initially stitched using a per image affine transformation, and a polynomial transformation model was applied to a subset of sections whose stitching quality had a local misalignment error of more than five pixels. Down-sampled 2D stitched sections were then roughly aligned in 3D. The rough alignment process ensured global consistency within the dataset and accounted for images from multiple autoTEMs with varied image sizes and resolutions. It is also corrected for locally varying misalignments such as scale differences and deformations between sections and aids the fine alignment process.

To further refine image alignment, we developed a set of convolutional networks to estimate pixel-wise displacement fields between pairs of neighbouring sections[33]. This process was able to correct non-linear misalignments around cracks and folds that occurred during sectioning. Although this fine alignment does not restore the missing data inside a fold, it was still effective in correcting the distortions caused by large folds (Fig. 4b,c), which caused large displacements between sections and were the main cause of reconstruction errors. Although imaging was performed with 4 nm resolution, the aligned imagery volume was generated at 8 nm resolution to decrease data size for subsequent processing.

## Automated reconstruction

We densely segmented cellular processes across the volume using affinity-predicting convolutional neural networks and mean affinity

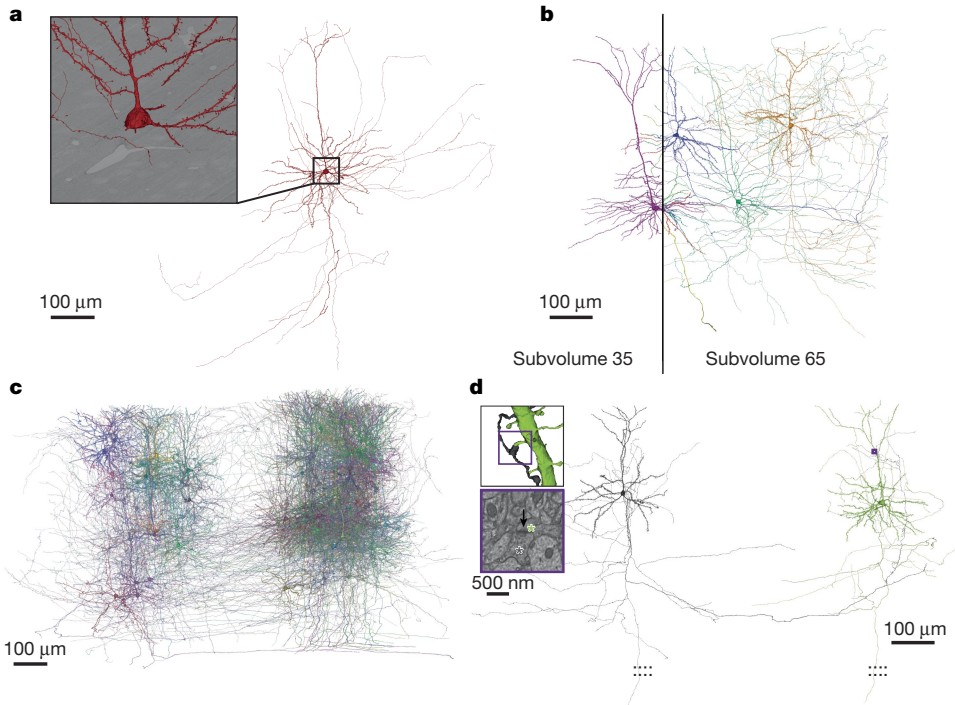

**Fig. 5 | Reconstruction. a**, A pyramidal cell reconstructed from the EM images (inset). **b**, Pyramidal cells from both subvolumes as they cross the subvolume boundary. **c**, A selection of 78 proofread pyramidal cells from subvolume 65. **d**, A distant pair of pyramidal cells connected by a synapse within subvolume 65.

agglomeration (Fig. 5a) Segmentation was not attempted where the alignment accuracy was deemed insufficient or tissue was missing or occluded over multiple sections.

The automatic segmentation produced highly accurate dendritic arbors before proofreading, enabling morphological identification of broad cell types. Most dendritic spines are properly associated with their dendritic trunk. Recovery of larger-caliber axons, those of inhibitory neurons, and the initial portions of excitatory neurons was also typically successful. Owing to the high frequency of imaging defects in the shallower and deeper portions of the dataset, processes near the pia and white matter often contain errors. Many non-neuronal objects are also well-segmented, including astrocytes, microglia and blood vessels. The two subvolumes of the dataset were segmented separately, but the alignment between the two is sufficient for manually tracing between them (Fig. 5b).

Nuclei were also automatically segmented ($n = 144,120$) within subvolume 65 using a distinct convolutional network[33]. To use nucleus shape to map cell classes across the dataset, we manually labelled a subset of the 2,751 nuclei in a 100-µm-square column of the dataset as non-neuronal, excitatory or inhibitory. We then developed machine learning models to automate distinguishing neurons from non-neuronal cells such as glia, as well as to classify cells at different levels of resolution[2,6] within the subvolume with high accuracy (Methods). The results of this nucleus segmentation, manual cell classification and model building are provided as part of this data resource.

Synaptic contacts were automatically segmented in the aligned EM image, and the presynaptic and postsynaptic partners from the cell segmentation were automatically assigned to identify each synapse (Fig. 5d). We automatically detected and associated a total of 524 million synaptic clefts across both subvolumes (subvolume 35: 186 million, subvolume 65: 337 million). We manually identified synapses in 70 small subvolumes ($n = 8,611$ synapses) distributed across the dataset, giving the automated detection an estimated precision of 96% and recall of 89% (Extended Data Fig. 1). We estimated partner assignment accuracy at 98% from a separate dataset

of manually annotated synapses ($n = 191$) that were held-out from training.

## Proofreading

Although the automated segmentation creates impressive reconstructions, proofreading is required to make those reconstructions more complete and accurate. The proofreading process involves merging additional segments of the neurons that were missing in the reconstruction, and splitting segments that were incorrectly associated with a neuron. To perform real-time collaborative proofreading in a petascale dataset, we developed the ChunkedGraph proofreading system[1] that can be used with Neuroglancer as a user interface or a REST (representational state transfer) application programming interface (API) for computationally driven edits. This flexibility enabled the proofreading methods to be tailored to different scientific needs, including manual, semi-automated and automated proofreading. Note that all proofreading was performed in subvolume 65.

The released segmentation now contains all 1,046,656 edits of the proofreading that had occurred as of 16 September 2024 and is being updated quarterly. Proofreading was performed by individual scientists and focused teams of proofreaders to both support targeted scientific discovery for companion studies[3–7,10] and correct errors that most affected general connectivity. Because of this, the level of completeness differs across these cells (Fig. 5), as neurons have been proofread as part of multiple Machine Intelligence from Cortical Networks (MICrONS) data analysis projects. For example, in the functional connectomics study, we proofread the full extent of axonal and dendritic arbors of 85 excitatory neurons within subvolume 65 (Fig. 5c), whereas for a broad columnar sample only the dendrites of 1,188 excitatory neurons were proofread. The result is a wide variation in edits per neuron with more edits generally corresponding to more extensive axons (100–1,000 edits per axon) (Extended Data Fig. 2). The most time-consuming task is extending axons, and thus this is where the data varies most across cells and studies. In total, the released dataset includes 1,433 neurons that

have proofread axons with varying levels of extension, where all incorrect mergers have been removed and many false splits corrected. From the proofread dendrites, we determined that 99% of inputs were correct when assigned to a postsynaptic soma in the automated segmentation. As a result, for the neurons with proofread axons all synapses—both input and output—are now correctly associated. A full-time proofreader can generate between 400–600 axon extension edits in a work week. The proofread excitatory neurons contain some of the most extensive axonal arbors reconstructed in the neocortex at EM resolution, with the longest excitatory axon measuring 18.9 mm with 2,483 synaptic outputs and inhibitory axons ranging from in length from 1.1 to 32.3 mm with a mean of 2,754 synaptic outputs (range 99–14,019) (Fig. 5f). In general, inhibitory axons were more complete in the automated reconstruction, probably because their axons are slightly thicker than those of most excitatory axons.

In addition to proofreading axons and dendrites, we made widespread edits to enhance the general dataset quality. Following the automated segmentation, there were 7,050 segmented objects consisting of a total of 17,753 neurons that were merged together (based on nucleus segmentation), preventing analysis of these cells. Using a combination of manual and automated error-detection workflows, we have split almost all neurons into single-soma objects, bringing the total number of individually segmented neurons to 84,035 (Extended Data Fig. 3). To work through such dataset-wide tasks more quickly, we developed and validated an automated error-detection and correction workflow using graph and morphological analysis to identify merge error locations and generate edits that could be executed using PyChunkedGraph (PCG)[1]. This automated approach (NEURD) was also used to remove false axon merges onto dendritic segments and split axon branches with abnormally high degree across the dataset[2], totalling more than 164,000 edits.

Proofreading is ongoing in the dataset with regular public updates, and there is now a project called the Virtual Observatory of the Cortex (https://www.microns-explorer.org/vortex) funded by the National Insitutes of Health (NIH), to which individual researchers can submit scientific requests to steer proofreading and annotation of the dataset in directions that will move their research questions forward.

## Functional–structural co-registration

Functional connectomics requires that cells are matched between the 2P calcium-imaging and EM coordinate frames. We achieved this using a three-phase approach combining expert annotations and automatic methods. In the first step, we generated a co-registration transform using a set of 2,934 expert-matched fiducials between the EM volume and the 2P structural dataset (1,994 somata and 942 blood vessels, mostly branch points, which are available as part of the resource; Methods). To evaluate the error of the transform we evaluated the distance in micrometres between the location of a fiducial after co-registration and its original location; a perfect co-registration would have residuals of 0 µm. The average residual was 3.8 µm.

For the second step we used the results of the transform to guide a group of experts to manually match 19,181 functional ROIs from 14 scans to 15,439 individual EM neurons (multiple functional ROIs can match to a single EM neuron if it was present in multiple scans). The results of manual matching provide both high-confidence matches for analysis and 'ground truth' for fully automated approaches. These results help to validate the first phase, as most matched ROIs have low residuals and high separation scores (Extended Data Fig. 4). Furthermore, as expected for successful matches, ROIs with at least moderate visual responses that are independently matched to the same neuron across multiple scans have higher signal correlations than adjacent neurons (Extended Data Fig. 4).

In the third and final step, we used two automated approaches to match the entire set of functional ROIs. The first approach used the EM-to-2P co-registration transform to move the centroids of all EM neurons (predicted from nucleus detections) to the 2P coordinate space, and then used minimum weight matching for bipartite graphs to assign functional ROIs to EM neurons. This method (referred to as the fiducial-based automatch table) resulted in 84,198 functional ROIs matched to 37,364 EM neurons. Considering all matches, this method achieved 83% precision relative to manual matchers, but filtering out matches in the bottom 30% of separation scores yields 90% precision, while still including 59,934 functional ROIs and 31,042 EM neurons. (Extended Data Fig. 5). The second automated approach used only the EM and 2P blood vessel segmentations to generate a novel co-registration between the two volumes, using a fine-scale deformable B-spline-based registration. Then, minimum weight matching for bipartite graphs was used to assign functional ROIs to EM neurons. This table (referred to as the vessel-based automatch table) contains 75,856 functional ROIs matched to 34,712 EM neurons. Remarkably, this fiducial-free method performed as well as the fiducial-based method, achieving 84% precision with manual matches. Filtering out matches in the bottom 30% of separation scores yielded 90% precision, while including 53,248 functional ROIs and 28,233 EM neurons (Extended Data Fig. 5). Finally, we tested whether taking only the matches for which both automated methods agree would increase the performance relative to manual matches. Indeed, this hybrid automated table achieves 89% agreement with no additional filtering, yielding 60,091 functional ROIs and 29,620 EM neurons (Extended Data Fig. 5).

## Integrated analysis

To create a resource for the neuroscience community, we have made the data from each of the steps described above—functional imaging, the EM subvolumes, segmentation and a variety of annotations—publicly available on the MICrONS Explorer website (https://www.microns-explorer.org/). From the site, users can browse through the large-scale EM imagery and segmentation results using Neuroglancer (https://github.com/google/neuroglancer); several example visualizations are provided to get started. All data are served from publicly readable cloud buckets hosted through Amazon Web Services (AWS) and Google Cloud Storage.

To enable systematic analysis without downloading hundreds of gigabytes of data, users can selectively access cloud-based data programmatically through a collection of open source Python clients (Extended Data Table 2). The functional data, including calcium traces, stimuli, behavioural measures and more, are available in a DataJoint database that can be accessed using DataJoint's Python API (https://datajoint.com/docs/), or is available as neurodata without borders (NWB) files on the Distributed Archives for Neurophysiology Data Integration (DANDI) Archive (https://dandiarchive.org/dandiset/000402). EM imagery and segmentation volumes can also be selectively accessed using cloud-volume (https://github.com/seung-lab/cloud-volume), a Python API that simplifies interacting with large-scale image data. Mesh files describing the shape of cells can be downloaded with cloud-volume, which also provides features for convenient mesh analysis, skeletonization and visualization. These meshes can be decomposed and richly annotated for automated proofreading and morphological analysis of processes and spines using NEURD[2] (https://github.com/reimerlab/NEURD). Annotations on the structural data, such as synapses and cell body locations, can be queried via CAVE client, a Python interface to the Connectome Annotation Versioning Engine (CAVE) APIs (Fig. 6a,b). CAVE encompasses a set of microservices for collaborative proofreading and analysis of large-scale volumetric data.

The first collection of annotation tables available through CAVE client focus on the larger subvolume of the dataset, which we refer to within the infrastructure as Minnie65, and which has been the current focus of proofreading and ongoing analysis (Extended Data Table 3). The largest table describes connectivity, contains all 337.3 million synapses and

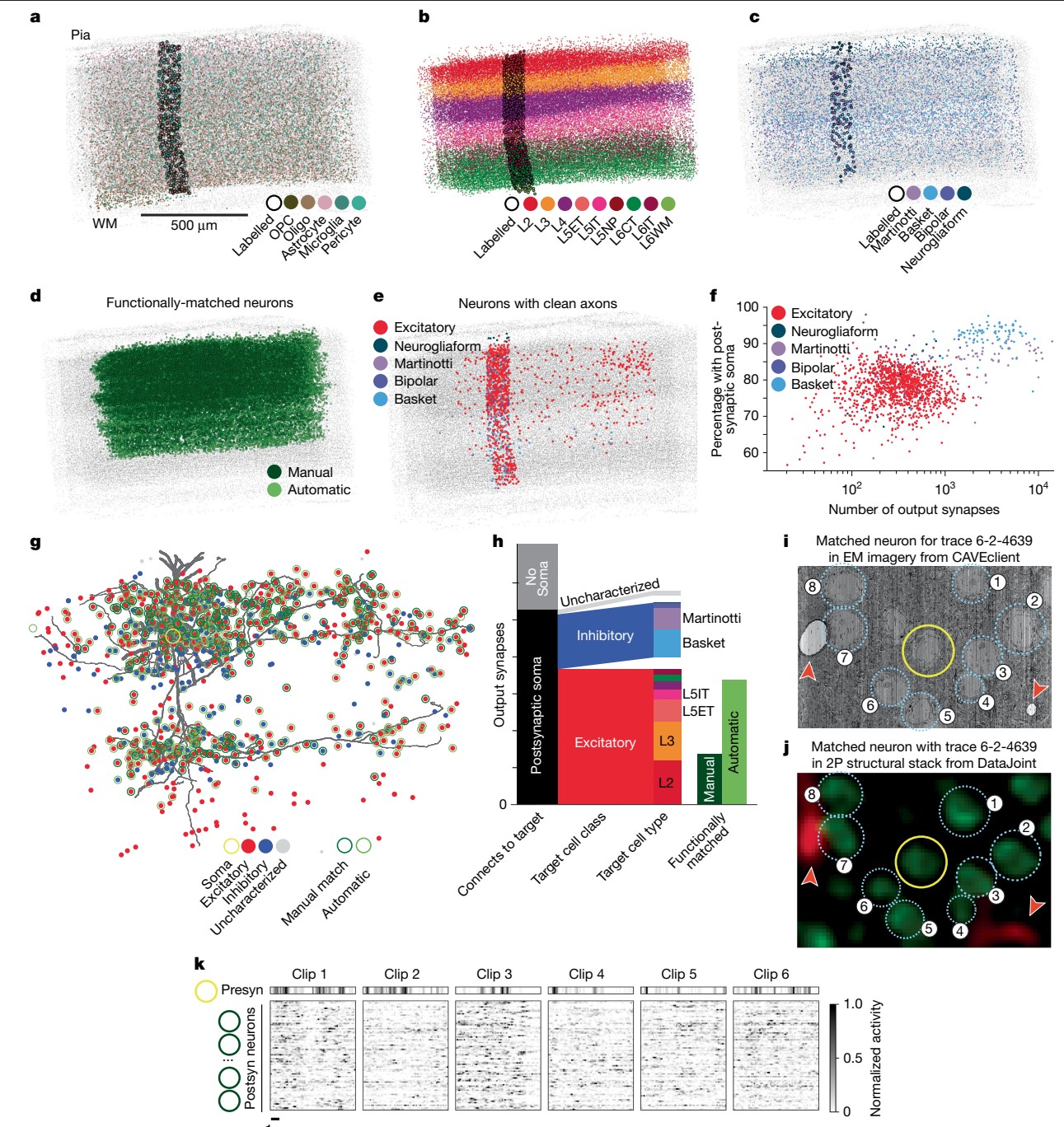

**Fig. 6 | Integrated analysis resources and examples. a–e,** Cell body locations and cell-are type classifications, all nucleus detections shown in light grey. **a,** Non-neuronal cells, manually typed (dark outlines) and classifier-based (no outline)[6]. OPC, oligodendrocyte precursor cell. **b,** Excitatory cells, labelled by unsupervised clustering of morphological features[4] (dark outline) and a model based on those labels[6]. L2, layer 2; L3, layer 3; L4, layer 4; L5ET, layer 5 extratelencephalic; L5IT, layer 5 intratelencephalic; L5NP, layer 5 near-projecting; L6CT, layer 6 cortico-thalamic; L6IT, layer 6 intratelencephalic; L6WM, layer 6 white matter. **c,** Inhibitory cells, classified by human experts[4] and trained models[6]. **d,** Neurons registered to in vivo functional traces. **e,** Proofreading status of neurons in subvolume 65: black dots (fully proofread), red (cleaned of false merges but potentially incomplete) and blue (dendrites cleaned/extended). **f,** The number of output synapses per neuron shown in **e** versus the fraction mapped to a single postsynaptic soma, coloured by cell class. **g,** A fully proofread pyramidal cell (nucleus ID: 294657, segment ID: 864691135701676411) with postsynaptic soma locations shown as coloured dots (by cell class). Cells with functionally co-registered regions are outlined in dark green. **h,** Quantification of synapses associated with different categories of postsynaptic cells. The first column shows the fraction that map to a single postsynaptic soma. The second column shows the fraction of those that are excitatory or inhibitory. The third column shows the fraction of cells that are in each sub-class based on the model shown in **b,c.** The fourth column shows the proportion that map to functionally co-registered cells. The cell and its synapses are viewable at https://neuroglancer-demo.appspot.com/#!gs://microns-static-links/mm3/data_fig/6f.json. **i,** EM image (**i**) and corresponding image from the 2p structural stack (**j**) centred on the cell shown in **g** (yellow circle). Red arrowheads indicate blood vessels. **k,** Functional responses of the presynaptic (presyn) neuron (**g**; yellow) and its functionally co-registered postsynaptic (postsyn) targets. Heat maps show average Δ*F*/*F* traces for the presynaptic neuron and postsynaptic targets, sorted by synaptic strength, in response to oracle clips from functional scans.

is searchable by presynaptic ID, postsynaptic ID and spatial location. In addition, there are several tables that describe the soma location of key cells, predictions for which cells are different non-neuronal (Fig. 6a), excitatory (Fig. 6b) and inhibitory (Fig. 6c) types. There are also annotations that denote which cells have been functional co-registered (Fig. 6d) and which cells have been proofread to different degrees of completion (Fig. 6e). In this release, the only table available for Minnie35 contains synapses, as its segmentation and alignment occurred later and little proofreading, annotation or analysis has been conducted within it. We expect that continued proofreading and analysis of the data will lead to updated and additional tables for both portions of the data in future data releases.

This collection of tools and public data enables analyses that integrate questions of connectivity, morphology and functional properties of neurons. Here, we provide an example to suggest how the data might be used together. The power of the dataset lies in the fact that when an axon is proofread, it contains hundreds to more than ten thousand output synapses (Fig. 6f). Furthermore, between 60 and 95% of those outputs can be accurately mapped onto their postsynaptic targets with a known soma location, depending on the cell type and its spatial location in the volume (Fig. 6f). This is because the segmentation is highly accurate for dendritic inputs, with a 99% input precision based on comparing proofread with non-proofread dendrites. To seed an analysis with an as-complete-as-possible cell, one might begin by using the proofreading table to identify a neuron with complete axons and dendrites and querying for all the synaptic inputs and outputs for the cell, in this case a L2/3 cell in VISp (Fig. 6g). For this particular proofread neuron, 74.5% (1,053 out of 1,412 synapses) are onto objects with a single nucleus (as determined from automated detection), with 275 synapses onto cells classified as inhibitory, 662 synapses onto cells classified as excitatory, and 116 synapses onto cells whose soma did not pass classification quality control (Fig. 6h). The remainder (25.4% 359 out of 1.412 synapses) are onto orphan fragments, composed of a mix of disconnected spine heads and stretches of dendrite. By filtering the synaptic targets with functionally matched neurons (Fig. 6k), one can further identify which targets have been matched to the functional experiments (365 out of 1,412) and use DataJoint to query the functional data or read NWB files deposited in the DANDI data archive (Fig. 6i–k). In this case, the targets include pyramidal cells in both L2/3 and L5. Subsequent investigation could examine the morphology of such cells in detail, or consider functional responses of their targets. We have provided example notebooks that walk through the above examples and more to help users get started. Together, these data provide a platform for analysis of the relationship between the synaptic structure, neuronal morphology and functional tuning of mouse visual circuits.

## Cell types

Connectivity and morphology are key properties of cell types, and the scale of this dataset enables an unprecedented exploration of the anatomical diversity of cortical neurons as well as a need to relate known cell types to EM data. We have taken multiple approaches to addressing these challenges in the accompanying studies. Two projects[3,4] applied data-driven methods to dendritic reconstructions to characterize excitatory neurons across cortical depth and visual areas, revealing intralaminar subtypes and inter-areal differences in populations. Another study linked transcriptomic types of inhibitory neurons to EM reconstructions, establishing a proof of concept for linking molecular cell types to anatomical cell types that use morphology and synapse connectivity[7]. Although these studies used proofread or post-processed neuronal reconstructions, not all segmented neurons in the dataset were amenable to such analysis due to truncation by dataset boundaries or segmentation quality. To push cell typing even in such difficult cases, a fourth study showed that key features of the soma and nucleus of a cell alone was sufficient to predict cell classes such as glia,

excitatory neuron or inhibitory neuron, as well as subclasses such as basket cells versus bipolar cells or microglia versus oligodendrocytes, or identify similar cells to a cell of interest[6]. Together, these approaches enable matching known cell types with EM neurons and using the EM data to discover new cell types.

The integration of cell-type classifications with additional modalities enables a powerful set of tools for discovery. Examining the output of proofread neurons, which includes more than 900,000 synaptic connections between neurons, reveals key differences in the interlaminar communication between excitatory and inhibitory neurons (Fig. 7a–c). The size of the dataset also allows for a comprehensive analysis of cell-type connectivity, including tracing across one or more steps along the synaptic network. A major finding from multiple studies of the MICrONS dataset is the widespread specificity of connectivity exhibited by various inhibitory[4,7] and excitatory[5] cell types. As an example of such analysis, we can follow a collection of layer 3 pyramidal neurons and compare their first-order (direct) connectivity onto excitatory cell types and inhibitory neurons as well as the second-order (two-hop) connectivity of those inhibitory neurons that are targeted by the layer 3 cells (Fig. 7d).

## Discussion

EM is widely recognized as the gold standard for identifying structural features of synapses, and most datasets, including the output of the MICrONS project, were primarily created to answer questions related to circuit-level connectivity. Regardless of the original intent, the scale and high resolution of the MICrONS dataset offers information that is far richer and of broader interest than just connectivity. For example, the imagery also reveals the intracellular machinery of cells, including the morphology of subcellular structures such as the nucleus, mitochondria, endoplasmic reticulum and microtubules. Furthermore, the segmentation includes non-neuronal cells such as microglia, astrocytes, oligodendrocyte precursor cells and oligodendrocytes, as well as the fine morphology of the cortical vasculature.

### Advances and limits in large-scale EM

The scale of large functional and EM datasets presents a wealth of opportunities for analysis and discovery. With advances in microscopy and computing power, it is now possible to work with datasets that are orders of magnitude larger than just a few years ago with millions of synapses and tens of thousands of recorded neurons. Among the key opportunities presented by this data is the ability to identify patterns and trends that may be hidden in smaller datasets, the ability to identify and validate general principles at a larger scale, and the ability to perform more sophisticated analyses—since with more data, it is possible to use more complex algorithms and models including machine learning techniques. The accessibility of these datasets also enhances hypothesis-driven approaches by enabling scientists to investigate whether specific types of connectivity exist among different cell types of interest. Additionally, the scale of the data and the availability of exploration tools to facilitate the discovery of anomalies or contradictions to current hypotheses and provide opportunities to address and resolve them effectively. Both of these approaches can help to identify patterns and trends that would be difficult to observe using smaller datasets.

However, larger datasets have limitations and challenges associated with them. When analysing the connectivity graph, it is essential to keep in mind that although, as shown by our results, the automatic segmentation of dendritic inputs is highly accurate, the automatic segmentation of axons is not as accurate. Therefore, it is essential to be aware of which processes have been proofread and to what extent. Additionally, it is worth considering that although each neuron in the dataset receives thousands of inputs, a percentage of synapses in the dataset are on detached spines. Depending on the scientific question

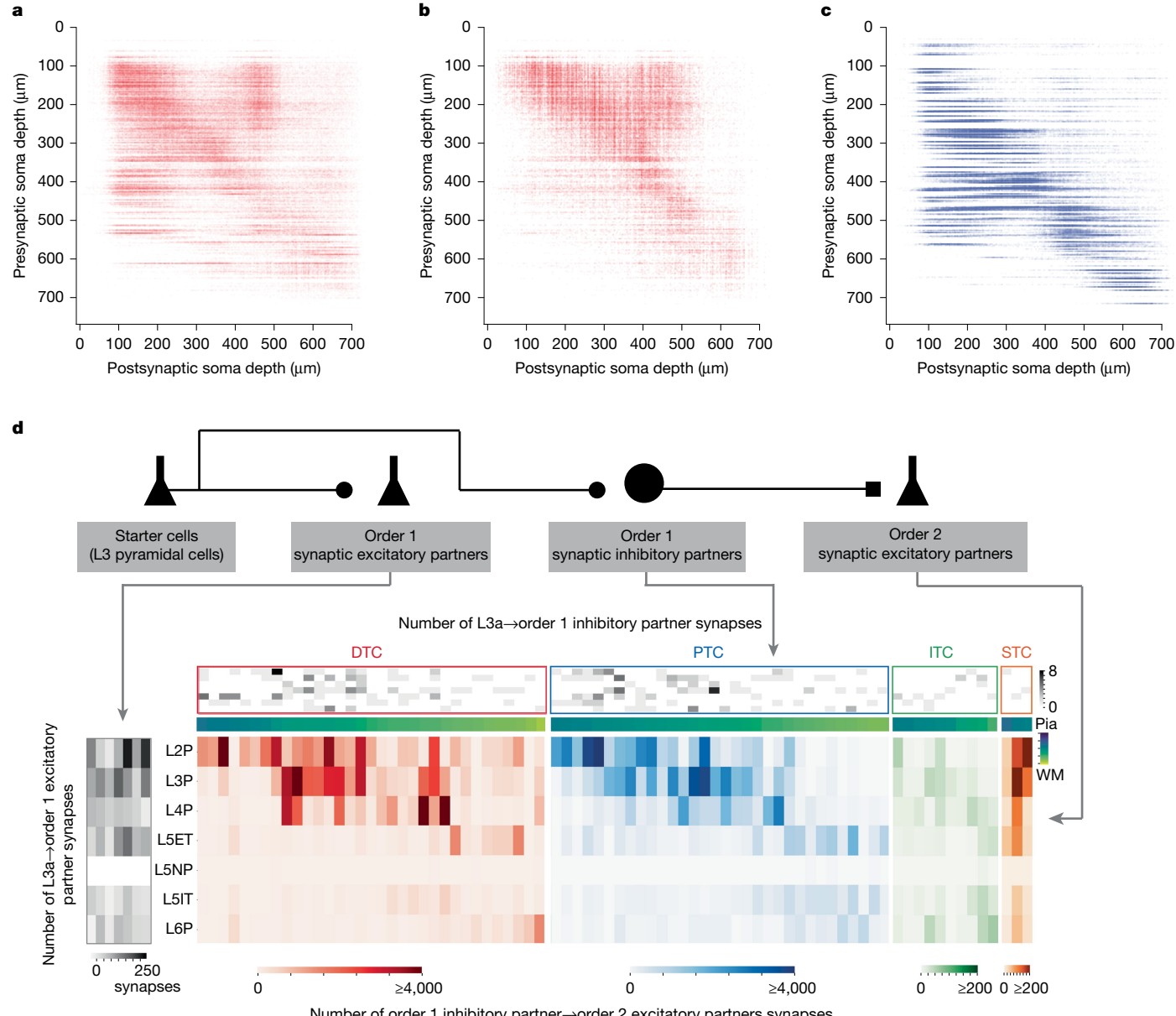

**Fig. 7 | Connectivity matrices and analysis. a–c**, Connectivity matrix for proofread neurons connecting to all postsynaptic targets of the predicted class: excitatory→excitatory (**a**); excitatory→inhibitory (**b**); inhibitory→excitatory (**c**). Each connection between two cells is represented by a dot, with the position on the *x* axis depicting the depth of the postsynaptic soma and the position on the *y* axis depicting the depth of the presynaptic cell. Dots are transparent, with darker shades indicate more connections between laminar depths. Layer boundaries are shown as dashed grey lines. **d**, First-order and second-order synaptic output heat maps of seven layer 3 pyramidal cells similar to the one shown in Fig. 6g. Left, total number of synapses that each layer 3 pyramidal cell makes with each of their order 1 postsynaptic excitatory cell types. Greyscale

heat map (top) showing number of synapses that each L3 pyramidal cell makes with their individual order 1 postsynaptic inhibitory partners, sorted by synaptic targeting types and soma depth from the pia to white matter (WM). Coloured heat map (bottom) showing total number of synapses that each order 1 inhibitory partner makes with each of their postsynaptic order 2 excitatory partners of layer 3 pyramidal cells, colour-coded by the synaptic targeting types of order 1 inhibitory partners. Inhibitory cell subclasses are represented as follows[4]: DTC, distal targeting cells (also known as Martinotti cells); PTC, proximal targeting cells (also known as basket cells); ITC, inhibitory targeting cells; STC, sparse targeting cells (mostly neurogliaform). L3a, layer 3a.

being asked, it is worth considering whether these detached spines may create bias in the conclusions drawn, such as distinguishing between excitatory and inhibitory inputs[5].

In the functional data, it is important to recognize that photon scattering and out-of-plane fluorescence may cause signal degradation and contamination with increasing depth from the pia surface, especially given the dense GCaMP6s expression in excitatory somas and neurites[39]. Caution should be taken to disentangle true biological variation in neuronal tuning across layers from these optical artefacts, by either matching controls at the same depth, or validating the finding with a

method that is less prone to these artefacts (such as electrophysiology or 2P microscopy with more sparse or targeted labelling). Furthermore, although all functional imaging was done in the same volume, it was done across several distinct imaging sessions. Technical factors as well as changes in the physiological state of the mouse should be taken into account when analysing functional recordings that were taken at different times. The simultaneous recordings of treadmill activity and pupillometry can be used to help account for variability due to state.

Developing and executing this pipeline took a large team effort, and so it is worth reflecting on the practical limitations and bottlenecks in

generating datasets of this scale. Proofreading and analysis remains the largest overall expense in terms of person hours, although it can be distributed across diverse scientific interests. Improvements in data quality, such as folds, membrane clarity and errors in computational image alignment are the most pressing technical issues that appear to limit the quality of the automated segmentation. The present dataset has already collected more than a million manual corrections to the automated segmentation, which are available for querying via CAVE[1]. We hope that these edits can be leveraged in the future to make more accurate automated segmentation, or a more extensively automated edit approach that can further increase the efficiency of proofreading. Analysis questions are often diverse in nature, so it is difficult to predict all the computational steps that are required, but having a more general framework and scalable technique of identifying specific features (such as cell types, spines and organelles) within the dataset would help increase efficiency, rather than using the specialized pipelines we used here. Some research in this direction has been applied to this dataset[40]. In terms of marginal costs of data generation, human labour remains the largest, followed by computational costs of automated segmentation and then the material costs of grid tape. Beyond these, there are no fundamental technical limitations to producing more data at this scale for other individual animals, species or brain regions.

## Comparison with other EM studies

The importance of high-resolution structural data was recognized early in invertebrate systems, particularly in the worm[41,42]. However, it is in the fly that connectomics as the pursuit of complete connectivity diagrams has had the strongest renaissance. EM volumes now describe the *Drosophila* nervous system at both larval[43] and adult[34,44] life stages and in both central brain[34,44] and nerve cord[45]. The size of the volume required to capture most central neurons and their synaptic connections in the fly is well-suited to EM. The whole fly brain fills about one-third of a $750 \times 350 \times 250$ μm$^3$ bounding box, and the nerve cord fills about one-quarter of a $950 \times 320 \times 200$ μm$^3$ bounding box[45], well within the bounds of contemporary EM methods. The creation of these datasets has spurred investment in both manual skeletonized reconstruction and automated dense reconstructions[33,44,46], with both centralized and community-minded efforts to proofread and mine them for biological insight[26,44,47–49]. In addition to the many targeted reconstructions in these datasets, large-scale proofread reconstructions from these datasets now include a manually traced full larval brain, a densely segmented and extensively proofread partial central brain and a densely segmented and proofread complete adult brain. These datasets collectively span nearly the entire fly nervous system and are driving a revolution in how fly systems neuroscience is being studied.

In the mammalian system, there is currently no EM dataset that contains a complete area, let alone a complete brain. There is however, as mentioned above, an established culture of making data open and publicly available[24,26,50–52]. In the past 10 years, there have been only three other rodent EM datasets with publicly available reconstructions that are at least 5% the size of the MICrONS multi-area dataset presented in this Article. One dataset is a $424 \times 429 \times 274$ μm$^3$ volume from P26 rat entorhinal cortex[53], with skeleton reconstructions of incomplete dendrites of 667 neurons, and skeleton reconstructions of local axons of 22 excitatory neurons averaging 550 μm in length. A dataset from mouse lateral geniculate nucleus that is $500 \times 400 \times 280$ μm$^3$ in size and contains around 3,000 neuronal cell bodies is publicly available[54]. This dataset is large enough that dendritic reconstructions from the centre of the volume are nearly complete, and it has a sparse manual segmentation, covering around 1% of the volume, which includes 304 thalamocortical cells and 162 axon fragments. The third dataset is a $424 \times 453 \times 360$ μm volume covering layer 4 of mouse primary somatosensory cortex, with manual reconstruction of 52 interneuronal dendrites and many axons[55].

It is critically important to compare circuit architectures across regions and species. The neocortex is of particular interest as it is expanded in human compared to mouse. There is already a large body of literature on the comparative aspects between the cortex of humans and of other species. This research includes morphological and electrical properties of neurons, density of spines, synapses and neurons, as well as biophysical properties and morphology of synaptic connections[56–58]. Of note, a recent EM connectomics dataset of the human medial temporal gyrus[59] vastly expands the possibilities of this comparison. This is a cubic millimetre scale volume, with a maximum extent of $3 \times 2$ mm and a thickness of 150 μm. This human dataset is publicly available, including a dense automated reconstruction of all objects, with around 16,000 neurons, 130 million synapses and an initial release of 104 proofread cells. These human connectomics data will doubtless yield critical insights. One practical difference from the volume described here is the aspect ratio of the human data, which is matched to the greater thickness of human cortex compared to mouse. To some extent, the wide and thin dimensions of the human dataset trades off completeness of local neurons and circuits in order to sample all layers, whereas the nearly cubic volume described here is more suitable for studying local circuits and long-range connections across areas. With the exception of the study by Hua et al.[55], the other studies mentioned above do not have corresponding functional characterizations of the neurons reconstructed in EM. By contrast, the functional connectomics data we have released includes both anatomy and activity of the same cells.

## Opportunity to map cell types at scale

In the mammalian nervous system, transcriptomics has been the most scalable approach for cell-type taxonomies. In smaller organisms such as the fly, for which we have both extensive gene expression maps, whole-brain neuronal reconstructions and nearly complete connectomes, integration across modalities has been a powerful engine of discovery. Moreover, the availability of connectomes in the fly have enabled a much higher resolution of cell types, with novel taxonomies and new cell types being discovered[44]. The accompanying studies[4–7] suggest that a similar path to cell-type discovery will be enabled by large-scale EM in the mammalian system with novel cell types and novel patterns of connectivity.

This wealth of structural data on cell types and circuits provides strong constraints on the nature of the computations that the brain performs, whereas genes provide constraints on how this structure is built and operates. Linking connectomics to transcriptomics is a first step for merging connectivity with molecular information and building cell-type-specific tools that are informed by how neurons connect. In one of the accompanying studies[7], we offer a proof of concept on how to achieve this link for Martinotti cells, using morphology as a common feature to integrate PatchSeq and EM datasets, suggesting a broader pathway for multimodal integration.

In this respect, our work parallels another milestone of connectomics, the completion of the *Drosophila* connectome[34,35,44,46]; only 20% of the neuron types described in the EM connectome of the central brain were previously described in the literature[44]. There is however an important difference to be drawn with *Drosophila*, in which a cell type often consists of just a few neurons that share similar functional properties that are reproducible across individuals. Owing to this stereotypy, a connectome mapped in one fly can usually be used by researchers studying neuronal function in other flies. Rules of connectivity based on cell types have proved sufficient for understanding and modelling many functions of increasingly complex neural circuits[60,61]. Conversely, a single cell type in a mammalian brain encompasses a huge number of cells, which generally exhibit different tuning preferences. This is why it is important to combine cortical connectomics with functional studies of the same neurons in the same brain. This is also why the mapping of cortical connectivity must go beyond rules that depend solely on cell types.

## Importance of functional connectomics

Almost 50 years after Crick described his "impossible" experiment, we have provided a first draft, but its full promise will take some time to achieve. Most importantly, complete segmentation still requires an extensive amount of proofreading for the largest datasets, such as the millimetre scale cortical reconstruction reported here. Similarly, simultaneously recording single action potentials from tens of thousands of neurons is constrained by sensor dynamics and optical sampling constraints.

Nonetheless, there has been steady progress. The first structure–function studies that combined 2P microscopy and EM examined how the wiring of mouse retina[27–31] and mouse visual cortex[24] related to functional properties. Lee et al.[25] related visual tuning properties of 50 functionally characterized neurons in primary visual cortex to their connectivity measured via EM reconstruction of a 450 × 450 × 150 μm volume. One thousand synapses were mapped by hand, yielding a graph of connectivity between 29 orientation-tuned cells (a subset of the characterized cells, as in the current dataset). Subsequently, our consortium used dense segmentation plus proofreading of a 250 × 140 × 90 μm dataset[26] from mouse layer 2/3 visual cortex, yielding many more overall connections, but still only twice the number of functionally characterized cells. Perhaps most impressively, In the olfactory bulb of the zebrafish, Wanner et al.[62] manually reconstructed almost all neurons ($n = 1,003$) within a 72 × 108 × 119 μm³ volume, in which responses to odours were measured in vivo. Their analysis of the 18,483 measured connections revealed how this structural network mediated de-correlation and variance normalization of the functional responses and demonstrates how larger measurements of network structure and function can provide mechanistic insights.

By contrast, the data released here contains tens of thousands of neurons with functionally characterized responses to visual stimuli and, because it is densely segmented and contains complete dendritic and local axonal arbors of centrally located cells, the opportunities to study connected neurons are orders of magnitude greater. As an example, from just 94 proofread excitatory axons, one can query 69,962 output synapses, which map to 20,112 distinct neuron soma in the volume.

Moreover, inspired by recent advancements in artificial intelligence, we also created a functional digital twin of the MICrONS mouse that can enable a more comprehensive analysis of function[10,11]. Specifically, we developed a 'foundation model'[11] for the mouse visual cortex using deep learning that was trained using large-scale datasets from multiple visual cortical areas and mice, recorded while they viewed ecological videos. The model demonstrated its generalization abilities by accurately predicting neuronal responses, not only to natural videos, but also to various new stimulus domains, such as coherent moving dots and noise patterns, as confirmed through in vivo testing[10,11]. By applying the foundation model to the MICrONS mouse data, we created a functional digital twin of this mouse, paving the way for a systematic exploration of the relationship between circuit structure and function for tens of thousands of neurons connected with millions of synapses. Combined with the anatomical data from this mouse, we can investigate the structure–function relationships for specific visual computations[8,9] and decipher the principles that determine the synaptic network in the cortex[10,11].

The most important goal of connectomics is to map the connections between cells, from cell body to axon to synapse, and back to cell body. In a large volume with complete and segmented dendrites and local axons, this can be achieved. Currently, the dendrites are nearly completely segmented (Fig. 6), but many axons require proofreading. A goal in future years will be to complete the segmentation through a combination of additional machine learning and improved proofreading. This echoes the successful strategy in the reconstruction of the fly adult brain, which started with the TEM volume[34], then added the tools developed by the MICrONS programme for segmentation and proofreading and led to the complete connectome[35]. If, in addition, most cell bodies have physiology with single-spike resolution, then Crick's experimental challenge will be met. These remaining hurdles may take some time to clear, but the next steps are becoming apparent.

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

**The MICrONS Consortium**

**J. Alexander Bae**[1,2]**, Mahaly Baptiste**[3,4]**, Maya R. Baptiste**[3,4]**, Caitlyn A. Bishop**[5]**, Agnes L. Bodor**[6]**, Derrick Brittain**[6]**, Victoria Brooks**[3,4]**, JoAnn Buchanan**[6]**, Daniel J. Bumbarger**[6]**, Manuel A. Castro**[1]**, Brendan Celii**[3,4,7]**, Erick Cobos**[3,4]**, Forrest Collman**[6✉]**, Nuno Maçarico da Costa**[6✉]**, Bethanny Danskin**[6]**, Sven Dorkenwald**[1,8]**, Leila Elabbady**[6]**, Paul G. Fahey**[3,4]**, Tim Fliss**[6]**, Emmanouil Froudarakis**[3,4,9]**, Jay Gager**[1]**, Clare Gamlin**[6]**, William Gray-Roncal**[5]**, Akhilesh Halageri**[1]**, James Hebditch**[1]**, Zhen Jia**[1,8]**, Emily Joyce**[6]**, Justin Ellis-Joyce**[5]**, Chris Jordan**[1]**, Daniel Kapner**[6]**, Nico Kemnitz**[1]**, Sam Kinn**[6]**, Lindsey M. Kitchell**[5]**, Selden Koolman**[1]**, Kai Kuehner**[1]**, Kisuk Lee**[1,10]**, Kai Li**[8]**, Ran Lu**[1]**, Thomas Macrina**[1,8]**, Gayathri Mahalingam**[6]**, Jordan Matelsky**[5]**, Sarah McReynolds**[3,4]**, Elanine Miranda**[3,4]**, Eric Mitchell**[1]**, Shanka Subhra Mondal**[1,2]**, Merlin Moore**[1]**, Shang Mu**[1]**, Taliah Muhammad**[3,4]**, Barak Nehoran**[1,8]**, Erika Neace**[6]**, Oluwaseun Ogedengbe**[1]**, Christos Papadopoulos**[3,4]**, Stelios Papadopoulos**[3,4]**, Saumil Patel**[3,4]**, Guadalupe Jovita Yasmin Perez Vega**[3,4]**, Xaq Pitkow**[3,4,7,11,12,13,14✉]**, Sergiy Popovych**[1,8]**, Anthony Ramos**[3,4]**, R. Clay Reid**[6✉]**, Jacob Reimer**[3,4✉]**, Patricia K. Rivlin**[5]**, Victoria Rose**[5]**, Zachary M. Sauter**[3,4]**, Casey M. Schneider-Mizell**[6]**, H. Sebastian Seung**[1,8✉]**, Ben Silverman**[1]**, William Silversmith**[1]**, Amy Sterling**[1]**, Fabian H. Sinz**[3,4,15,16]**, Cameron L. Smith**[3,4]**, Rachael Swanstrom**[6]**, Shelby Suckow**[6]**, Marc Takeno**[6]**, Zheng H. Tan**[3,4]**, Andreas S. Tolias**[3,4,7,14,17✉]**, Russel Torres**[6]**, Nicholas L. Turner**[1,8]**, Edgar Y. Walker**[3,4,15,18]**, Tianyu Wang**[19]**, Adrian Wanner**[1,20]**, Brock A. Wester**[5]**, Grace Williams**[6]**, Sarah Williams**[1]**, Kyle Willie**[1]**, Ryan Willie**[1]**, William Wong**[1]**, Jingpeng Wu**[1]**, Chris Xu**[19]**, Runzhe Yang**[1,8]**, Dimitri Yatsenko**[3,4]**, Fei Ye**[3,4]**, Wenjing Yin**[6]**, Rob Young**[6]**, Szi-chieh Yu**[1]**, Daniel Xenes**[5]** & Chi Zhang**[6]

## Methods

### Mouse lines

All procedures were approved by the Institutional Animal Care and Use Committee (IACUC) of Baylor College of Medicine. All results described here are from a single male mouse, age 65 days at onset of experiments, expressing GCaMP6s in excitatory neurons via *Slc17a7*-Cre[65] and Ai162[66] heterozygous transgenic lines (recommended and generously shared by H. Zeng at Allen Institute for Brain Science; JAX stock 023527 and 031562, respectively). In order to select this animal, 31 (12 female, 19 male) GCaMP6-expressing animals underwent surgery as described below. Of these, eight animals were chosen based on a variety of criteria including surgical success and animal recovery, the accessibility of lateral higher visual areas in the cranial window, the degree of vascular occlusion, and the success of cortical tissue block extraction and staining. Of these 8 animals, one was chosen for 40-nm slicing and EM imaging based on overall quality using these criteria.

### Timeline

Mouse birth date: 19 December 2017
Surgery: 21 February 2018 (P64)
2P imaging start: 4 March 2018 (P75)
2P imaging end: 9 March 2018 (P80)
Structural Stack: 21 March 2018 (P83)
Perfusion: 16 March 2018 (P87)

### Surgery

Anaesthesia was induced with 3% isoflurane and maintained with 1.5–2% isoflurane during the surgical procedure. Mice were injected with 5–10 mg kg$^{-1}$ ketoprofen subcutaneously at the start of the surgery. Anaesthetized mice were placed in a stereotaxic head holder (Kopf Instruments) and their body temperature was maintained at 37 °C throughout the surgery using a homeothermic blanket system (Harvard Instruments). After shaving the scalp, bupivicane (0.05 ml, 0.5%, Marcaine) was applied subcutaneously, and after 10–20 min an approximately 1 cm$^2$ area of skin was removed above the skull and the underlying fascia was scraped and removed. The wound margins were sealed with a thin layer of surgical glue (VetBond, 3 M), and a 13-mm stainless steel washer clamped in the headbar was attached with dental cement (Dentsply Grip Cement). At this point, the mouse was removed from the stereotax and the skull was held stationary on a small platform by means of the newly attached headbar. Using a surgical drill and HP 1/2 burr, a 4-mm-diameter circular craniotomy was made centred on the border between primary visual cortex and lateromedial visual cortex (V1, lateral–medial; 3.5 mm lateral of the midline, -1 mm anterior to the lambda suture), followed by a durotomy. The exposed cortex was washed with artificial cerebrospinal fluid (25 mM NaCl, 5 mM KCl, 10 mM glucose, 10 mM HEPES, 2 mM CaCl$_2$, 2 mM MgSO$_4$) with 0.3 mg ml$^{-1}$ gentamicin sulfate (Aspen Veterinary Resources). The cortical window was then sealed with a 4-mm coverslip (Warner Instruments), using cyanoacrylate glue (VetBond). The mouse was allowed to recover for 1 day prior to imaging. After imaging, the washer was released from the headbar and the mouse was returned to the home cage. Prior to surgery and throughout the imaging period, mice were singly housed and maintained on a reverse 12-h light cycle (off at 11:00, on at 23:00).

### 2P imaging

Mice were head-mounted above a cylindrical treadmill and calcium imaging was performed using Chameleon Ti-Sapphire laser (Coherent) tuned to 920 nm and a large FOV mesoscope[67] equipped with a custom objective (excitation NA 0.6, collection NA 1.0, 21 mm focal length). Laser power after the objective was increased exponentially as a function of depth from the surface according to:

$$P = P_0 \times e^{(z/L_z)}$$

Here $P$ is the laser power used at target depth $z$, $P_0$ is the power used at the surface (not exceeding 10 mW), and $L_z$ is the depth constant (not less than 150 μm). Maximum laser output of 115 mW was used for scans approximately 450–500 μm from the surface and below.

### Monitor positioning

Visual stimuli were presented to the left eye with a 31.8 × 56.5 cm (height × width) monitor (ASUS PB258Q) with a resolution of 1,080 × 1,920 pixels positioned 15 cm away from the eye. When the monitor is centred on and perpendicular to the surface of the eye at the closest point, this corresponds to a visual angle of ~3.8° cm$^{-1}$ at the nearest point and 0.7° cm$^{-1}$ at the most remote corner of the monitor. As the craniotomy coverslip placement during surgery and the resulting mouse positioning relative to the objective is optimized for imaging quality and stability, uncontrolled variance in animal skull position relative to the washer used for head-mounting was compensated with tailored monitor positioning on a six-dimensional monitor arm. The pitch of the monitor was kept in the vertical position for all animals, while the roll was visually matched to the roll of the animal's head beneath the headbar by the experimenter. In order to optimize the translational monitor position for centred visual cortex stimulation with respect to the imaging FOV, we used a dot stimulus with a bright background (maximum pixel intensity) and a single dark square dot (minimum pixel intensity). Dot locations were randomly ordered from a 5 × 8 grid to tile the screen, with 15 repetitions of 200 ms presentation at each location. The final monitor position for each animal was chosen in order to centre the population receptive field of the scan field ROI on the monitor, with the yaw of the monitor visually matched to be perpendicular to and 15 cm from the nearest surface of the eye at that position. An L-bracket on a six-dimensional arm was fitted to the corner of the monitor at this location and locked in position, so that the monitor could be returned to the chosen position between scans and across days.

### Imaging site selection

The craniotomy window was leveled with regards to the objective with six degrees of freedom, five of which were locked between days to allow us to return to the same imaging site using the z axis. Pixel-wise responses from a 3,000 × 3,000 μm ROI spanning the cortical window (150 μm from surface, five 600 × 3,000 μm fields, 0.2 pixels per μm) to drifting bar stimuli were used to generate a sign map for delineating visual areas[68]. Our target imaging site was a 1,200 × 1,100 × 500 μm volume (anteroposterior × mediolateral × radial depth) spanning layer 2 to layer 6 at the conjunction of VISp and three higher visual areas: VISlm, VISrl and VISal[69]. This resulted in an imaging volume that was roughly 50% VISp and 50% higher visual area (HVA). This target was chosen to maximize the number of visual areas within the reconstructed cortical volume, as well as maximizing the overlap in represented visual space. The imaging site was further optimized to minimize vascular occlusion and to minimize motion artefact, especially where the brain curves away from the skull/coverslip towards the lateral aspect.

Once the imaging volume was chosen, a second retinotopic mapping scan with the same stimulus was collected at 12.6 Hz and matching the imaging volume FOV with four 600 × 1,100 μm fields per frame at 0.4 pixels per μm *xy* resolution to tile a 1,200 × 1,100 μm FOV at 2 depths (2 planes per depth, with no overlap between coplanar fields). Area boundaries on the sign map were manually annotated.

### 2P functional imaging

Of 19 completed scans over 6 days of imaging, 14 are described here (Extended Data Table 1). Scan placement targeted 10–15 μm increments in depth to maximize coverage of the volume in depth.

For 11 scans, imaging was performed at 6.3 Hz, collecting eight $620 \times 1,100$ μm fields per frame at 0.4 pixel per μm $xy$ resolution to tile a $1,200 \times 1,100$ μm (width × height) FOV at four depths (two planes per depth, 40 μm overlap between coplanar fields).

For 2 scans, imaging was performed at 8.6 Hz, collecting six $620 \times 1,100$ μm fields per frame at 0.4 pixels per μm $xy$ resolution to tile a $1,200 \times 1,100$ μm (width × height) FOV at 3 depths (2 planes per depth, 40 μm overlap between coplanar fields).

For 1 scan, imaging was performed at 9.6 Hz, collecting four $620 \times 1,000$ μm fields per frame at 0.6 pixels per μm $xy$ resolution to tile a $1,200 \times 1,000$ μm (width × height) FOV at 2 depths (2 planes per depth, 40 μm overlap between coplanar fields).

The higher-resolution scans were designed to enable future analysis efforts to extract signals from large apical dendrites for example using EM-Assisted Source Extraction (EASE[70]). In addition to locking the craniotomy window mount between days, the target imaging site was manually matched each day to preceding scans within several micrometres using structural features including horizontal blood vessels (which have a distinctive $z$-profile) and patterns of somata (identifiable by GCaMP6s exclusion as dark spots).

The full 2P imaging processing pipeline is available at (https://github.com/cajal/pipeline). Raster correction for bidirectional scanning phase row misalignment was performed by iterative greedy search at increasing resolution for the raster phase resulting in the maximum cross-correlation between odd and even rows. Motion correction for global tissue movement was performed by shifting each frame in $x$ and $y$ to maximize the correlation between the cross-power spectra of a single scan frame and a template image, generated from the Gaussian-smoothed average of the Anscombe transform from the middle 2,000 frames of the scan. Neurons were automatically segmented using constrained non-negative matrix factorization, then deconvolved to extract estimates of spiking activity, within the CaImAn pipeline[71]. Cells were further selected by a classifier trained to separate somata versus artefacts based on segmented cell masks, resulting in exclusion of 8.1% of masks. The functional data is available in a DataJoint[72] database and can also be read as NWB files deposited in the DANDI data archive[73].

## 2P structural stack

Approximately 55 min prior to collecting the stack, the mouse was injected subcutaneously with 60 μl of 8.3 mM Dextran Texas Red fluorescent dye (Invitrogen, D3329). The stack was composed of 30 repeats of three $620 \times 1,300$ μm (width × height) fields per depth in 2 channels (green and red, respectively), tiling a $1,400 \times 1,300$ μm FOV (460 μm total overlap in width) at 335 depths from 21 μm above the surface to 649 μm below the surface. The green channel average image across repetitions for each field was enhanced with local contrast normalization using a Gaussian filter to calculate the local pixel means and standard deviations. The resulting image was then Gaussian smoothed and sharpened using a Laplacian filter. Enhanced and sharpened fields were independently stitched at each depth. The resulting stitched planes were independently horizontally and vertically aligned by maximizing the correlation of the cross-power spectrum of their Fourier transformations. Finally, the resulting alignment was detrended in $z$ using a Hann filter with a size of 60 μm to remove the influence of vessels passing through the fields. The resulting transform was applied to the original average images resulting in a structural 2P $1,412 \times 1,322 \times 670$ μm (width × height × depth) volume at $0.5 \times 0.5 \times 0.5$ pixels per μm resolution in both red and green channels.

Owing to tissue deformation from day to day across such a wide FOV, some cells are recorded in more than one scan. To assure we count cells only once, we subsample our recorded cells based on proximity in 3D space. Functional scan fields were independently registered using an affine transformation matrix with 9 parameters estimated via gradient ascent on the correlation between the sharpened average scanning

plane and the extracted plane from the sharpened stack. Using the 3D centroids of all segmented cells, we iteratively group the closest 2 cells from different scans until all pairs of cells are at least 10 μm apart or a further join produces an unrealistically tall mask (20 μm in $z$). Sequential registration of sections of each functional scan into the structural stack was performed to assess the level of drift in the $z$ dimension. All scans had less than 10-μm drift over the 1.5-h recording, and for most of them drift was limited to <5 μm.

Fields from the FOV-matched retinotopy scan described above were registered into the stack using the same approach, and the manually annotated area masks were transformed into the stack. These area masks were extended vertically across all depths, and functional units inherit their area membership from their stack $xy$ coordinates.

## Eye and face camera

Video images of the eye and face of the mouse were captured throughout the experiment. A hot mirror (Thorlabs FM02) positioned between the animal's left eye and the stimulus monitor was used to reflect an IR image onto a camera (Genie Nano C1920M, Teledyne Dalsa) without obscuring the visual stimulus. An infrared 940 nm LED (Thorlabs M940L2) illuminated the right side of the animal, backlighting the silhouette of the face. The position of the mirror and camera were manually calibrated per session and focused on the pupil. FOV was manually cropped for each session (ranging from $828 \times 1,217$ pixels to $1,080 \times 1920$ pixels at ~20 Hz), such that the FOV contained the superior, frontal, and inferior portions of the facial silhouette as well as the left eye in its entirety. Frame times were time stamped in the behavioural clock for alignment to the stimulus and scan frame times. Video was compressed using Labview's MJPEG codec with quality constant of 600 and stored the frames in AVI file.

Light diffusing from the laser during scanning through the pupil was used to capture pupil diameter and eye movements. Notably, scans using wide ranges in laser power to scan both superficial and deep planes resulted in a variable pupil intensity between frames. A custom semi-automated user interface in Python was built for dynamic adaptation of fitting parameters throughout the scan to maximize pupil tracking accuracy and coverage. The video was manually cropped to a rectangular region that includes the entirety of the eye at all time points. The video was further manually masked to exclude high intensity regions in the surrounding eyelids and fur. In cases where a whisker is present and occluding the pupil at some time points, a merge mask was drawn to bridge ROIs drawn on both sides of the whisker into a single ROI. For each frame, the original and filtered image was visible to the user. The filtered image was an exponentially weighted temporal running average, which undergoes exponentiation, Gaussian blur, automatic Otsu thresholding into a binary image, and finally pixel-wise erosion/dilation. In cases where only one ROI was present, the contour of the binary ROI was fit with an ellipse by minimizing least squares error, and for ellipses greater than the minimum contour length the $xy$ centre and major and minor radii were stored. In cases where more than one ROI was present, the tracking was automatically halted until the user either resolved the ambiguity, or the frame was not tracked (a NaN (Not a Number) is stored). Processing parameters were under dynamic control of the user, with instructions to use the minimally sufficient parameters that result in reliably and continuous tracing of the pupil, as evidenced by plotting of the fitted ROI over the original image. Users could also return to previous points in the trace for re-tracking with modified processing parameters, as well as manually exclude periods of the trace in which insufficient reliable pupil boundary was visible for tracking.

## Treadmill

The mouse was head-restrained during imaging but could walk on a treadmill. Rostro-caudal treadmill movement was measured using a rotary optical encoder (Accu-Coder 15T-01SF-2000NV1ROC-F03-S1)

with a resolution of 8,000 pulses per revolution, and was recorded at ~57–100 Hz in order to extract locomotion velocity.

## Stimulus composition

The stimulus was designed to cover a sufficiently large feature space to support training highly accurate models that predict neural responses to arbitrary visual stimuli[11,38,74,75]. Each scan stimulus was approximately 84 min in duration and comprised:

- Oracle natural videos: 6 natural video clips, 2 from each category. 10 s each, 10 repeats per scan, 10 min total. Conserved across all scans.
- Unique natural videos: 144 natural videos, 48 from each category. 10 s each, 1 repeat per scan, 24 min total. Unique to each scan.
- 2× repeat natural videos: 90 natural videos, 30 from each category. 10 s each, 2 repeats (one in each half of the scan), 30 min total. Conserved across all scans.
- Local directional parametric stimulus (Trippy): 20 seeds, 15 s each, 2 repeats (one in each half of the scan), 10 min total. 10 seeds conserved across all scans, 10 unique to each scan.
- Global directional parametric stimulus (Monet2): 20 seeds, 15 s each, 2 repeats (one in each half of the scan), 10 min total. 10 seeds conserved across all scans, 10 unique to each scan.

Each scan was also preceded by 0.15–5.5 min with the monitor on, and followed by 8.3–21.2 min with the monitor off, in order to collect spontaneous neural activity.

## Natural visual stimulus

The visual stimulus was composed of dynamic stimuli, primarily including natural video but also including generated parametric stimuli with strong local or global directional component. Natural video clips were 10 s clips from one of three categories:

- Cinematic, from the following sources: *Mad Max: Fury Road* (2015), *Star Wars: Episode VII—The Force Awakens* (2015), *The Matrix* (1999), *The Matrix Reloaded* (2003), *The Matrix Revolutions* (2003), *Koyaanisqatsi: Life Out of Balance* (1982), *Powaqqatsi: Life in Transformation* (1988) and *Naqoyqatsi: Life as War* (2002).
- Sports-1M collection[37], with the following keywords: cycling, mountain unicycling, bicycle, BMX, cyclo-cross, cross-country cycling, road bicycle racing, downhill mountain biking, freeride, dirt jumping, slopestyle, skiing, skijoring, Alpine skiing, freestyle skiing, Greco-Roman wrestling, luge, canyoning, adventure racing, street-luge, riverboarding, snowboarding, mountainboarding, aggressive inline skating, carting, freestyle motocross, f1 powerboat racing, basketball and base jumping.
- Rendered 3D video of first-person POV random exploration of a virtual environment with moving objects, produced in a customized version of Unreal Engine 4 with modifications that enable precise control and logging of frame timing and camera positions to ensure repeatability across multiple rendering runs. Environments and assets were purchased from Unreal Engine Marketplace. Assets chosen for diversity of appearance were translated along a piecewise linear trajectory, and rotated with a piecewise constant angular velocity. Intervals between change points were drawn from a uniform distribution from 1 to 5 s. If a moving object encountered an environmental object, it bounced off and continued along a linear trajectory reflected across the surface normal. The first-person POV camera followed the same trajectory process as the moving objects. Light sources were the default for the environment. Latent variable images were generated by re-generating the scenes and trajectories, rendering different properties, including absolute depth, object identification number and surface normals.

All natural videos were temporally resampled to 30 frames per second, and were converted to greyscale with 256 × 144 pixel resolution with FFmpeg (ibx264 at YUV4:2:0 8 bit). Stimuli were automatically filtered for upper 50th percentile Lucas–Kanade optical flow and temporal contrast of the central region of each clip. All natural videos included in these experiments were further manually screened for unsuitable characteristics (for example, fragments of rendered videos in which the first-person perspective would enter a corner and become 'trapped' or follow an unnatural camera trajectory, or fragments of cinematic or Sports-1M containing screen text or other post-processing editing).

## Global directional parametric stimulus

To probe neuronal tuning to orientation and direction of motion, a visual stimulus (Monet2) was designed in the form of smoothened Gaussian noise with coherent orientation and motion. In brief, an independently identically distributed (i.i.d.) Gaussian noise video was passed through a temporal low-pass Hamming filter (4 Hz) and a 2D Gaussian spatial filter ($\sigma = 3.0°$ at the nearest point on the monitor to the mouse). Each 15-s block consisted of 16 equal periods of motion along one of 16 unique directions of motion between 0–360° with a velocity of 42.8° s$^{-1}$ at the nearest point on the monitor. The video was spatial filtered to introduce a spatial orientation bias perpendicular to the direction of movement by applying a bandpass Hanning filter $G(\omega; c)$ in the polar coordinates in the frequency domain for $\omega = \phi - \theta$ where $\phi$ is the polar angle coordinate and $\theta$ is the movement direction $\theta$. Then:

$$G(\omega; c) = \sqrt{c}\, H(c\omega)$$

and

$$H(\omega) = \frac{1}{2} + \frac{1}{2}\cos \omega \text{ if } |\omega| < \pi \text{ and } 0 \text{ otherwise}$$

Here, $c = 2.5$ is an orientation selectivity coefficient. At this value, the resulting orientation kernel's size is 72° full width at half maximum in spatial coordinates.

## Local directional parametric stimulus Trippy

To probe the tuning of neurons to local spatial features including orientation, direction, spatial and temporal frequency, the Trippy stimulus was synthesized by applying the cosine function to a smoothened noise video. In brief, a phase movie was generated as an i.i.d. uniform noise video with 4 Hz temporal bandwidth. The video was up-sampled to 60 Hz with the Hanning temporal kernel. An increasing trend of $8\pi$ s$^{-1}$ was added to the video to produce drifting grating movements whereas the noise component added local variations of the spatial features. The video was spatially up-sampled to the full screen with a 2D Gaussian kernel with a sigma of 5.97 cm or 22.5° at the nearest point. The resulting stimulus yielded the local phase video of the gratings, from which all visual features are derived analytically.

## Stimulus alignment

A photodiode (TAOS TSL253) was sealed to the top left corner of the monitor, where stimulus sequence information was encoded in a three-level signal according to the binary encoding of the flip number assigned in order. This signal was recorded at 10 MHz on the behaviour clock (MasterClock PCIe-OSC-HSO-2 card). The signal underwent a sine convolution, allowing for local peak detection to recover the binary signal. The encoded binary signal was reconstructed for 89% of trials. A linear fit was applied to the trial timestamps in the behavioural and stimulus clocks, and the offset of that fit was applied to the data to align the two clocks, allowing linear interpolation between them.

## Oracle score

We used six natural video conditions that were present in all scans and repeated ten times per scan to calculate an oracle score representing the reliability of the trace response to repeated visual stimuli.

This score was computed as the jackknife mean of correlations between the leave-one-out mean across repeated stimuli with the remaining trial.

## Tissue preparation

After optical imaging at Baylor College of Medicine, candidate mice were shipped via overnight air freight to the Allen Institute. All procedures were carried out in accordance with the Institutional Animal Care and Use Committee at the Allen Institute for Brain Science. All mice were housed in individually ventilated cages, 20–26 °C, 30–70% relative humidity, with a 12-h light:dark cycle. Mice were transcardially perfused with a fixative mixture of 2.5% paraformaldehyde, 1.25% glutaraldehyde, and 2 mM calcium chloride, in 0.08 M sodium cacodylate buffer, pH 7.4. After dissection, the neurophysiological recording site was identified by mapping the brain surface vasculature. A thick (1,200-µm) slice was cut with a vibratome and post-fixed in perfusate solution for 12–48 h. Slices were extensively washed and prepared for reduced osmium treatment based on the protocol of Hua et al.[76]. All steps were performed at room temperature, unless indicated otherwise. Osmium tetroxide (2%, 78 mM) with 8% v/v formamide (1.77 M) in 0.1 M sodium cacodylate buffer, pH 7.4, for 180 min, was the first osmication step. Potassium ferricyanide 2.5% (76 mM) in 0.1 M sodium cacodylate, 90 min, was then used to reduce the osmium. The second osmium step was at a concentration of 2% in 0.1 M sodium cacodylate, for 150 min. Samples were washed with water, then immersed in thiocarbohydrazide (TCH) for further intensification of the staining (1% TCH (94 mM) in water, 40 °C, for 50 min). After washing with water, samples were immersed in a third osmium immersion of 2% in water for 90 min. After extensive washing in water, lead aspartate (Walton's (20 mM lead nitrate in 30 mM aspartate buffer, pH 5.5), 50 °C, 120 min) was used to enhance contrast. After two rounds of water wash steps, samples proceeded through a graded ethanol dehydration series (50%, 70%, 90% w/v in water, 30 min each at 4 °C, then 3× 100%, 30 min each at room temperature). Two rounds of 100% acetonitrile (30 min each) served as a transitional solvent step before proceeding to epoxy resin (EMS Hard Plus). A progressive resin infiltration series (1:2 resin:acetonitrile (33% v/v), 1:1 resin:acetonitrile (50% v/v), 2:1 resin acetonitrile (66% v/v), then 2× 100% resin, each step for 24 h or more, on a gyrotary shaker) was done before final embedding in 100% resin in small coffin molds. Epoxy was cured at 60 °C for 96 h before unmolding and mounting on microtome sample stubs for trimming.

The surface of the brain in the neurophysiology ROI was highly irregular, with depressions and elevations that made it impossible to trim all the resin from the surface of the cortex without removing layer 1 (L1) and some portions of layer 2 (L2). Though empty resin increases the number of folds in resulting sections, we left some resin so as to keep the upper layers (L1 and L2) intact to preserve inter-areal connectivity and the apical tufts of pyramidal neurons. Similarly, white matter was also maintained in the block to preserve inter-areal connections despite the risk of increased sectioning artefacts that then have to be corrected through proofreading.

## Ultrathin sectioning

The sections were then collected at a nominal thickness of 40 nm using a modified ATUMtome[63] (RMC/Boeckeler) onto 6 reels of grid tape[45]. The knife was cleaned every 100–500 sections, occasionally leading to the loss of a very thin partial section (≪40 nm). Thermal expansion of the block as sectioning resumed post-cleaning resulted in a short series of sections substantially thicker than the nominal cutting thickness. The sectioning took place in two sessions, the first session took 8 consecutive days on a 24 h a day schedule and contained sections 1 to 14773. The loss rate on this initial session was low, but before section 7931 there were two events that led to consecutive section loss (due to these consecutive section losses we decided to not reconstruct the region containing sections 1 to 7931 even though the imagery was collected). The first event that led to consecutive section loss was due

to sections being collected onto apertures with damaged films. To prevent this from happening again, we installed a camera that monitors the aperture before collection. The second event was due to an accident where the knife bumped the block and nicked a region near the edge of the ROI. At the end of this session we started seeing differential compression between the resin and the surface of the cortex. Because this could lead to severe section artefacts, we paused to trim additional empty resin from the block and also replaced the knife. The second session lasted five consecutive days and an additional 13,199 sections were cut. Due to the interruption, block shape changes and knife replacement, there are approximately 45 partial sections at the start of this session; importantly, these do not represent tissue loss (see stitching and alignment section). As will be described later, the EM dataset is subdivided into two subvolumes due to sectioning and imaging events that resulted in loss of a series of sections.

## TEM imaging

The parallel imaging pipeline described here[63] converts a fleet of TEMs into high-throughput automated image systems capable of 24/7 continuous operation. It is built upon a standard JEOL 1200EXII 120 kV TEM that has been modified with customized hardware and software. The key hardware modifications include an extended column and a custom electron-sensitive scintillator. A single large-format CMOS camera outfitted with a low distortion lens is used to grab image frames at an average speed of 100 ms. The autoTEM is also equipped with a nano-positioning sample stage that offers fast, high-fidelity montaging of large tissue sections and an advanced reel-to-reel tape translation system that accurately locates each section using index barcodes for random access on the GridTape. In order for the autoTEM system to control the state of the microscope without human intervention and ensure consistent data quality, we also developed customized software infrastructure piTEAM that provides a convenient GUI-based operating system for image acquisition, TEM image database, real-time image processing and quality control, and closed-loop feedback for error-detection and system protection etc. During imaging, the reel-to-reel GridStage moves the tape and locates targeting aperture through its barcode. The 2D montage is then acquired through raster scanning the ROI area of tissue. Images along with metadata files are transferred to the data storage server. We perform image quality control on all the data and reimage sections that fail the screening. Pixel sizes for all systems were calibrated within the range between 3.95 and 4.05 nm per pixel and the montages had a typical size of 1.2 mm × 0.82 mm. The EM dataset contains raw tile images with two different sizes because two cameras with two different resolutions were used during acquisition. The most commonly used was a 20-megapixel camera that required 5,000 individual tiles to capture the 1 mm² montage of each section. During the dataset acquisition, three autoTEMs were upgraded with 50-megapixel camera sensors, which increased the frame size and reduced the total number of tiles required per montage to ~2,600

## Volume assembly

The images in the serial section are first corrected for lens distortion effects. A nonlinear transformation of higher order is computed for each section using a set of 10 × 10 highly overlapping images collected at regular intervals during imaging[64]. The lens distortion correction transformations should represent the dynamic distortion effects from the TEM lens system and hence require an acquisition of highly overlapping calibration montages at regular intervals. Overlapping image pairs are identified within each section and point correspondences are extracted for every pair using a feature based approach. In our stitching and alignment pipeline, we use SIFT (scale invariant feature transform) feature descriptors to identify and extract these point correspondences. Per image transformation parameters are estimated by a regularized solver algorithm. The algorithm minimizes the sum of squared distances between the point correspondences between

these tile images. Deforming the tiles within a section based on these transformations results in a seamless registration of the section. A down-sampled version of these stitched sections are produced for estimating a per section transformation that roughly aligns these sections in 3D. A process similar to 2D stitching is followed here, where the point correspondences are computed between pairs of sections that are within a desired distance in z direction. The per section transformation is then applied to all the tile images within the section to obtain a rough aligned volume. MIPmaps are utilized throughout the stitching process for faster processing without compromise in stitching quality.

The rough aligned volume is rendered to disk for further fine alignment. The software tools used to stitch and align the dataset are available in our github repository (https://github.com/AllenInstitute/asap-modules). The volume assembly process is entirely based on image metadata and transformations manipulations and is supported by the Render service (https://github.com/saalfeldlab/render).

Cracks larger than 30 μm in 34 sections were corrected by manually defining transforms. The smaller and more numerous cracks and folds in the dataset were automatically identified using convolutional networks trained on manually labelled samples using $64 \times 64 \times 40$ nm³ resolution image. The same was done to identify voxels which were considered tissue. The rough alignment was iteratively refined in a coarse-to-fine hierarchy[77], using an approach based on a convolutional network to estimate displacements between a pair of images[78]. Displacement fields were estimated between pairs of neighbouring sections, then combined to produce a final displacement field for each image to further transform the image stack. Alignment was first refined using $1,024 \times 1,024 \times 40$ nm³ images, then $64 \times 64 \times 40$ nm³ images.

The composite image of the partial sections was created using the tissue mask previously computed. Pixels in a partial section which were not included in the tissue mask were set to the value of the nearest pixel in a higher-indexed section that was considered tissue. This composite image was used for downstream processing, but not included with the released images.

## Segmentation

Remaining misalignments were detected by cross-correlating patches of image in the same location between two sections, after transforming into the frequency domain and applying a high-pass filter. Combining with the tissue map previously computed, a mask was generated that sets the output of later processing steps to zero in locations with poor alignment. This is called the segmentation output mask.

Using the outlined method[79], a convolutional network was trained to estimate inter-voxel affinities that represent the potential for neuronal boundaries between adjacent image voxels. A convolutional network was also trained to perform a semantic segmentation of the image for neurite classifications, including: (1) soma plus nucleus; (2) axon; (3) dendrite; (4) glia; and (5) blood vessel. Following the described methods[80], both networks were applied to the entire dataset at $8 \times 8 \times 40$ nm³ in overlapping chunks to produce a consistent prediction of the affinity and neurite classification maps. The segmentation output mask was applied to the predictions.

The affinity map was processed with a distributed watershed and clustering algorithm to produce an over-segmented image, where the watershed domains are agglomerated using single-linkage clustering with size thresholds[81,82]. The over-segmentation was then processed by a distributed mean affinity clustering algorithm[81,82] to create the final segmentation. We augmented the standard mean affinity criterion with constraints based on segment sizes and neurite classification maps during the agglomeration process to prevent neuron-glia mergers as well as axon–dendrite and axon–soma mergers.

## Synapse detection and assignment

A convolutional network was trained to predict whether a given voxel participated in a synaptic cleft. Inference on the entire dataset was processed using the methods described previously[80] (using $8 \times 8 \times 40$ nm³ images). These synaptic cleft predictions were segmented using connected components, and components smaller than 40 voxels were removed.

A separate network was trained to perform synaptic partner assignment by predicting the voxels of the synaptic partners given the synaptic cleft as an attentional signal[83]. This assignment network was run for each detected cleft, and coordinates of both the presynaptic and postsynaptic partner predictions were logged along with each cleft prediction.

To evaluate precision and recall, we manually identified synapses within 70 small subvolumes ($n = 8,611$ synapses) spread throughout the dataset[84].

## Nucleus detection

A convolutional network was trained to predict whether a voxel participated in a cell nucleus. Following the methods described previously[80], a nucleus prediction map was produced on the entire dataset at $64 \times 64 \times 40$ nm³. The nucleus prediction was thresholded at 0.5, and segmented using connected components.

## Proofreading

Extensive manual, semi-automated, and fully automated proofreading of the segmentation data was performed by multiple teams to improve the accuracy of the neural circuit reconstruction.

Critical to enabling these coordinated proofreading activities is the central ChunkedGraph system[1,85,86], which maintains a dynamic segmentation dataset, and supports real-time collaborative proofreading on petascale datasets though scalable software interfaces to receive edit requests from various proofreading platforms and support querying and analysis on edit history.

Multiple proofreading platforms and interfaces were developed and leveraged to support the large-scale proofreading activities performed by various teams at Princeton University, the Allen Institute for Brain Science, Baylor College of Medicine, the Johns Hopkins University Applied Physics Laboratory, and ariadne.ai (individual proofreaders are listed in Acknowledgements). Below we outline the methods for these major proofreading activities focused on improving the completeness of neurons within and proximal to the main cortical column, splitting of merged multi-soma objects distributed throughout the image volume, and distributed application of automated proofreading edits to split erroneously merged neuron segments.

**Manual proofreading of dendritic and axonal processes.** Following the methods described previously[26,85,87] proofreaders from Princeton University, the Allen Institute for Brain Science, Baylor College of Medicine, and ariadne.ai used a modified version of Neuroglancer with annotation capabilities as a user interface to make manual split and merge edits to neurons with somata spatially located throughout the dataset. The choice of which neurons to proofread was based on the scientific needs of different projects, which are described in the accompanying studies[4,5,7,10].

Proofreading was aided by on-demand highlighting of branch points and tips on user-defined regions of a neuron based on rapid skeletonization (https://github.com/AllenInstitute/Guidebook). This approach quickly directed proofreader attention to potential false merges and locations for extension, as well as allowed a clear record of regions of an arbor that had been evaluated.

For dendrites, we checked all branch points for correctness and all tips to see if they could be extended. False merges of simple axon fragments onto dendrites were often not corrected in the raw data, since they could be computationally filtered for analysis after skeletonization (see next section). Detached spine heads were not comprehensively proofread. Dendrites that were proofread are identified in CAVE table proofreading_status_and_strategy as status_dendrite = "true".

For axons, we began by 'cleaning' axons of false merges by looking at all branch points. We then performed an extension of axonal tips, the degree of this extension depended on the scientific goals of the different project. The different proofreading strategies were as follows:
(1) Comprehensive extension: each axon end and branch point was visited and checked to see if it was possible to extend until either their biological completion or reached an incomplete end (incomplete ends were due to either the axon reaching the borders of the volume or an artefact that curtailed its continuation). Label: axon_fully_extended.
(2) Substantial extension: each axon branch point was visited and checked, many but not all ends were visited and many but not all ends were done. Label: axon_partially_extended.
(3) Inter_areal_extension: a subset of axons that projected either from a HVA to V1, or from V1 to a HVA were preferentially extended to look specifically at inter-areal connections. Label: axon_interareal
(4) Local cylinder cutting: a subset of pyramidal cells were proofread in a local cylinder which had a 300-µm radius centred around the column featured in Schneider-Mizell et al.[4]. For layer 2/3 cells the cylinder had a a floor at the layer 4/5 border, for layer 4 cells it had a floor at the layer 5/6 border. Any axon leaving the cylinder was cut and
(5) At least 100 synapses: axons were extended until at least 100 synapses were present on the axon to get a sampling of their output connectivity profile. Label: also axon_partially_extended.

Axons that were proofread are identified in CAVE table proofreading_status_and_strategy as status_axon='true' and the proofreading strategy label associated with each axon is described in the column 'strategy_axon'.

**Manual proofreading to split incorrectly merged cells.** Proofreading was also performed to correctively split multi-soma objects containing more than one neuronal soma, which had been incorrectly merged from the agglomeration step in the reconstruction process. This proofreading was performed by the Johns Hopkins University Applied Physics Laboratory, Princeton University, the Allen Institute for Brain Science, and Baylor College of Medicine. These erroneously merged multi-soma objects were specifically targeted given their number, distribution throughout the volume, and subsequent impact on global neural connectivity[88] (Extended Data Fig. 3). As an example, multi-soma objects comprised up to 20% of the synaptic targets for 78 excitatory cells that with proofreading status 'comprehensive extension'. Although the majority of multi-soma objects contained 2 to 25 nuclei (Extended Data Fig. 3a), one large multi-soma object contained 172 neuronal nuclei due to proximity to a major blood vessel present in a substantial portion of the image volume.

Different Neuroglancer web-based applications[1,85,86,88] were used to perform this proofreading, but most edits were performed using NeuVue[88]. NeuVue enables scalable task management across dozens of concurrent users, as well as provide efficient queuing, review, and execution of proofreading edits by integrating with primary data management APIs such as CAVE and PCG. Multi-soma objects used to generate proofreading tasks were originally identified using the nucleus detection table available through CAVE. Additionally, algorithms were employed in a semi-automated workflow to detect the presence of incorrect merges and proposed potential corrective split locations in the segmentation for proofreaders to review and apply[2].

**Proofreading through automated error-detection and correction framework.** Following methods described elsewhere[2] automated error-detection and error-correction methods were utilized using the Neural De-composition (NEURD) framework to apply edits to split incorrectly merged axonal and dendritic segments distributed across the image volume. These automated methods leveraged graph filter and graph analysis algorithms to accurately identify errors in the reconstruction and generate corrective solutions. Validation and refinement of these methods were performed through manual review of proposed automated edits through the NeuVue platform[88].

## Co-registration

**Transform.** We initially manually matched 2,934 fiducials between the EM volume and the 2P structural dataset (1,994 somata and 942 blood vessels, mostly branch points, which are available as part of the resource). Though the fiducials cover the total volume of the dataset it is worth noting that below 400 µm from the surface there is much lower signal to noise in the 2P structural dataset requiring more effort to identify somata, therefore we made use of more vascular fiducials. The fiducial annotation was done using a down-sampled EM dataset with pixel sizes 256 nm ($x$), 256 nm ($y$) and 940 nm ($z$).

Using the fiducials, a transform between the EM dataset and the 2P structural stack was calculated (Methods). To evaluate the error of the transform we evaluated the distance in micrometres between the location of a fiducial after co-registration and its original location; a perfect co-registration would have residuals of 0 µm. The average residual was 3.8 µm.

For calculating the transform we introduced a staged approach to separate the gross transformation between the EM volume and the 2P space from the finer nonlinear deformations needed to get good residuals. This was done by taking advantage of the infrastructure created for the alignment of the EM dataset described above.

The full 3D transform is a list of eight transforms that fall into four groups with different purposes:
(1) The first group is a single transform that is a second-order polynomial transform between the two datasets. This first group serves to scale and rotate the optical dataset into EM space, followed by a single global nonlinear term, leaving an average residual of ~10 µm.
(2) The second group of transforms addresses an issue we saw in the residuals: there were systematic trends in the residuals, both positive and negative, that aligned well with the EM $z$ axis. These trends are spaced in a way that is indicative of changing shape of the EM data on approximately the length scale between knife cleanings or tape changes. We addressed this with a transform that binned the data into $z$ ranges and applied a further second-order polynomial to each bin. We did this in a 2-step hierarchical fashion, first with 5 $z$ bins, followed by a second with 21 $z$ bins. These steps removed the systematic trends in the residuals versus $z$ and the average residuals dropped to 5.6 µm and 4.6 µm respectively.
(3) The third group is a set of hierarchical thin plate spline transforms. We used successively finer grids of control points of even $n \times n \times n$ spacing in the volume. We used 4 steps with $n = [3, 5, 10, 12]$. The idea here is to account for deformations on larger length scales first, so that the highest order transforms introduce smaller changes in position. The average residuals in these steps were 3.9, 3.5, 3.1 and 2.9 µm accomplished with average control point motions of 12.5, 7.5, 3.8 and 1.6 µm.
(4) The final group is a single thin plate spline transform. The control points for this transform are no longer an evenly spaced grid. Instead, each fiducial point is assigned to be a control point. This transform minimizes the residuals almost perfectly (as it should for the control points which are identical to the fiducials; 0.003 µm on average; Fig. 3) and accomplishes this final step by moving each data point on average another 2.9 µm. This last transform is very sensitive to error in fiducial location but provides the co-registration with minimal residuals. This last transform is also more likely to create errors in regions with strong distortions, as for example the edges of the dataset.

Since the nature of transform 4 is to effectively set the residuals to zero for the control points, we used a new measure to evaluate the error of the transform. We created 2,933 3D transforms, each time leaving

out one fiducial and then evaluated the residual of the left-out point. We call this measure 'leave-one-out' residuals and it evaluates how well the transform does with a new point.

**Assigning manual matches.** A custom user interface was used to visualize images from both the functional data and EM data side-by-side to manually associate functional ROIs to their matching EM cell counterpart and vice versa. To visualize the functional scans, summary images were generated by averaging the scan over time (average image) and correlating pixels with neighbour pixels over time (correlation image). The product of the average and correlation images were used to clearly visualize cell body locations. Using the per field affine registration into the 2P structural stack (Fig. 3b), a representative image of labelled vasculature corresponding to the registered field was extracted from the red channel of the stack. EM imagery and EM nucleus segmentation was resized to $1 \mu m^3$ resolution, and transformed into the 2P structural stack coordinates using the co-registration transform, allowing an EM image corresponding to the registered field to be extracted. The overlay of the extracted vessel field and extracted EM image were used to confirm local alignment of the vasculature visible in both domains. Soma identity was assessed by comparing the spatial structure of the target soma and nearby somas in the functional image to soma locations from the EM cell nuclei image. Using the tool, matchers generated a hypothesis for which EM cell nucleus matched to a given functional unit or vice versa. A custom version of Neuroglancer (Seung laboratory; https://github.com/seung-lab/neuroglancer) was used to visualize the region of interest in the ground truth EM data for match confirmation. The breakdown in the number of unique neuron matches per 2P scan is shown in Extended Data Fig. 4a. The resulting matches are uploaded to CAVE table coregistration_manual_v4. The latest recommended manual match table can be found at https://www.microns-explorer.org/cortical-mm3#f-coreg.

**Evaluating manual matches.** In addition to the matches, the manual co-registration table includes two metrics that help assess confidence. The first, residual, measures the distance between the matched 2P functional unit centroid and EM neuron soma centroid, after transformation with the EM to 2P fiducial-based transform (Extended Data Fig. 4b, top). The second metric, separation score, measures the difference in residuals between the match and the nearest non-matched EM neuron. (Extended Data Fig. 4b, bottom) Negative separation indicates that the nearest EM neuron to the functional unit after transformation was not chosen by the matchers. Smaller residuals and larger separation scores indicate higher confidence matches, as is the case for a majority of matches (Extended Data Fig. 4c). To help validate the manual matches, for every EM neuron that was independently matched to at least two scans, the in vivo signal correlation (correlation between trial-averaged responses to oracle stimuli) was computed between the matched unit in scan A to the matched unit in scan B. In addition, for each neuron, two control correlations were computed, the matched unit in scan A to the nearest unit not matched to the neuron in scan B, and vice versa (Extended Data Fig. 4d). As expected, the distribution of oracle scores between the matched neurons and control neurons are qualitatively similar, with a slight right-shift towards higher oracle scores for matches, as higher oracle scores were prioritized for matching (Extended Data Fig. 4e). The comparison of signal correlation between matched neurons and their control counterparts exhibits a strong trend, with a clustering in the upper left quadrant and most data points positioned above the diagonal. This indicates that the matched neurons consistently have stronger signal correlations compared to their nearest counterparts, and high signal correlation overall, especially when the oracle score is larger (Extended Data Fig. 4f). Filtering by oracle score further refines the trend, highlighting that high oracle score neurons (score >0.2) show even more distinct separation, with matched neurons maintaining superior signal correlation values compared to the nearest-neighbour matches (Extended Data Fig. 4g,h).

**Generating the fiducial-based automatch.** To generate the fiducial-based automatch, we utilized the EM-to-2P co-registration transform to map all EM neuron nucleus centroids (retrieved from the CAVE table nucleus_neuron_svm) into the 2P functional space. Next, we applied the minimum weight matching algorithm for bipartite graphs[89] using the linear_sum_assignment function from the scipy.optimize module[90] to perform the matching. The resulting automatch table is stored in the CAVE table coregistration_auto_phase3_fwd, which also includes the associated residual and separation scores. The latest recommended fiducial-based automatch table can be found at https://www.microns-explorer.org/cortical-mm3#f-coreg.

**Vessel-based co-registration and automatch.** To achieve co-registration starting with the 2P structural stacks and EM segmentation and without the use of fiducials, we employed a multi-scale B-spline registration[91] using only vasculature data. This non-rigid transformation method corrects the extreme nonlinear tissue distortions caused by shrinkage from 2P to EM. Both the EM segmentation and the 2P structural stack volumes were subsampled to match 1-μm voxel resolution, ensuring consistent scaling and indexing between the volumes.

Pre-processing on the vessels was necessary to address inconsistent signal quality in the 2P data, especially for vessels located deeper in the cortex, which emit lower fluorescence. A Meijering neurite filter[92] was applied to the vessels, using the eigenvectors of the Hessian matrix to detect vessels effectively.

An additional filtering step mitigated discrepancies in z resolution and errors from false splits in the EM segmentation. To address the z direction smearing in 2P due to anisotropy, both the 2P and EM volumes were binarized, skeletonized and further processed by removing small isolated segments. A Gaussian filter was convolved over the skeletons, forming tubes of constant radius for co-registration. Another round of skeletonization and Gaussian filtering was applied to correct for false splits in thicker vessels.

The final co-registration was computed using SimpleITK's B-spline algorithm[93], treating the EM volume as the 'moving' volume. Initially, centroid alignment was achieved via template matching within a small subvolume. Despite tissue shrinkage, the volumes were locally aligned well enough to yield good correlations. The B-spline transformation was performed across multiple scales, progressing from coarse grids with strong smoothing to finer grids with minimal smoothing. The Limited-Memory Broyden-Fletcher-Goldfarb-Shanno (L-BFGS) optimizer with 600 iterations was used, sampling 1% of the points to handle large matrices. The resulting flow field and its inverse defined how each voxel mapped between spaces.

For the final step, the flow field was applied to both the EM nuclear segmentation and the 2P unit centroids. Minimum weight matching was performed (as described in 'Generating the fiducial-based automatch') to establish match assignments, using excitatory neuron centroids from the CAVE table aibs_metamodel_mtypes_v661_v2. The final table is uploaded to apl_functional_coreg_vess_fwd with the associated residual and separation scores. The latest recommended vessel-based automatch table can be found at https://www.microns-explorer.org/cortical-mm3#f-coreg.

**Generating the fiducial-vessel agreement automatch table.** To generate the fiducial-vessel agreement automatch table, first, for each table described above (coregistration_auto_phase3_fwd, apl_functional_coreg_vess_fwd), the residual and separation scores were transformed into percentiles. Then, the two tables were merged on keys: 'session', 'scan_idx', 'field', 'unit_id' and 'target_id'.

**Evaluating automatch tables.** To evaluate the automatch tables, we computed precision and recall using manual matches as ground truth. To ensure a fair comparison, we first restricted both the automatch and manual match tables to only contain rows where the functional unit or

EM neuron was commonly attempted. For calculating precision and recall, true positives were rows common to both tables, false positives were rows only in the automatch table, and false negatives were rows only in the manual match table. The precision-recall curves can be used to select an automatch, and/ or a metric with which to threshold matches (Extended Data Fig. 5a). In addition, heat maps are provided indicating precision levels (Extended Data Fig. 5b) and number of automatches remaining (Extended Data Fig. 5c) for jointly applied residual and separation percentile thresholds. To apply a threshold, first convert the residual and separation (named 'score' in the table) to percentiles. Then for residual, apply the threshold as a maximum, taking the matches below the threshold. Conversely, for separation, apply the threshold as a minimum.

### Cell classification

We analysed the nucleus segmentations for features such as volume, surface area, fraction of membrane within folds and depth in cortex. We trained a support vector machine (SVM) machine classifier to use these features to detect which nucleus detections were likely neurons within the volume, with 96.9% precision and 99.6% recall. This model was trained based upon data from an independent dataset, and the performance numbers are based upon evaluating the concordance of the model with the manual cell-type calls within the volume. This model predicted 82,247 neurons detected within the larger subvolume. For the neurons, we extracted additional features from the somatic region of the cell, including its volume, surface area, and density of synapses. Dimensionality reduction on this feature space revealed a clear separation between neurons with well-segmented somatic regions ($n = 69,957$) from those with fragmented segmentations or sizable merges with other objects ($n = 12,290$). Combining those features with the nucleus features, we trained a multi-layer perceptron classifier to distinguish excitatory from inhibitory neurons among the well-segmented subset, using the 80% of the manual labelled data as a training set, and 20% as a validation set to choose hyper-parameters. After running the classifier across the entire dataset, we then tested the performance by sampling an additional 350 cells (250 excitatory and 100 inhibitory). We estimate from this test that the classifier had an overall accuracy of 97% with an estimated 96% precision and 94% recall for inhibitory calls.

### Reporting summary

Further information on research design is available in the Nature Portfolio Reporting Summary linked to this article.

## Data availability

EM imagery, segmentation and annotation data is available via https://www.micronsexplorer.org/cortical-mm3 and from https://bossdb.org/project/microns-minnie.

## Code availability

Code for analysis and generation of figures was generated in Python Jupiter notebooks and is available at https://github.com/AllenInstitute/MicronsFunctionalConnectomics, making extensive use of CAVE analysis infrastructure[1] (available at https://github.com/CAVEconnectome) and CloudVolume[94] to interact with data infrastructure, and libraries Matplotlib[95], Numpy[96] and Pandas for general computation and data visualization.

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

**Acknowledgements** The authors thank D. Markowitz, the IARPA MICrONS Program Manager, who coordinated this work during all three phases of the MICrONS programme; IARPA programme managers J. Vogelstein and D. Markowitz for co-developing the MICrONS programme; J. Wang, IARPA SETA for her assistance; J. Philips, S. Coulter and the Program Management team at the AIBS for their guidance for project strategy and operations; H. Zeng, E. Lein, C. Koch and A. Jones for their support and leadership; the Manufacturing and Processing Engineering team at the AIBS for their help in implementing the EM imaging and sectioning pipeline; B. Youngstrom, S. Kendrick and the Allen Institute IT team for support with infrastructure, data management and data transfer; the facilities, finance and legal teams at the AIBS for their support on the MICrONS contract; S. Saalfeld, K. Khairy and E. Trautman for help with the parameters for 2D stitching and rough alignment of the dataset; Z. Hanson and J. Singh for their contribution to manual matching of functional ROIs to EM nuclei; D. Kim for his contribution to pupil tracking; R. Raju for his contribution to parametric stimuli development; A. Mok and D. Ouzounov for their contribution to three-photon imaging development; G. McGrath for computer system administration; M. Husseini, L. Jackel and J. Jackel for project administration at Princeton University; S. Hider, T. Gion, D. Pryor, D. Kleissas, L. Rodriguez, M. Wilt and the team from the John Hopkins University Applied Physics Laboratory (APL), as well as Marysol Encarnación and Martha Cervantes from the CIRCUIT Program at APL for supporting data assessments on the neural circuit reconstruction and data infrastructure

through the Brain Observatory Storage Service and Database (BossDB; https://bossdb.org/; NIH/NIMH R24 MH114785); F. Chance, B. Aimone and everyone at Sandia National Laboratories for their support and assistance; the 'Connectomics at Google' team for developing Neuroglancer and computational resource donations, in particular J. Maitin-Shepard for authoring Neuroglancer and help creating the reformatted sharded multi-resolution meshes and imagery files used to display the data; Amazon, the AWS Open Data Program, and the AWS Open Science platform for providing data and computational resources; and Intel for their assistance. The authors also thank the following individuals for their work proofreading neurons in the MICrONS dataset: N. Smith (24,101 edits), D. Panchal (19,384 edits), M. Cook (17,088 edits), C. Ordish (14,333 edits), Niyati (13,897), Z. Sorangwala (13,777 edits), Nirali (13,317 edits), Sholka (11,569 edits), K. Shah (10,570 edits), D. Patel (10,368 edits), Dhara (9,871 edits), Anuja (9,337 edits), Zeba (8,742 edits), A. Rajput (8,674 edits), C. Smith (8,281 edits), Hemal (8,084 edits), Harshil (8,022 edits), C. Knecht (7,199), S. Pal (7,036 edits), D. Rami (6,850 edits), Sweksha (6,766 edits), Priyanka (6,485 edits), Yashvi (6,306 edits), Frank (5,711 edits), K. Raval (5,638 edits), D. Dalal (5,597 edits), E. Phillips (5,454 edits), Hetvi (5,358 edits), Yuvaraju (4,787 edits), G. Hopkins (4,505 edits), Neha (2,968 edits), A. Pandey (2,426 edits), Vaishakhi (2,227 edits), Twinkal (1,270 edits), D. Parodi (1,007 edits), Kinjal (980 edits), R. Xu (947 edits), Kashish (911 edits), C. Moore (828 edits), V. Lung (804 edits), S. Wu (746 edits), T. Gaito (643 edits), J. Gayk (570 edits), L. Fozo (506 edits), Diksha (438 edits), M. Baptiste (416 edits), E. Macgregor (383 edits), E. Miranda (354 edits), S. Bare (220 edits). The work was supported by the Intelligence Advanced Research Projects Activity (IARPA) via Department of Interior/ Interior Business Center (DoI/IBC) contract numbers D16PC00003, D16PC00004, D16PC0005 and 2017-17032700004. The US Government is authorized to reproduce and distribute reprints for Governmental purposes notwithstanding any copyright annotation thereon. H.S.S. also acknowledges support from NIH/NINDS U19 NS104648, NIH/NEI R01 EY027036, NIH/NIMH U01 MH114824, NIH/NIMH U01 MH117072 NIH/NINDS R01 NS104926, NIH/NIMH RF1 MH117815, NIH/NIMH RF1 MH123400 and the Mathers Foundation, as well as assistance from Google, Amazon and Intel. X.P. acknowledges support from NSF CAREER grant IOS-1552868. X.P. and A.T. acknowledge support from NSF NeuroNex grant 1707400. A.T. acknowledges support from National Institute of Mental Health and National Institute of Neurological Disorders And Stroke under award number U19MH114830. R.C.R. acknowledges support from NSF NeuroNex 2 award 2014862, NIH U24NS120053 and NIH 1RF1MH128840-01. We thank the Allen Institute for Brain Science founder, Paul G. Allen, for his vision, encouragement and support. Disclaimer: the views and conclusions contained herein are those of the authors and should not be interpreted as necessarily representing the official policies or endorsements, either expressed or implied, of IARPA, DoI/IBC, or the US Government.

**Author contributions** Conceptualization: H.S.S., F.C., R.C.R., N.M.d.C., A.S.T., J.R. and X.P. Methodology: J.A.B., M.A.C., S.D., A.H., Z.J., C.J., N.K., K.K., K. Lee, K. Li, R.L., T.M., E. Mitchell, S. Mu, S.S.M., B.N., O.O., S.P., W.S., N.L.T., R.W., W.W., J.W., R.Y., C.M.S.M., A.B., F.C., D.B., J.B., M.T., R.T., G.M., D.B., W.Y., R.C.R., N.M.d.C., L.E., D.K., S.K., T.F., J.R., P.G.F., S.P., E.F., C.X., T.W., F.H.S., D.Y., E.Y.W. and B.C. Software: J.A.B., M.A.C., S.D., A.H., Z.J., C.J., N.K., K.K., K. Ler, K. Li, R.L., T.M., E. Mitchell, S. Mu, S.S.M., B.N., O.O., S.P., W.S., N.L.T., R.W., W.W., J.W., R.Y., C.M.S.M., F.C., D.B., R.T., G.M., W.Y., L.E., D.K., T.F., S.P., E.C., T.M., C.A.B., J.J., L.M.K., V.R., D.X. and J.M. Validation: C.A.B., W.G.R., P.K.R. and J.M. Investigation: J.A.B., M.A.C., S.D., A.H., Z.J., C.J., N.K., K. Lee, K. Li, R.L., T.M., E. Mitchell, S. Mu, S.S.M., B.N., O.O., S.P., H.S.S., W.S., N.L.T., R.W., W.W., J.W., R.Y., C.M.S.M., A.B., F.C., D.B., J.B., M.T., R.T., G.M., D.B., W.Y., R.C.R., N.M.d.C., L.E., D.K., T.F., A.S.T., J.R., P.G.F., S.P., E.F., S.P., E.C., T.M. and X.P. Resource: Z.H.T. Data curation: S.D., J.G., J.H., S.K., M.M., B.S., S.W., K.W., S.Y., C.M.S.M., A.B., F.C., J.B., M.T., N.M.d.C., C.G., P.G.F., S.P., E.F., E.C., S. McReynolds, M.B., E. Miranda, F.Y., A.W., E.J., C.Z., C.A.B., J.J., L.M.K., P.K.R., V.R., B.W., D.X., E.N., R.S., B.D., V.B., G.J.Y.P.V. and Z.M.S. Writing, original draft: S.D., T.M., H.S.S., C.M.S.M., F.C., R.C.R., N.M.d.C., A.S.T., J.R., P.G.F., S.P., C.P., B.C. and X.P. Writing, review and editing: J.A.B., M.A.C., A.H., Z.J., C.J., N.K., K. Lee, K. Li, R.L., E. Mitchell, S. Mu, S.S.M., B.N., O.O., S.P., W.S., N.L.T., R.W., W.W., J.W., R.Y., A.B., D.B., J.B., M.T., R.T., G.M., D.B., W.Y., L.E., D.K., T.F., E.F., S.P., C.X., T.W., E.C., C.L.S., A.R., T.M., P.K.R., J.J., D.X., C.A.B. and B.W. Visualization: S.D., A.S., F.C. and N.M.d.C. Supervision: H.S.S., S.Y., F.C., R.C.R., N.M.d.C., A.S.T., J.R., X.P., W.G.R., L.M.K., P.K.R., B.W. and D.X. Project administration: T.M., N.M.d.C., S.S., J.R., R.Y., W.G.R. and B.W. Funding acquisition: H.S.S., R.C.R., N.M.d.C., A.S.T., J.R., X.P. and B.W.

**Competing interests** S. Seung and T. Macrina disclose a competing interest in ZettaAI; J. Reimer and A. S. Tolias disclose a competing interest in Vathes. The other authors declare no competing interests.

**Additional information**
**Correspondence and requests for materials** should be addressed to Forrest Collman, Nuno Maçarico da Costa, Xaq Pitkow, R. Clay Reid, Jacob Reimer, H. Sebastian Seung or Andreas S. Tolias.

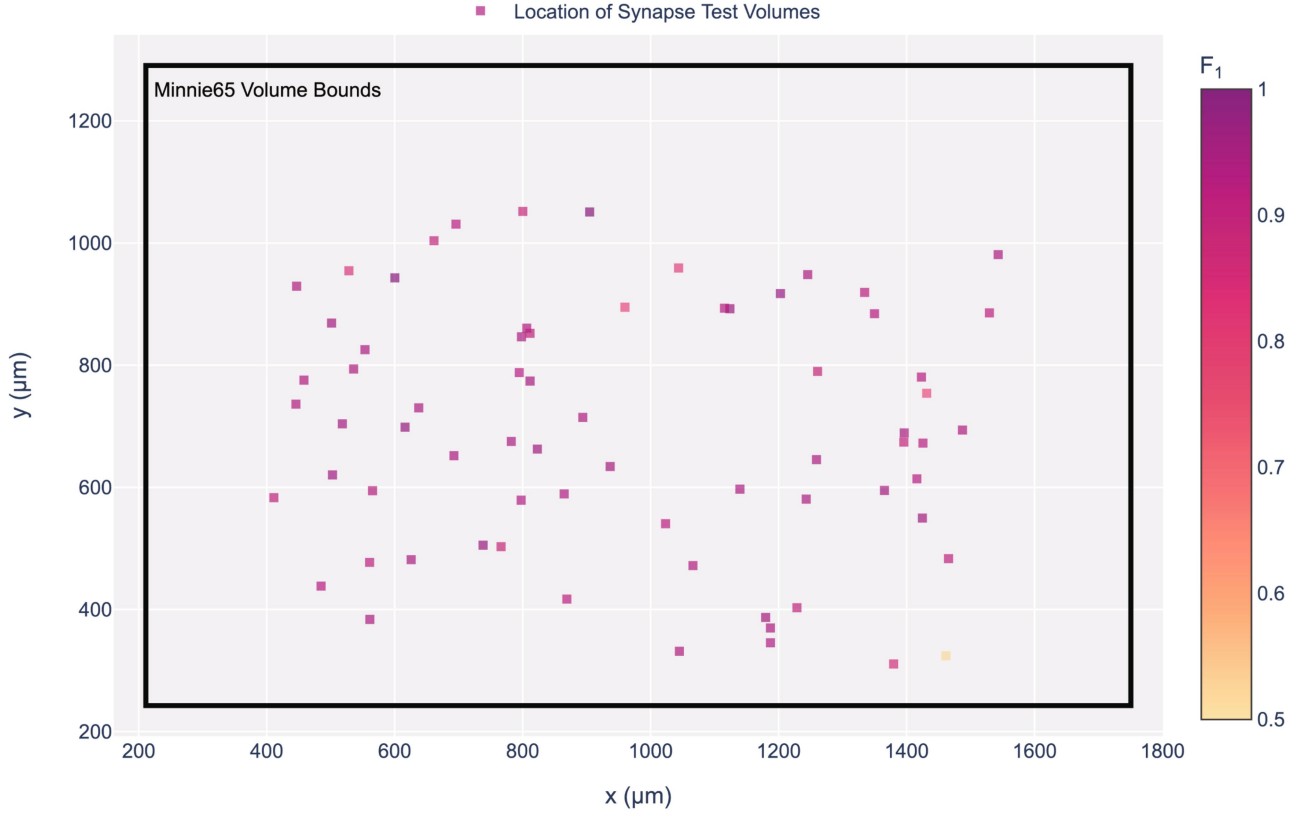

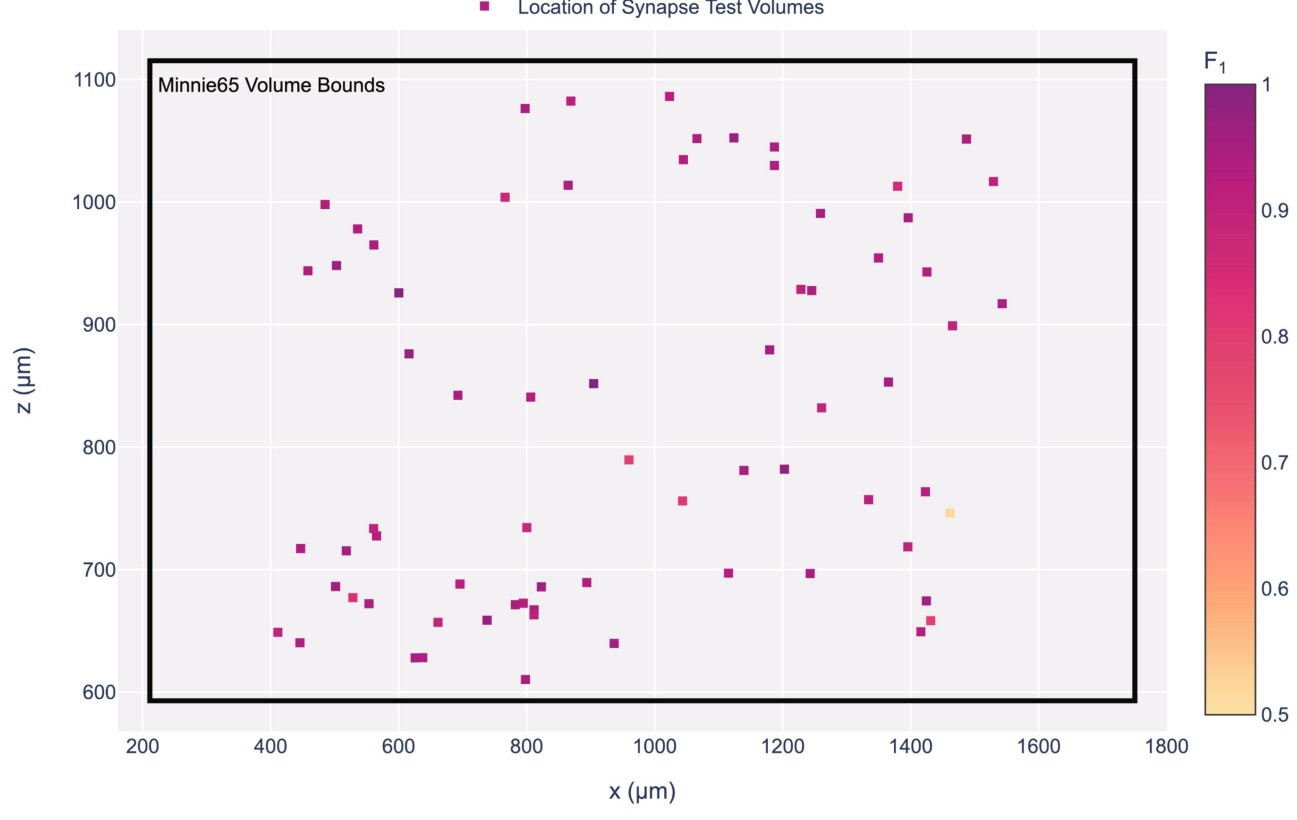

**Extended Data Fig. 1** | See next page for caption.

**Extended Data Fig. 1 | Test Volume Locations for Validation of Automated Synapse Detection.** Location and distribution of test subvolumes (x = 5.5 μm, y = 5.5 μm, z = 5.5 μm) throughout the whole subvolume 65 that were used for validation of automated synaptic contact segmentation. Identification and annotation of synaptic contacts (n = 8,611 synapses) were performed manually within each subvolume and compared with automated results to calculate subvolume and combined precision (96%), recall (89%), and F1 scores (92%), with test subvolume F1 scores visualized by color within each plot. The two panels show a coronal (**a**) and top (**b**) view of the location of the sampling sites. In (**a**) the vertical axis represents the pia to white matter direction and the horizontal axis represents the medial-lateral direction. In (**b**) the vertical axis represents the anterior-posterior direction and the horizontal axis represents the medial-lateral direction.

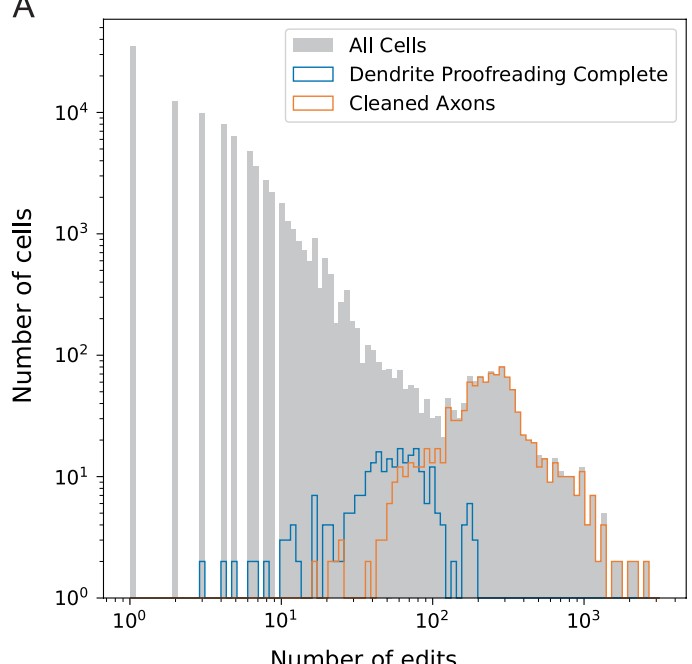

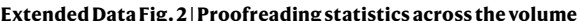

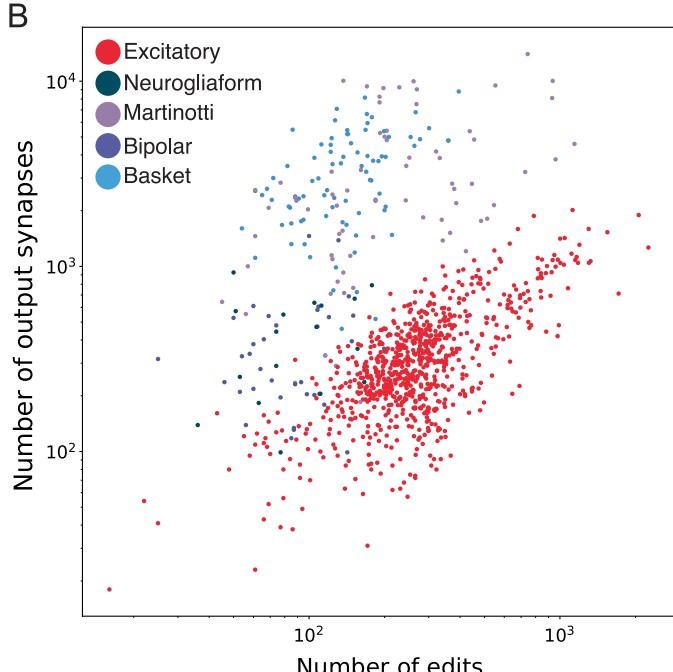

**Extended Data Fig. 2 | Proofreading statistics across the volume.**
**(A)** Histogram of number of edits across all the objects associated with nuclei. Distribution of neurons with complete dendritic proofreading highlighted in blue, and neurons with clean axons in orange. Cells with complete dendritic proofreading have often had some axon edits as well, so this is an upper bound on the number of edits required to fully extend dendrites. Most cells have had very little proofreading and have been mostly touched by automated methods. Note plot is on a log-log scale. **(B)** Number of edits compared to number of output synapses in reconstruction. For all the clean axons for which we have cell type annotations, the number of edits versus the number of outputs is plotted on a log log scale. Data points are colored with respect to their broad cell class. Generally, more extensively reconstructed axons have more edits, but there are also strong cell and cell-type specific effects. This reflects systematic differences in the thickness of axons of different cell types, as well as variation in how much of the axon is contained within the volume and the quality of the segmentation in different locations in the dataset.

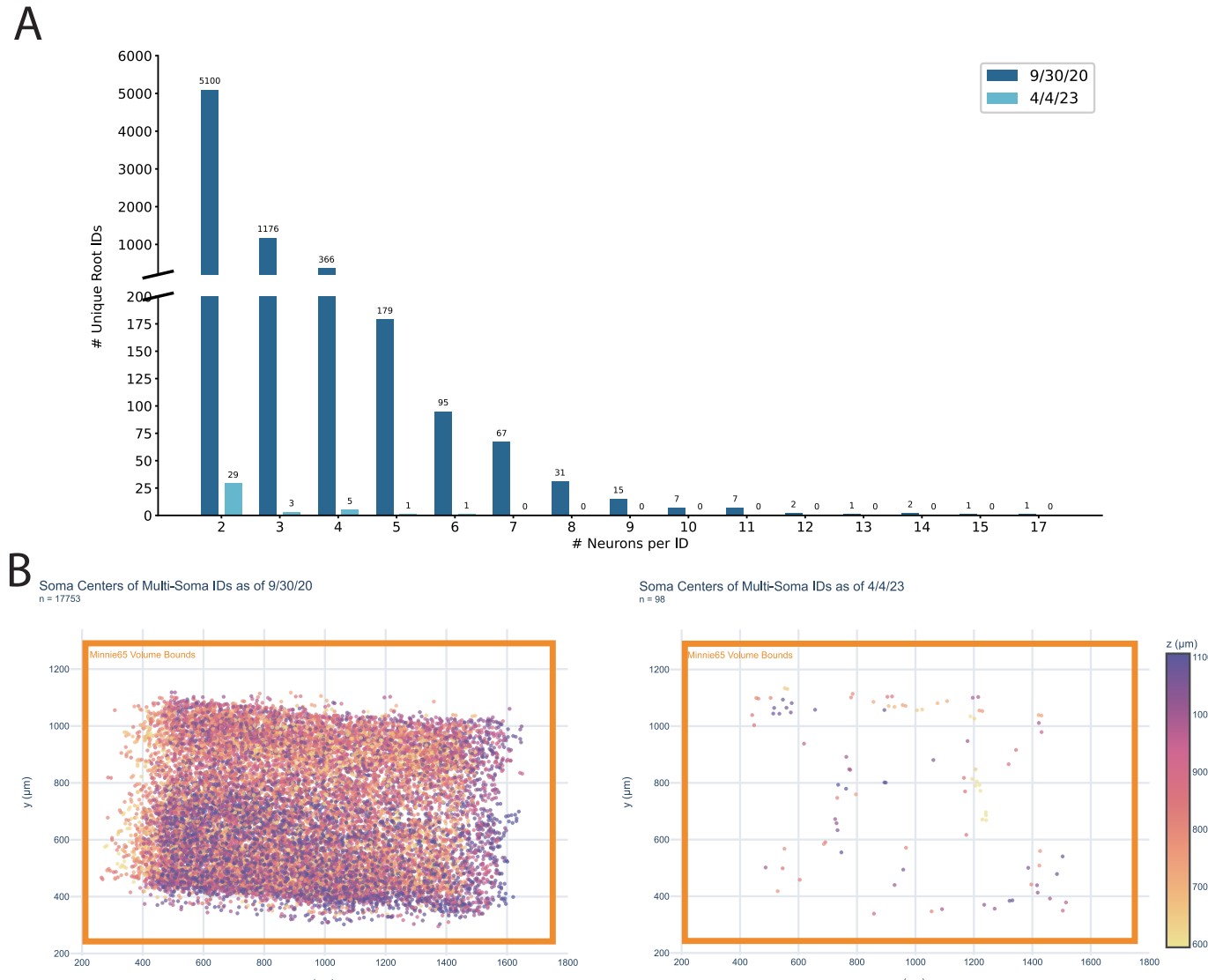

**Extended Data Fig. 3 | Multi-Soma Proofreading.** (A) Distribution of multi-soma IDs by number of neuronal nuclei was monitored throughout proofreading. Difference in multi-neuron root IDs before APL proofreading (dark blue) and after (light blue). Note that this shows the number of neurons per ID, which means that non-neuron somas are not counted. This figure was derived using the soma classification table: nucleus_ref_neuron_svm. Note that a small number of multi-soma IDs were skipped during APL proofreading because they contain low quality neurons merged to myelinated axons or they were falsely classified as neuronal (e.g. blood vessels); (B) Spatial distribution of multi-neuron ID soma centers (soma locations of merged cells containing ≥ 2 neuronal nuclei) before APL proofreading and after. Both are a lateral view of the volume that shows distribution across layers, from pia (top) to white matter (bottom). Color-bar represents depth.

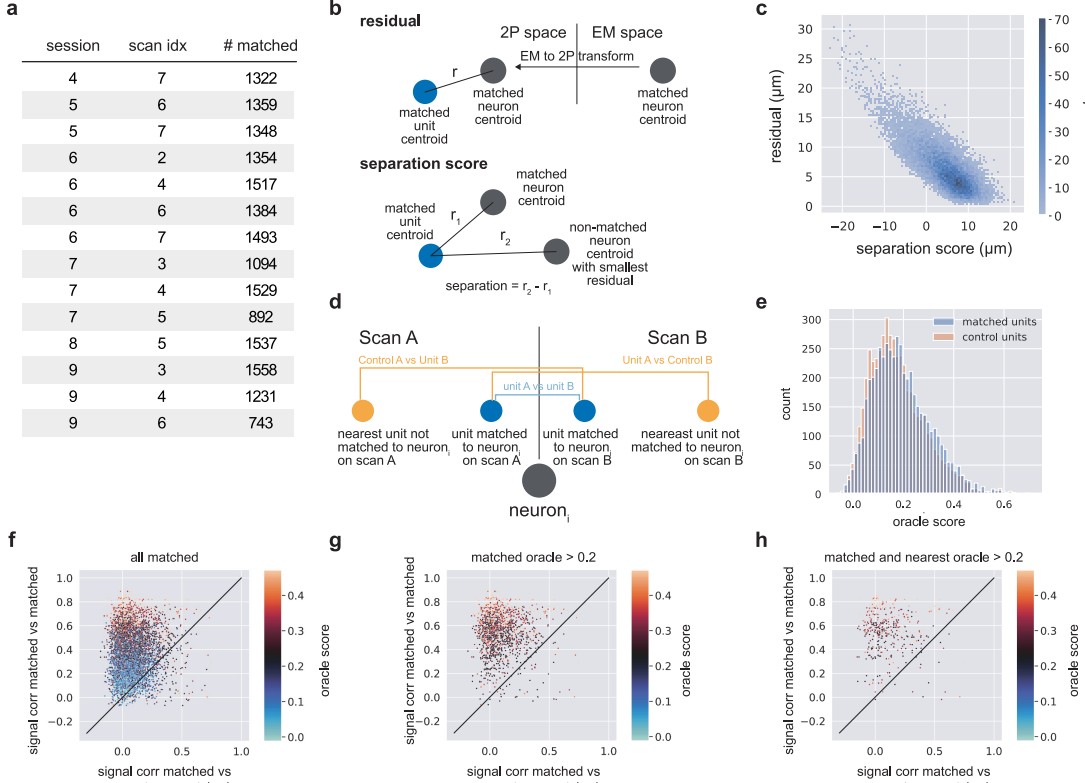

**Extended Data Fig. 4 | Manual co-registration metrics. a)** The number of matched neuronal EM nuclei by session/scan **b)** Schematic of the residual and separation score metric. **Residual:** For a matched EM nucleus to a functional ROI (unit), the residual is computed as the euclidean distance between the nucleus centroid and unit centroid after transforming the nucleus centroid from EM to 2P space with the spline-based co-registration. **Separation score**: For a matched EM nucleus to a functional unit the separation score is computed as the difference between the residual of the matched pair and the residual of the nearest EM neuronal nucleus that was not matched to the unit. **c)** 2D histogram

of separation score and residual. **d)** Schematic of in vivo signal correlation analysis (see Methods). **e)** The distribution of oracle scores for matched units and the nearest unit controls. **f)** Scatter plot of signal correlations for all matched units (y-axis) vs the signal correlations for the nearest unit controls (x-axis) and colored by oracle score. Note that each matched unit pair has two data points on the plot for each of the two control correlations. **g)** Same as in **f)** restricted to matched units with oracle >0.2. **h)** same as in **f)** restricted to matched units and control units with oracle > 0.2.

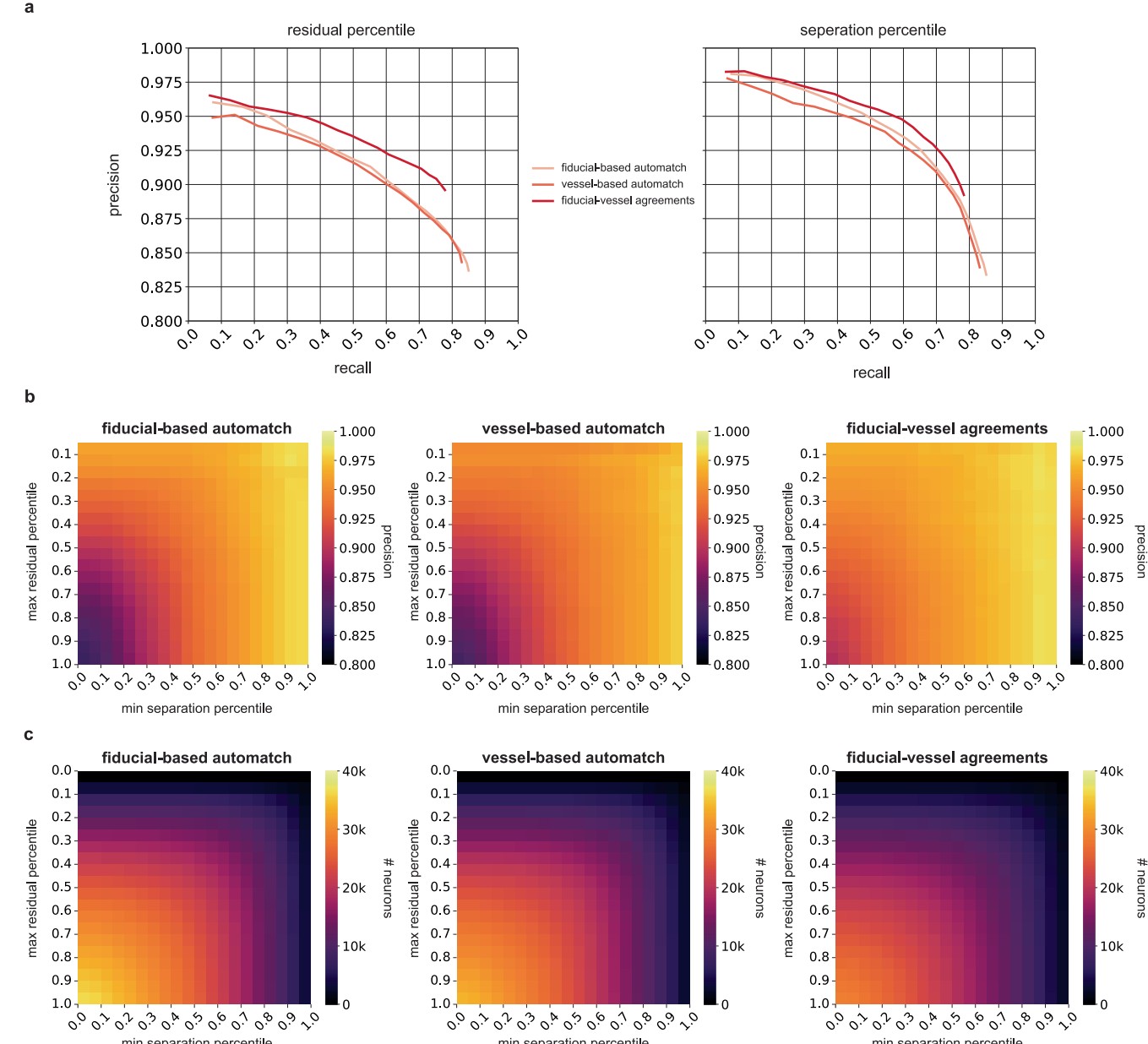

**Extended Data Fig. 5 | Comparison of Fiducial-Based, Vessel-Based, and Combined Automatch Approaches. a)** Precision-recall curves showing performance relative to manual matches (used as the ground truth) across residual (left) and separation percentiles (right) for fiducial-based, vessel-based, and fiducial-vessel agreement automatch methods. **b)** Heatmaps of max residual percentile and min separation percentile colored by precision relative to manual matches, for the fiducial-based (left), vessel-based (middle) and fiducial-vessel agreement (right) automatches. Max residual percentile represents the threshold below which matches were included, while min separation percentile represents the threshold above which matches were included. **c)** Heatmaps of max residual percentile and min separation percentile colored by the number of neurons remaining after thresholds were applied, for the fiducial-based (left), vessel-based (middle) and fiducial-vessel agreement (right) automatches.

**Extended Data Table 1 | Two-photon functional scans in the dataset release**

| session | scan_idx | # frames | # fields | fps |
|---|---|---|---|---|
| 4 | 7 | 40000 | 8 | 6.3 |
| 5 | 6 | 40000 | 8 | 6.3 |
| 5 | 7 | 40000 | 8 | 6.3 |
| 6 | 2 | 40000 | 8 | 6.3 |
| 6 | 4 | 40000 | 8 | 6.3 |
| 6 | 6 | 40000 | 8 | 6.3 |
| 6 | 7 | 40000 | 8 | 6.3 |
| 7 | 3 | 40000 | 8 | 6.3 |
| 7 | 4 | 40000 | 8 | 6.3 |
| 7 | 5 | 40000 | 8 | 6.3 |
| 8 | 5 | 40000 | 8 | 6.3 |
| 9 | 3 | 50000 | 6 | 8.6 |
| 9 | 4 | 50000 | 6 | 8.6 |
| 9 | 6 | 57000 | 4 | 9.6 |

List of scans used for the acquisition of the functional dataset along with their descriptive parameters.

**Extended Data Table 2 | Data Access Tools**

| Tool Name | Data Products | URLs |
|---|---|---|
| cloud-volume | EM imagery, voxel segmentation and meshes | https://www.github.com/seung-lab/cloud-volume |
| MeshParty | EM meshes and skeletons | https://github.com/sdorkenw/MeshParty |
| CAVEclient | EM structural annotations (synapses, cell-types, etc), segmentation edits, local geometric properties | https://github.com/seung-lab/CAVEclient |
| microns-nda | Functional imaging DataJoint database | https://github.com/cajal/microns-nda-access |

A list of programmatic open source tools for data access, what data products they help you access, and a link to their open source repository where there are further instructions and examples.

## Extended Data Table 3 | CAVE Tables

| Table Name | # of Annotations | Description |
|---|---|---|
| pni_synapses_v2 | 337,312,429 | The locations of synapses and the segment ids of the pre and post-synaptic automated synapse detection. |
| nucleus_detection_v0 | 144,120 | The locations of nuclei detected via a fully automated method. |
| nucleus_alternative_points | 8,388 | A reference annotation table marking alternative segment_id lookup locations for a subset of nuclei in nucleus_detection_v0 that is more accurate than the centroid location listed there. |
| nucleus_ref_neuron_svm | 144,120 | A reference annotation indicating the output of a model detecting which nucleus detections are neurons versus which are not [6]. |
| coregistration_manual_v4 | 19,181 | A table indicating the association between individual units in the functional imaging data and nuclei in the structural data, derived from human powered matching. Includes residual and separation scores to help assess confidence. |
| coregistration_auto_phase3_fwd | 84,198 | A table indicating the association between individual units in the functional imaging data and nuclei in the structural data, derived from the automated procedure using the Allen Institute EM > 2P spline based transform. Includes residuals and separation scores to help assess confidence. |
| apl_functional_coreg_vess_fwd | 75,856 | A table indicating the association between individual units in the functional imaging data and nuclei in the structural data, derived from the automated procedure using vessels. Includes residuals and separation scores to help assess confidence. |
| proofreading_status_and_strategy | 1,621 | A table indicating which neurons have been proofread on their axons or dendrites and what strategy was used. |
| aibs_column_nonneuronal_ref | 542 | Cell type reference annotations from a human expert of non-neuronal cells located amongst the column defined by [4]. |
| allen_v1_column_types_slanted_ref | 1,357 | Neuron cell type reference annotations from human experts of neuronal cells located amongst the column defined by [4]. |
| allen_column_mtypes_v2 | 1,351 | Neuron cell type reference annotations from data driven unsupervised clustering of neuronal cells |
| aibs_metamodel_mtypes_v661_v2 | 72,158 | Reference annotations indicating the output of a model predicting cell types across the dataset based on the labels from allen_column_mtypes_v1 [6]. |
| aibs_metamodel_celltypes_v661 | 94,014 | Reference annotations indicating the output of a model predicting cell classes based on the labels from allen_v1_column_types_slanted_ref and aibs_column_nonneuronal_ref. [6] |
| baylor_log_reg_cell_type_coarse_v1 | 55,063 | Reference annotations indicated the output of a logistic regression model predicting whether the nucleus is part of an excitatory or inhibitory cell [2]. |
| baylor_gnn_cell_type_fine_model_v2 | 49,051 | Reference annotations indicated the output of a graph neural network model predicting the cell type based on the human labels in allen_v1_column_types_slanted_ref. [2] |
| pt_synapse_targets | 4,528 | A reference annotation table on synapses that include manual corrections and identification of excitatory versus inhibitory targets used in Bodor et. al 2023. |
| l5et_column | 39 | A reference annotation for layer 5 ET cells in the column that had fully extended axons that were used in Bodor et. al 2023 |
| bodor_pt_target_proofread | 56 | A proofread collection of numerically strong BC and MC targets downstream of the L5-ET cells in the table bodor_pt_cells. |
| bodor_pt_cells | 12 | A list of layer 5 thick tufted neurons analyzed in Bodor et al 2023 |
| nucleus_functional_area_assignment | 144120 | A reference table on nuclei that assigns each nucleus to a functional area based on where it lives in the in visual cortex. |
| functional_properties_v3_bcm | 12,094 | A table that contains basic statistics about the visual response properties of neurons in the dataset for rapid analysis. |
| vortex_compartment_targets | 116,282 | A table of manual annotations of what kind of compartment (shaft, soma, spine) a subset of neurons makes, funded by VORTEX. |
| vortex_manual_nodes_of_ranvier | 32 | A table of manual annotations of locations of nodes of ranvier, funded by VORTEX. |
| vortex_astrocyte_proofreading_status | 26 | A table of astrocytes that have undergone cleaning to remove falsely merged objects, funded by VORTEX. |
| vortex_manual_myelination_v0 | 53,517 | A manual table of locations along axons where a subsets of cells are myelinated, funded by VORTEX. |

List of annotation tables that are part of the public release. Each table can be queried via the CAVE client and downloaded as a CSV from publicly available locations.

Nuno Macarico da Costa
Forrest Collman
Jacob Reimer
Xaq Pitkow
Andreas S. Tolias
R. Clay Reid

# Reporting Summary

## Statistics

For all statistical analyses, confirm that the following items are present in the figure legend, table legend, main text, or Methods section.

| n/a | Confirmed | |
|---|---|---|
| ☐ | ☒ | The exact sample size (*n*) for each experimental group/condition, given as a discrete number and unit of measurement |
| ☐ | ☒ | A statement on whether measurements were taken from distinct samples or whether the same sample was measured repeatedly |
| ☒ | ☐ | The statistical test(s) used AND whether they are one- or two-sided<br>*Only common tests should be described solely by name; describe more complex techniques in the Methods section.* |
| ☒ | ☐ | A description of all covariates tested |
| ☒ | ☐ | A description of any assumptions or corrections, such as tests of normality and adjustment for multiple comparisons |
| ☒ | ☐ | A full description of the statistical parameters including central tendency (e.g. means) or other basic estimates (e.g. regression coefficient) AND variation (e.g. standard deviation) or associated estimates of uncertainty (e.g. confidence intervals) |
| ☒ | ☐ | For null hypothesis testing, the test statistic (e.g. $F$, $t$, $r$) with confidence intervals, effect sizes, degrees of freedom and $P$ value noted<br>*Give P values as exact values whenever suitable.* |
| ☒ | ☐ | For Bayesian analysis, information on the choice of priors and Markov chain Monte Carlo settings |
| ☒ | ☐ | For hierarchical and complex designs, identification of the appropriate level for tests and full reporting of outcomes |
| ☒ | ☐ | Estimates of effect sizes (e.g. Cohen's *d*, Pearson's *r*), indicating how they were calculated |

*Our web collection on statistics for biologists contains articles on many of the points above.*

## Software and code

Policy information about availability of computer code

| Data collection | Tools and software for data collection have been described in: Yin, W., Brittain, D., Borseth, J. et al. A petascale automated imaging pipeline for mapping neuronal circuits with high-throughput transmission electron microscopy. Nat Commun11, 4949 (2020). https://doi.org/10.1038/s41467-020-18659-3<br><br>The software tools used to stitch and align the dataset is available in our github repository https://github.com/AllenInstitute/asap-modules. The volume assembly process is entirely based on image meta-data and transformations manipulations and is supported by the Render service (https://github.com/saalfeldlab/render). |
|---|---|
| Data analysis | Code for analysis and generation of figures was generated in Python Jupiter notebooks and will be available in https://github.com/AllenInstitute/MicronsFunctionalConnectomics<br>CAVE analysis infrastructure available at: https://github.com/CAVEconnectome |

For manuscripts utilizing custom algorithms or software that are central to the research but not yet described in published literature, software must be made available to editors and reviewers. We strongly encourage code deposition in a community repository (e.g. GitHub). See the Nature Portfolio guidelines for submitting code & software for further information.

## Data

Policy information about <u>availability of data</u>
All manuscripts must include a <u>data availability statement</u>. This statement should provide the following information, where applicable:

- Accession codes, unique identifiers, or web links for publicly available datasets
- A description of any restrictions on data availability
- For clinical datasets or third party data, please ensure that the statement adheres to our <u>policy</u>

> Image and segmentation data is available via https://www.microns-explorer.org.

## Research involving human participants, their data, or biological material

Policy information about studies with <u>human participants or human data</u>. See also policy information about <u>sex, gender (identity/presentation), and sexual orientation</u> and <u>race, ethnicity and racism</u>.

| | |
|---|---|
| Reporting on sex and gender | N/A |
| Reporting on race, ethnicity, or other socially relevant groupings | N/A |
| Population characteristics | N/A |
| Recruitment | N/A |
| Ethics oversight | N/A |

Note that full information on the approval of the study protocol must also be provided in the manuscript.

# Field-specific reporting

Please select the one below that is the best fit for your research. If you are not sure, read the appropriate sections before making your selection.

☒ Life sciences          ☐ Behavioural & social sciences          ☐ Ecological, evolutionary & environmental sciences

For a reference copy of the document with all sections, see <u>nature.com/documents/nr-reporting-summary-flat.pdf</u>

# Life sciences study design

All studies must disclose on these points even when the disclosure is negative.

| | |
|---|---|
| Sample size | No explicit sample size calculation was performed. The spatial extent of sampling was chosen based on the goal of including both primary and secondary visual areas |
| Data exclusions | Neurons without proofread axons were excluded from the analysis in Figure 7 |
| Replication | No experimental replication was performed. |
| Randomization | No randomization was performed |
| Blinding | No blinding was performed |

# Reporting for specific materials, systems and methods

We require information from authors about some types of materials, experimental systems and methods used in many studies. Here, indicate whether each material, system or method listed is relevant to your study. If you are not sure if a list item applies to your research, read the appropriate section before selecting a response.

## Materials & experimental systems

| n/a | Involved in the study |
|-----|----------------------|
| ☒ | Antibodies |
| ☒ | Eukaryotic cell lines |
| ☒ | Palaeontology and archaeology |
| ☐ | ☒ Animals and other organisms |
| ☒ | Clinical data |
| ☒ | Dual use research of concern |
| ☒ | Plants |

## Methods

| n/a | Involved in the study |
|-----|----------------------|
| ☒ | ChIP-seq |
| ☒ | Flow cytometry |
| ☒ | MRI-based neuroimaging |

## Animals and other research organisms

Policy information about [studies involving animals](); [ARRIVE guidelines]() recommended for reporting animal research, and [Sex and Gender in Research]()

| Laboratory animals | Slc17a7-IRES2-Cre-D knock-in mice (Jackson Laboratory, Stock No. 023527) and Ai162 mice (Jackson Laboratory, Stock No. 031562) |
|---|---|
| Wild animals | N/A |
| Reporting on sex | Male mouse |
| Field-collected samples | N/A |
| Ethics oversight | All procedures were approved by the Institutional Animal Care and Use Committee at Allen Institute of Brain Science or Baylor College of Medicine. |

Note that full information on the approval of the study protocol must also be provided in the manuscript.

## Plants

| Seed stocks | N/A |
|---|---|
| Novel plant genotypes | N/A |
| Authentication | N/A |

