## [Peer Review File · Nature]

Functional connectomics spanning multiple areas of mouse visual cortex

Corresponding Author: Dr Nuno Macarico da Costa

Version 0:

Reviewer comments:

Referee #1

(Remarks to the Author)

“Functional connectomics spanning multiple areas of mouse visual cortex” is a landmark achievement in systems neuroscience, an impressively large functional connectomics dataset with dense calcium imaging of a millimeter scale volume of mouse visual cortex combined with EM-based anatomical analysis with synaptic resolution. Using the publicly available resource connected to this dataset, interesting biorxiv papers have emerged. These studies have begun to illuminate visual processing, cellular morphology, and cortical connectivity in this significantly large region of visual cortex. More interesting papers will surely emerge.

This is an odd paper to review as a Research Article to Nature. This paper could be subsumed within the Methods of one of the biorxiv papers, also by members of the MICrONS Consortium. This manuscript does not describe any new scientific discovery or technical breakthrough (the more advanced techniques that they use are published elsewhere). With admirable transparency and thoroughness, the authors describe the intelligent and thoughtful way in which a breathtaking functional/structural dataset was acquired with the best available tools. This manuscript marks a milestone, and points the way forward to how similar datasets will be acquired in the future. But the job of mining new insights into neuroscience is left to others. I don't think the authors would disagree.

The question is whether this belongs as a Research Article in Nature as a scientific journal or as “breaking news” in Nature as a magazine. Left without room to “review” any hypothesis or scientific conclusion, I can only endorse the physiological measurements, connectomics, and data curation as first-rate. This paper will surely be very highly cited, Nature's figure of merit

Minor point:

The study describes “two events that led to consecutive section loss”. Might the authors be less obscure, and simply describe what happened. It would also be good to clearly describe the consequence of this loss. What fraction of neurons were untraceable through the gap?

Referee #2

(Remarks to the Author)

A. Summary of the key results

This is an effort of unprecedented scale to map structure-to-function, one of the holy grails of neuroscience. The authors use a mouse that went through a spectrum of visual stimuli (passive viewing), recorded its activity in multiple visual areas and, finally, dissect a volume combining primary and higher visual areas to map cell-to-cell connectivity via dense EM reconstruction. Importantly, several innovative methods are developed and implemented to get cell type-specific connectivity between transcriptomic types as opposed to spiny vs. non-spiny neurons. The main strength of this approach is how the data set is packaged and offered open access to the science community including tools to further analyze and scrutinize. In that sense, it is a remarkable feat and should be commended. One of the drawbacks of the current review is that reviewers do not

have access to the full set of papers coming out of this effort so that I cannot assess the novelty of the final data sets and findings, I am only in position to review the current manuscript and the methods implemented here. Therefore, I cannot comment whether/how the present effort uncovered fundamentally novel principles of visual processing.

B. Originality and significance: if not novel, please include reference

The current effort and the accompanying effort are novel in these sense that a connectomics data set of such size and detail is presented and analyzed for the first time. In addition, the tools and web interface accompanying the data set are unique and of central importance. While similar EM efforts have been undertaken in other species (and the authors cite the relevant citations) , this is the first time such an effort takes place for a mammalian cortical area – the authors claim that in other reports the data set reveals new computational principles putatively linking to function. Producing such an EM data set in the first place requires several methodological advances that are described in the methods section. Of fundamental importance is the ability to discern cell types based on morphological features – if true, this is of very high importance. Unfortunately, only hints of this important result are shown in the current manuscript (in figure 6).

C. Data & methodology: validity of approach, quality of data, quality of presentation

The validity of the approach and the data quality are of very high standards. Several technological innovations has to occur for the data set to be produced in the first place. The fact that manual and computer-based methods both resulted in comparatively high scores in co-registration speaks to the high quality of the raw data in the first place. If anything, some of the innovations remain obscured in the manuscript because they are not presented adequately (see also below). In that respect, having access to the other manuscripts of this series (especially the cell type-specific methodologies) would be invaluable as a reviewer so as to assess the entire work and familiarize with all novelties of the work.

D. Appropriate use of statistics and treatment of uncertainties

“Therefore, in addition to the functional scans, high-resolution (0.5 - 1 px/μm) structural volumes were acquired for registration with the subsequent EM data. At the end of each imaging day, individual imaging fields of the functional scans were independently registered into a structural stack (Fig 3b,c). This allowed us to target scans in subsequent sessions to optimize coverage across depth.”

Can the authors comment on the robustness and estimate the uncertainty of this approach?

E. Conclusions: robustness, validity, reliability

The paper is mostly an introduction and presentation of the methods (mainly EM reconstructions) and therefore does not try to make any strong conclusions except for the technological advancements achieved. For mapping the connectomics in mammalian microcircuits, this is an entirely novel data set resulting from multiple technological advancements. The fact that the data set is made accessible online with tools integrated to aid its analysis , speaks to the validity and robustness. Moreover, the comparatively high confidence in the manual and computer/ML-assisted mapping between connections, further supports that the current data set is annotated using the latest technologies.

F. Suggested improvements: experiments, data for possible revision

For such a large-scale effort, it is still important to mention potential shortcomings and future improvements. In the Discussion , the authors make comments like “However, it’s equally important to note that the automatic segmentation of axons is not as accurate. Therefore, it’s essential to be mindful of which processes have been proofread and which have not.” Two questions. First, which other limitations did the authors identify in their EM quest? Second, what sort of EM-related advancement or computational analysis techniques would tackle them ? What is the main methodological limitation to go from single-animal EM reconstruction to multiple animals ?

“This public resource (Fig 1) includes pyramidal neurons from all layers (underlined text contains links to public data)...”

How were these cell types characterized / identified ? In the text the ability to discern cell classes from EM stacks is presented as semi-intuitive yet this is highly non-standard. The authors should make an effort to explain how these classes were identified in their data otherwise it is left up to the reader to infer how this happened.

“In this respect, our work parallels another milestone of connectomics, the imminent completion of the *Drosophila* connectome 41–43; only 20% of the neuron types described in the EM connectome of the central brain were previously described in the literature 41.” In my opinion, the whole paragraph is ill-posed. The authors seem to put so much effort in attempt to convey the message that mammals are not flies. The fact that a detailed connectomics data set of the fly was recently published do not mean the current set is questioned for novelty. If anything, it renders the current work even more pressing since one can now compare fly vs. mouse connectomes head-to-head and hopefully uncover deep computational principles, differences and similarities. I rarely make this recommendation but I would advise the authors to revise and put their and the other team’s effort in a more positive light. In its current version , the paragraph seems to imply these efforts are in competition..

“functional digital twin”: a digital twin has been widely used and, in fact, often miss-used term. Why not stay close to the data and simply call the accompanying paper a “data-driven computational model”? Going over the accompanying paper (called a digital twin), by no means does this “foundational model” satisfy several conditions of a digital twin.. Therefore, I would avoid using this strong claim and keep close to the data.

“Further development of the MICrONS technologies should eventually render Crick’s dream not only possible, but routine.”

So are the authors implying that , in the future, we will do large-scale reconstructions and then automatically know “the way all its neurons are firing”? Because experience has shown this is not possible. What does the current approach change in that statement especially since important details (e.g. synaptic strength, time-evolution, etc.) is missing from the connectomics approach? In my opinion, this statement is far-fetched and no data is presented to support it. Please revise.

“The rough alignment process ensured global consistency within the dataset and accounted for images also corrected for locally varying misalignments such as scale differences and deformations between sections and aids the fine alignment process.”

Please offer statistics here about consistency, acceptance error and misalignment rate. Also how did this approach compare to the convolutional network approach the authors mention in the next paragraph? When did the authors opt from the first vs. the second approach or did they always apply both ?

How much proofreading was required for the entire data set? Are there metrics the authors can offer, e.g., proofreading time per cell class or even per cell ? The authors shows some very interesting data in figure S2, it would be nice to mention the

main findings in the main ms.

G. References: appropriate credit to previous work?

The citations are plenty and do justice to multiple contributors in the EM field.

H. Clarity and context: lucidity of abstract/summary, appropriateness of abstract, introduction and conclusions

How cell types were identified from the current data set is not well-presented in the current manuscript. It could be because this manuscript is accompanied by others where the details of identifying transcriptomic types will be expanded but, for such an interesting finding, I was expecting more to be revealed in the current manuscript.

Minor improvements:

“restart the ultramicrotome at a moment's notice if there was a risk of multiple” -> “immediately” ?

“imaged by a fleet of five customized automated ” -> “imaged by five customized automated ” ?

Version 1:

Reviewer comments:

Referee #1

(Remarks to the Author)

The authors have adequately addressed my small query. I now appreciate the value of this stand-alone-manuscript that will be the primary citation for the dataset (as opposed to the other papers that draw scientific conclusions). This now makes sense to me.

Dear Editor,

We are submitting our revised manuscript for consideration for publication in Nature as part of the MICrONS package. We are pleased that the reviewers have recognized the broad impact and significance of this landmark dataset. We have tried to mitigate the confusion regarding the nature of the manuscript within the package, emphasizing that we consider this manuscript as the primary citation for the dataset in all current and future publications related to scientific findings or technical developments. We appreciate the valuable insight and advice provided by the reviewers, and we have incorporated their suggestions into this revision. Following their advice, we added a section on cell types as well as a new Figure 7, that further describes the use of integrating synapse connectivity and cell types. We hope that these revisions better align with their expectations and we address their comments below.

Thank you very much

Nuno Macarico da Costa, Forrest Collman, Jake Reimer, Xaq Pitkow, Andreas S. Tolias, Sebastian Seung, Clay Reid

Referee #1

R#1-1: *“Functional connectomics spanning multiple areas of mouse visual cortex” is a landmark achievement in systems neuroscience, an impressively large functional connectomics dataset with dense calcium imaging of a millimeter scale volume of mouse visual cortex combined with EM-based anatomical analysis with synaptic resolution. Using the publicly available resource connected to this dataset, interesting biorxiv papers have emerged. These studies have begun to illuminate visual processing, cellular morphology, and cortical connectivity in this significantly large region of visual cortex, . More interesting papers will surely emerge.*

This is an odd paper to review as a Research Article to Nature. This paper could be subsumed within the Methods of one of the biorxiv papers, also by members of the MICrONS Consortium. This manuscript does not describe any new scientific discovery or technical breakthrough (the more advanced techniques that they use are published elsewhere). With admirable transparency and thoroughness, the authors describe the intelligent and thoughtful way in which a breathtaking functional/structural dataset was acquired with the best available tools. This manuscript marks a milestone, and points the way forward to how similar datasets will be acquired in the future. But the job of mining new insights into neuroscience is left to others. I don't think the authors would disagree.

The question is whether this belongs as a Research Article in Nature as a scientific journal or as “breaking news” in Nature as a magazine. Left without room to “review”

any hypothesis or scientific conclusion, I can only endorse the physiological measurements, connectomics, and data curation as first-rate. This paper will surely be very highly cited, Nature's figure of merit

R#1-1 Answer:

We are grateful to the reviewer for their positive and encouraging feedback, as well as for recognizing the landmark impact of the dataset. Our primary objective with this manuscript was to describe a valuable resource for the scientific community. We fully concur with the reviewer's assessment that "the job of mining new insights into neuroscience is left to others". While we do engage in some of this exploration in our accompanying manuscripts (Schneider-Mizell et al, Gamlin et al, Bodor et al, Ellabady et al, Weis et al), where we present novel insights into the connectivity of various cell types, we also believe it is crucial to include the necessary information for others to conduct their own mining. Furthermore, while the individual techniques might be covered in other papers, the combination of these techniques is also unprecedented and novel, and this is the only paper that describes the combination and can serve as the primary citation for the dataset. As mentioned above, we submitted other manuscripts where we provide some of the neuroscience insights that the reviewer hoped to see, and in this revision, we have made a more concerted effort to integrate these findings into the text.

We believe that by providing both the dataset and the initial neuroscience findings, we are facilitating a dual-purpose contribution: enabling others to mine the data for further insights and serving as a clear citation for the dataset. To further support others mining the data we just started a program providing proofreading and annotation systems for scientific question driven from the community on topics as diverse as cortical computation, cell biology, neurovascular interaction and glia biology (www.microns-explorer.org/vortex/). All of these applications of the dataset convinced us that a primary citation manuscript made the most sense.

Minor point:

R#1-2: *The study describes "two events that led to consecutive section loss". Might the authors be less obscure, and simply describe what happened. It would also be good to clearly describe the consequence of this loss. What fraction of neurons were untraceable through the gap?*

R#1-2 Answer:

We have updated the methods section of the manuscript to provide additional details regarding the regions affected by consecutive section loss.

“The first event that led to consecutive section loss was due to sections being collected onto apertures with damaged films. To prevent this from happening again, we installed a camera that monitors the aperture before collection. The second event was due to an accident where the knife bumped the block and nicked a region near the edge of the ROI”

In the 'The Electron Microscopy volume' section, we previously mentioned our selection of sections for image processing and segmentation, focusing on those after the consecutive loss (sections 7,931–27,904). Since we did not trace through the gap we don't know which fraction of neurons would be untraceable. Now, we have also clarified this aspect in the 'Ultrathin Sectioning' section.

Referee #2

R#2-1: *This is an effort of unprecedented scale to map structure-to-function, one of the holy grails of neuroscience. The authors use a mouse that went through a spectrum of visual stimuli (passive viewing), recorded its activity in multiple visual areas and, finally, dissect a volume combining primary and higher visual areas to map cell-to-cell connectivity via dense EM reconstruction. Importantly, several innovative methods are developed and implemented to get cell type-specific connectivity between transcriptomic types as opposed to spiny vs. non-spiny neurons. The main strength of this approach is how the data set is packaged and offered open access to the science community including tools to further analyze and scrutinize. In that sense, it is a remarkable feat and should be commended. One of the drawbacks of the current review is that reviewers do not have access to the full set of papers coming out of this effort so that I cannot assess the novelty of the final data sets and findings, I am only in position to review the current manuscript and the methods implemented here. Therefore, I cannot comment whether/how the present effort uncovered fundamentally novel principles of visual processing.*

R#2-1 Answer:

We appreciate the reviewer's positive and constructive feedback. All the manuscript of the package are available publicly on bioRxiv and below we present a list with links:

1. Functional connectomics spanning multiple areas of mouse visual cortex (<https://www.biorxiv.org/content/10.1101/2021.07.28.454025v3>)
(This paper)
2. Cell-type specific inhibitory circuits in a connectomic census of visual cortex (<https://www.biorxiv.org/content/10.1101/2023.01.23.525290v2>)
(in press at Nature)

3. Functional connectomics reveals general wiring rule in mouse visual cortex (<https://www.biorxiv.org/content/10.1101/2023.03.13.531369v1>)
(in revision at Nature)
4. Quantitative Census of Local Somatic Features in Mouse Visual Cortex (<https://www.biorxiv.org/content/10.1101/2022.07.20.499976v1>)
(in press at Nature)
5. Petascale neural circuit reconstruction: automated methods (<https://www.biorxiv.org/content/10.1101/2021.08.04.455162v1>)
(in revision at Nature Methods)
6. Connectome Annotation Versioning Engine (<https://www.biorxiv.org/content/10.1101/2023.07.26.550598v1>)
(in press at Nature Methods)
7. Integrating EM and Patch-seq data: Synaptic connectivity and target specificity of predicted Sst transcriptomic types (<https://www.biorxiv.org/content/10.1101/2023.03.22.533857v1>)
(in revision at Nature)
8. NEURD: automated proofreading and feature extraction for connectomics (<https://www.biorxiv.org/content/10.1101/2023.03.14.532674v1>)
(in press at Nature)
9. Large-scale unsupervised discovery of excitatory morphological cell types in mouse visual cortex (<https://www.biorxiv.org/content/10.1101/2022.12.22.521541v1>)
(in press at Nature Communications)
10. Towards a Foundation Model of the Mouse Visual Cortex (<https://www.biorxiv.org/content/10.1101/2023.03.21.533548v1>)
(in revision at Nature)
11. Bipartite invariance in mouse primary visual cortex (<https://www.biorxiv.org/content/10.1101/2023.03.15.532836v1>)
(in revision at Nature)
12. Pattern completion and disruption characterize contextual modulation in mouse visual cortex (<https://www.biorxiv.org/content/10.1101/2023.03.13.532473v1>)
(in revision at Nature)
13. The Synaptic Architecture of Layer 5 Thick Tufted Excitatory Neurons in the Visual Cortex of Mice (<https://www.biorxiv.org/content/10.1101/2023.10.18.562531v1>)
(currently in revision at Nature Neuroscience)

R#2-2: *The current effort and the accompanying effort are novel in these sense that a connectomics data set of such size and detail is presented and analyzed for the first time. In addition, the tools and web interface accompanying the data set are unique and of central importance. While similar EM efforts have been undertaken in other species (and the authors cite the relevant citations) , this is the first time such an effort*

takes place for a mammalian cortical area – the authors claim that in other reports the data set reveals new computational principles putatively linking to function. Producing such an EM data set in the first place requires several methodological advances that are described in the methods section. Of fundamental importance is the ability to discern cell types based on morphological features – if true, this is of very high importance. Unfortunately, only hints of this important result are shown in the current manuscript (in figure 6).

R#2-2 Answer: We agree with the reviewer and perform several cell type analysis, both of excitatory and inhibitory types. While we show that we can use morphology to discern cell types, we also demonstrate that connectivity, including synaptic features, can further validate or refine morphological classification. The results are presented in separate accompanying manuscripts:

(1) In the study conducted by Schneider-Mizell et al., we analyzed a cortical column and introduced a novel excitatory cell type classification method that takes into account both synaptic and morphological properties of dendritic arbors. This classification was validated through the examination of differential input from inhibitory neurons. Notably, our results revealed a remarkable level of specificity in the connectivity patterns of interneurons. Furthermore, we present a data-driven organizing principle for cortical inhibition. This principle highlighted that small yet diverse groups of inhibitory cell types collectively target the same neuronal subpopulations. This finding suggests a high degree of coordinated inhibition across various cell types and subclasses, shedding light on the intricate dynamics of cortical inhibition. In addition to these findings, our study introduced a novel class of interneurons. Unlike the well-studied VIP interneurons, this new class targets basket cells instead of SST interneurons. This discovery presents a distinct pathway with the potential to modulate fundamental inhibitory computations, such as gain control or population sparsity, in a previously unexplored manner. It represents an exciting avenue for further research and understanding of neural circuitry.

(2) In Elabbady et al. we leverage local subcellular features within the somatic region. These characteristics include features like nuclear infolding and somatic synaptic innervation. By analyzing these features, we can generate highly accurate predictions of cell types. What's noteworthy is that these predictions only necessitate minor proofreading of the neuron data. With these refined predictions in hand, we are able to extrapolate and predict cell types across tens of thousands of neurons spanning different cortical depths and cortical areas. This approach has been essential to scale the mapping of cortical connectivity across cell types.

(3) In Weis et al., we employed graph-based machine learning techniques to extract a morphological "bar code" that characterizes over 30,000 excitatory neurons spanning three distinct cortical areas (V1, RL, and AL).

Our findings revealed interesting trends within these neurons. Specifically, we observed that neurons located in layer 2/3 displayed a consistent pattern of decreasing dendritic arbor width and smaller tuft size as we moved deeper into the cortical layers. In layer 4, we noticed that nontufted neurons were predominantly concentrated in the primary visual cortex, whereas neurons with tufts were more prevalent in higher visual areas. Additionally, our investigation identified a unique group of layer 4 neurons in the primary visual cortex (V1) bordering layer 5. These neurons exhibited a tendency to extend their dendrites toward shallower layers while avoiding deeper layers. In summary, our study unveiled a significant degree of dendritic morphological diversity among excitatory neurons, both within and across cortical layers. However, this diversity generally followed a continuous spectrum, with only a few noteworthy exceptions, particularly in the deeper layers.

(4) In Gamlin et al., we establish a relationship between the connectivity data derived from the MICrONS datasets and the transcriptomics and electrophysiology obtained from Patch-seq experiments. This connection is established by leveraging common morphological features that are present in both datasets. To achieve this integration across different experimental modalities and cellular properties, we use a specific example: the well-defined morphological cells known as Martinotti cells. These cells are also known for being Somatostatin (Sst+) positive. By using morphology as a bridge, we are able to align and relate the findings across these diverse data sources, facilitating a more comprehensive understanding of cell types.

We have also now added a new section on cell types as well as a new Figure 7 that demonstrate the use of cell types to make connectivity across multiple hops in a circuit. See also reply R#2.7

R#2-3: The validity of the approach and the data quality are of very high standards. Several technological innovations has to occur for the data set to be produced in the first place. The fact that manual and computer-based methods both resulted in comparatively high scores in co-registration speaks to the high quality of the raw data in the first place. If anything, some of the innovations remain obscured in the manuscript because they are not presented adequately (see also below). In that respect, having access to the other manuscripts of this series (especially the cell type-specific methodologies) would be invaluable as a reviewer so as to assess the entire work and familiarize with all novelties of the work.

R#2-3 Answer:

We have made all manuscripts available in bioRxiv and list them above in our to comment R#2-1.

R#2-4: *“Therefore, in addition to the functional scans, high-resolution (0.5 - 1 px/μm) structural volumes were acquired for registration with the subsequent EM data. At the*

end of each imaging day, individual imaging fields of the functional scans were independently registered into a structural stack (Fig 3b,c). This allowed us to target scans in subsequent sessions to optimize coverage across depth.”

Can the authors comment on the robustness and estimate the uncertainty of this approach?

R#2-4 Answer:

The original purpose of the 2p-field → 2p-stack registration was to provide a common coordinate framework in 2p-stack coordinates for measuring the position of imaging fields (2p-fields) across days. Fig 1 shows the scale of a single 2p-field in its registered position inside the 2p-stack.

Fig 1: Depiction of a 2p-field registered inside the 2p-stack

One way to assess the quality of 2p-field → 2p-stack registration is by manual inspection, comparing the features on the summary image of the 2p-field (height x width, collapsed over time) to the stack at its registered position. To this end we have made publicly available a Neuroglancer link on the project website which contains every imaging field in its registered position in the stack. By toggling the imaging field layers on and off, and comparing cell bodies and vessels in the field vs the stack, one can manually inspect the quality of the registration at any region of interest.

To quantify the success of 2p-field → 2p-stack registration we also use the correlation between the 2p-field summary image and the image extracted from the 2p-stack at the coordinates of the registered 2p-field. A perfect registration should give a high correlation, but less than 1 because of variations between datasets. Fig 2 shows the distribution of correlation scores across all imaging fields in this dataset. The bulk of the imaging fields have high correlations (> 0.6) and those that don't are usually from very superficial or very deep imaging fields where registration is more challenging. Fig 3 shows the distribution of correlation scores by depth.

Fig 2: distribution of correlation scores across imaging fields

Fig 3: correlation score by depth

Beyond this however, the 2p-stack was also used to coregister the 2p data with the EM data. The quality of both of these registration steps (2p-field → 2p-stack and 2p-stack → EM) was crucial for the ability of humans to manually associate 2p functional ROI's (functional units) to neurons in the EM. Therefore, one way to measure the robustness of the 2p-field → 2p-stack registration is to refer to the quality of the coregistration. Supplemental Figure 4 quantifies the success of coregistration by leveraging the fact that some functional units obtained from different scans were independently coregistered to the same EM neuron, indicating that the neuron was scanned multiple times. If these functional units did come from the same cell, then the expectation is that their signal correlations (correlations of activity traces after averaging over repeated stimulus presentations) should be higher than the signal correlation with the next closest functional unit that competed for assignment during the matching

step. The results show that for a vast majority of these cases, the coregistered functional units not only have higher signal correlations than the control units but also have a high signal correlation overall. From this we conclude that the entire workflow from functional data to EM, including 2p-field→ 2p-stack registration was robust.

With the coregistered cells, we can also see that field correlation scores on the lower end (<0.6) were of sufficient quality such that tens to hundreds of cells could be confidently matched. Fig 4 shows the number of coregistered cells per field vs field correlation score. Lastly, to indicate whether the 2p-field→ 2p-stack registration was robust enough to fulfill the original purpose to appropriately space imaging fields across cortical depth, in Fig 5 we plot the mean depth and std of coregistered cells per imaging field sorted by depth relative to pia and white matter boundaries. The relatively even spacing of these points tiling depth indicates that the original purpose was fulfilled.

Fig 4: # coregistered cells per field vs correlation score

Fig 5: mean depth (std) of all coregistered cells per imaging field, sorted by depth, with approximate depths of pia and white matter (wm) indicated by horizontal lines

R#2-5: *The paper is mostly an introduction and presentation of the methods (mainly EM reconstructions) and therefore does not try to make any strong conclusions except for the technological advancements achieved. For mapping the connectomics in mammalian microcircuits, this is an entirely novel data set resulting from multiple technological advancements. The fact that the data set is made accessible online with tools integrated to aid its analysis, speaks to the validity and robustness. Moreover, the comparatively high confidence in the manual and computer/ML-assisted mapping between connections, further supports that the current data set is annotated using the latest technologies.*

R#2-5 Answer:

We thank the reviewers for their comments, and want to emphasize our strong commitment to ensuring the online accessibility of both the data and the analysis tools. Currently, we are actively supporting other scientific projects that utilize this dataset by offering proofreading and annotation services (www.microns-explorer.org/vortex/).

F. Suggested improvements: experiments, data for possible revision

R#2-6: For such a large-scale effort, it is still important to mention potential shortcomings and future improvements. In the Discussion, the authors make comments like “However, it's equally important to note that the automatic segmentation of axons is not as accurate. Therefore, it's essential to be mindful of

which processes have been proofread and which have not.” Two questions. First, which other limitations did the authors identify in their EM quest? Second, what sort of EM-related advancement or computational analysis techniques would tackle them? What is the main methodological limitation to go from single-animal EM reconstruction to multiple animals?

R#2-6 Answer: We acknowledge we could have written more about the challenges and limitations we have experienced in the dataset, and have added to the discussion reflecting on the bottlenecks and likely areas for improvement. That said, we don't think there are any methodological limitations to go from single-animal EM to multiple animals. We have in fact already completed acquisition, segmentation and proofreading of a second mm³ scale volume at the Allen Institute that we are hoping to release and publicize soon.

R#2-7: “This public resource (Fig 1) includes pyramidal neurons from all layers (underlined text contains links to public data)...” How were these cell types characterized / identified? In the text the ability to discern cell classes from EM stacks is presented as semi-intuitive yet this is highly non-standard. The authors should make an effort to explain how these classes were identified in their data otherwise it is left up to the reader to infer how this happened.

R#2-7 Answer: We have added a section on cell types that explains how the data presented here has been used to identify different cell types. In this manuscript, the description is necessarily brief due to space constraints. However, several companion manuscripts (see below) provide detailed accounts of the various methods used. All of these methods rely on anatomical features derived from EM data but take different approaches. Below is the section from the main text:

“ Two projects (Schneider-Mizell et al accepted in Nature, and Weis) et all under revision) applied data-driven methods to dendritic reconstructions to characterize excitatory neurons across cortical depth and visual areas, revealing intralaminar subtypes and interareal differences in populations. Another study linked transcriptomic types of inhibitory neurons to EM reconstructions, establishing a proof of concept for linking molecular cell types to anatomical cell types that use morphology and synapse connectivity (Gamlin et al, under revision). While these studies used proofread or post-processed neuronal reconstructions, not all segmented neurons in the dataset were amenable to such analysis due to truncation by dataset boundaries or segmentation quality. To push cell typing even in such difficult cases, a fourth study showed that key features of the soma and nucleus of a cell alone was

sufficient to predict cell classes such as glia, excitatory neuron, or inhibitory neuron as well as subclasses such as basket cells versus bipolar cells or microglia versus oligodendrocytes or identify similar cells to a cell of interest (Elabbady, accepted in Nature). Together, these approaches enable not only matching known cell types with EM neurons, but using the EM data to discover new cell types. ”.

Using these classifications of cell types, one of the consistent findings across multiple studies is the specificity of cell connectivity. The data presented in this manuscript, along with the companion paper, provide a foundation to expand upon both classical and contemporary works that use connectivity as a defining characteristic of cell types. We have begun to explore this concept further in one of the companion manuscripts (Schneider-Mizell et al., accepted in Nature). See also reply R#2.2

R#2-8:“In this respect, our work parallels another milestone of connectomics, the imminent completion of the *Drosophila* connectome 41–43; only 20% of the neuron types described in the EM connectome of the central brain were previously described in the literature 41.” In my opinion, the whole paragraph is ill-posed. The authors seem to put so much effort in attempt to convey the message that mammals are not flies. The fact that a detailed connectomics data set of the fly was recently published do not mean the current set is questioned for novelty. If anything, it renders the current work even more pressing since one can now compare fly vs. mouse connectomes head-to-head and hopefully uncover deep computational principles, differences and similarities. I rarely make this recommendation but I would advise the authors to revise and put their and the other team’s effort in a more positive light. In its current version , the paragraph seems to imply these efforts are in competition.

R#2-8 Answer: In response to this criticism, and for the sake of brevity, we have completely removed the paragraph from the Introduction. A comparison with *Drosophila* remains in the Discussion. Our statement “only 20% of the neuron types described in the EM connectome of the central brain were previously described in the literature” was actually meant to praise this work by pointing out how many new cell types were discovered in a single study. It was not meant to be competitive, but we understand that the statement was unclear and confusing. We fully agree that fly and mouse circuits share common principles, and that the differences are also fascinating.

R#2-9: “functional digital twin”: a digital twin has been widely used and, in fact, often miss-used term. Why not stay close to the data and simply call the accompanying paper a “data-driven computational model”? Going over the accompanying paper (called a digital twin), by no means does this “foundational model” satisfy several

conditions of a digital twin.. Therefore, I would avoid using this strong claim and keep close to the data.

R#2-9 Answer: We have expunged the terms “digital twin” and “foundational model” from the Introduction, and instead clarified our description of the model and its applications while attempting to steer clear of jargon that might be misconstrued. However, in the pages of this private response, we would like to take the opportunity to explain what we originally meant by these terms. We recognize that "digital twin" is a term that is often used broadly and sometimes misapplied. However, our decision to use "functional digital twin" is intentional, aiming to highlight our model's specific goal: replicating functional activation patterns in the MICrONS mouse brain enabling new in silico experiments. This term accurately reflects both the purpose and the capabilities of our model. As described in our accompanying paper (Wang et al. link above), our model excels in predicting responses across various visual stimulus domains, owing to its extensive training data from a cohort of mice. Indeed, a key strength of our model is its ability to generalize to novel stimulus types, such as static images, random dots, and coherent noise, which it was not explicitly trained on. Beyond neural prediction, we have also utilized the digital twin to correlate functional properties with connectivity and morphological characteristics of cell types. This application showcases the model's wide-ranging utility for multiple downstream applications, and positions our model similarly to a foundation model in AI, emphasizing its extensive training base and adaptability for diverse downstream applications and tasks.

The term digital twin is to highlight the way we envision its use by the community, i.e. to run new in silico experiments with novel visual stimuli to derive functional properties that then can be related to structure and connectivity given the uniqueness of this data set. For validating new in silico findings, our approach is exemplified in our companion papers, such as the method for mapping center-surround interactions in neurons in natural stimulus conditions (Fu et al.) . By conducting parallel experiments in separate cohorts of mice and their digital twins, we demonstrate a strong correlation between in vivo and in silico center surround interactions characterization. This process builds confidence in using the digital twin of the MICrONS mouse for mapping new neuronal properties and correlating them with anatomy and demonstrates how the community can use the deep learning data-driven digital twin pipeline approach we developed.

To further support innovative research using the digital twin of the MICrONS mouse, we are in the process of developing an API designed to streamline in silico experiments. This initiative aims to make these digital twins more accessible and user-friendly, especially for researchers who may not have extensive experience in deep learning. Our goal is to enable a wider range of scientists to engage with and benefit from our MICrONS data and models. We recognize that our model, like any tool in the scientific arsenal, is subject to continuous improvement. We are actively pursuing enhancements in network architectures and data integration, as elaborated in our revised manuscript. In line with our commitment to open science and collaborative progress, we have already released a significant cohort of high-entropy stimulus data even before publication, open-source, from many mice. This release is intended to encourage and enable the broader research community to train new models and to refine and surpass the performance of our current model. We want to emphasize that our current model, while representing the pinnacle of our capabilities at this moment, is not the final iteration. We anticipate and welcome future advancements and improvements from the scientific community, leveraging the data and tools we have provided.

Thank you for your guidance in helping us present our work more clearly and accurately.

R#2-10:“Further development of the MICrONS technologies should eventually render Crick’s dream not only possible, but routine.” So are the authors implying that , in the future, we will do large-scale reconstructions and then automatically know “the way all its neurons are firing”? Because experience has shown this is not possible. What does the current approach change in that statement especially since important details (e.g. synaptic strength, time-evolution, etc.) is missing from the connectomics approach? In my opinion, this statement is far-fetched and no data is presented to support it. Please revise.

R#2-10 Answer:

We have expunged this sentence from the manuscript, since it implied to the reviewer a meaning that we did not intend at all. We would still like to explain to the reviewer what we did mean. We interpreted Crick’s dream as being about observation, not inference. He wished to observe the activity of a

population of neurons, and also observe their connectivity. He wanted data, and we are attempting to provide it. We are not implying that the wiring diagram by itself is sufficient for inferring or predicting the firing patterns of neurons. The goal of this paper is to observe both. We are sorry for the confusion. Our study demonstrates the feasibility of observing both activity and connectivity. That doesn't mean we will suddenly understand everything. Nor is it a claim that we will measure everything.

R#2-11:“The rough alignment process ensured global consistency within the dataset and accounted for images also corrected for locally varying misalignments such as scale differences and deformations between sections and aids the fine alignment process.”

Please offer statistics here about consistency, acceptance error and misalignment rate. Also how did this approach compare to the convolutional network approach the authors mention in the next paragraph? When did the authors opt from the first vs. the second approach or did they always apply both ?

R#2-11 Answer:

The image alignment pipeline consists of two main components: a coarse alignment step followed by a fine alignment step. It's important to note that these two components are not alternative approaches from which one can choose; rather, they represent sequential steps in the alignment process.

In the first step, known as coarse alignment, a global deformation of the stitched sections is performed. This process results in a volume that is roughly aligned. This initial alignment is achieved by generating point correspondences from 2D stitched section montages, which are materialized at a 1% scale.

The second step, fine alignment, builds upon the rough alignment achieved in the first step. Fine alignment is accomplished using a convolutional network.

To address the reviewer's request for statistics in the methods section, we measure the quality of global 3D alignment by computing angular residuals between pairs of sections (Mahalingam et al. 2022). This computation is limited to sections within a distance of three sections along the z-axis. The angular residual is calculated based on the point correspondences between a section and its neighboring sections.

Specifically, it is defined as the angle between two vectors formed by a point coordinate from the first section and its corresponding point coordinate from a neighboring section. The origin of these two vectors is set at the centroid of the point coordinates of the first section. The median of these angular residuals serves as a quality metric for assessing the global 3D alignment.

For a more detailed explanation of this metric, please refer to our previous publication (Mahalingam et al. 2022). In Figure 6, we provide a visual representation of the alignment process. Notably, in Figure 6.f, which pertains to the dataset described in this manuscript (Mouse 1), shows that the median angular residuals are very low, indicating a high-quality global 3D alignment. While this rough alignment is very good it is not sufficient for automatic segmentation, and hence step 2 with the convolution network.

Figure 6 from Mahalingam et al 2022 (a) View of the global nonlinear 3D aligned volume from the xz plane. The figure shows the view of the global nonlinear 3D alignment of the sections with the volume sliced at position marked by the red lines in (e). (b) View of the global nonlinear 3D aligned volume from the yz plane. Figure shows the view of the volume sliced at position marked by the red lines in (e). (c) Zoomed-in area from (a) showing the quality of global nonlinear 3D alignment in the xz plane. (d) Zoomed-in area from (b) showing the quality of global nonlinear 3D alignment in the yz plane. (e) Maximum pixel intensity projection of the global nonlinear 3D aligned sections in the z-axis showing the overall alignment of sections within the volume. The red lines represent the slicing location in both xz and yz plane for the cross-sectional slices shown in (a) and (b). (f) A plot showing the distribution of median angular residuals from serial sections grouped by the dataset.

R#2-12:How much proofreading was required for the entire data set? Are there metrics the authors can offer, e.g., proofreading time per cell class or even per cell ? The authors shows some very interesting data in figure S2, it would be nice to mention the main findings in the main ms.

R#2-12 Answer: In total, we've made 1,046,656 edits to the dataset. These edits encompass both manual adjustments made during proofreading and automated edits using the approach outlined by Celii et al. Some edits were aimed at fully extending a neuron's arbor, while others focused on proofreading specific portions of an arbor, such as the dendrite or axon. We elaborate on this process in the "Proofreading" section of both the Results and Methods.

The number of edits needed for proofreading varies depending on factors like cell class, arbor size, profile thickness, and location within the dataset. On average, a trained proofreader can produce 400-600 edits per week. The number of edits required per fully extended cell ranges from 100 to 1000 (see figure s2). So this means a cell can be completed in a few days to a week and a half of work.

G. References: appropriate credit to previous work?

The citations are plenty and do justice to multiple contributors in the EM field.

H. Clarity and context: lucidity of abstract/summary, appropriateness of abstract, introduction and conclusions

R#2-13:

How cell types were identified from the current data set is not well-presented in the current manuscript. It could be because this manuscript is accompanied by others where the details of identifying transcriptomic types will be expanded but, for such an interesting finding, I was expecting more to be revealed in the current manuscript.

R#2-13 Answer: The manuscript is indeed accompanied by several other studies that expand on the topic of cell types focusing on morphology and connectivity ((Schneider-Mizell et al, Bodor et al, Ellabady et al, Weis et al), but also on the link to Transcriptomic cell types (Gamlin et al). See also reply R2.2 and R2.7

R#2-14:

Minor improvements:

“restart the ultramicrotome at a moment's notice if there was a risk of multiple” -> “immediately” ?

“imaged by a fleet of five customized automated ” -> “imaged by five customized automated ” ?

R#2-14 Answer: We have modified the text according to the reviewer suggestions